# Betrixaban activates cGAS and ERVs to promote dual nucleic-sensing antiviral immunity

Xingyu Chen[1,2], Yang Zhao [1,2 ✉], Yunfei Xie[1], Tianyi Liu[1], Haocheng Wang[1], Xiao Wang[1 ✉], Xuefei Guo[1 ✉] & Fuping You[1 ✉]

## Abstract

**Broad-spectrum host-directed antivirals are urgently needed, as virus-targeted drugs often suffer from narrow specificity and rapid resistance. Here, we reported that Betrixaban (BT), an FDA-approved oral Factor Xa inhibitor, induced a robust antiviral state through dual innate immune pathways. Mechanistically, we identified BT as the first small molecule to directly bind and activate the DNA sensor cGAS to induce cGAMP production. Concurrently, BT inhibited histone deacetylases (HDACs), leading to chromatin de-repression of endogenous retroviruses (ERVs) and production of immunostimulatory double-stranded RNA (dsRNA) that engaged RIG-I/MDA5. These combined signal cascades triggered strong type I interferon responses and conferred broad-spectrum antiviral protection against RNA and DNA viruses in vitro and in vivo. These findings unveil a unique host-directed antiviral strategy wherein a small-molecule drug engages dual nucleic acid-sensing pathway, and suggest repurposing BT as an orally broad-spectrum antiviral.**

**Keywords** Betrixaban; cGAS-STING; RIG-I/MDA5; Endogenous Retroviruses; Host-directed Antiviral
**Subject Categories** Immunology; Microbiology, Virology & Host Pathogen Interaction; Pharmacology & Drug Discovery

## Introduction

Viral infections remain a major global health challenge (Artenie et al, 2025; Proal et al, 2025). The innate immune system is the first line of defense against viruses (Seth et al, 2006; Taniguchi et al, 2001). It detects pathogen nucleic acids, triggers type I interferon (IFN) and inflammatory responses (Alexopoulou et al, 2001; Collins et al, 2015). Cytosolic DNA is sensed by cGAS, which produces the second messenger cGAMP to activate STING and induce IFN-stimulated genes (Sun et al, 2013). Similarly, the helicase RIG-I-like receptors detect viral RNA, activating IRF3/NF-κB signaling to drive antiviral programs (Chiu et al, 2009; Kato

et al, 2005). These pathways provide broad protection against diverse DNA and RNA viruses.

Numerous strategies have been explored to pharmacologically boost innate antiviral immunity. For example, Inarigivir soproxil, a small-molecule agonist of RIG-I, produces significant reductions in viral load in chronic hepatitis B patients during a Phase II clinical trial (Yuen et al, 2023). STING agonists, such as natural cyclic dinucleotides and their analogs (ADU-S100, MK-1454), as well as non-nucleotide agonist (diABZI-4, Fangchinoline, MSA-2), are also under study as antiviral or vaccine adjuvants (Humphries et al, 2021; McIntosh et al, 2022; Meric-Bernstam et al, 2022; Skouboe et al, 2018; Wang et al, 2024). Non-nucleic cGAS agonists such as manganese have been reported to sensitize cGAS to cytosolic DNA and are essential for antiviral host defense (Wang et al, 2018).

In parallel, epigenetic drugs can induce a viral mimicry effect because DNA demethylating agents and histone deacetylase (HDAC) inhibitors de-repress endogenous retroviral (ERV) elements (Goyal et al, 2023; Roulois et al, 2015). DNA methyltransferase inhibitors such as 5-azacytidine and decitabine, together with HDAC inhibitors like trichostatin A, relieve epigenetic silencing at ERV loci, leading to the accumulation of double-stranded RNA that activates innate sensors(Chiappinelli et al, 2015; George et al, 2019; Roulois et al, 2015). Such approaches have shown promise in enhancing anti-tumor immunity and could be harnessed to fight viral infections.

Despite these advances, important gaps remain. First, no direct small-molecule agonist of cGAS has been reported to date, even though small-molecule cGAS inhibitors have only recently emerged (Lama et al, 2019). Next, no dual-pathway oral agonists exist; most innate agonists studied so far target only one receptor pathway and often require parenteral administration or exhibit limited stability. Last, many epigenetic modulators and viral mimetics have systemic effects and potential toxicities, such as myelosuppression, hepatotoxicity, and neurotoxicity, by dysregulating critical homeostatic pathways in healthy tissues. In summary, safe and orally bioavailable agents that broadly stimulate antiviral immunity across multiple innate sensors are urgently needed.

Here, we reported that Betrixaban (BT), an FDA-approved oral Factor Xa inhibitor, unexpectedly fulfilled these criteria (Connolly et al, 2013). We found that BT acted as a dual-acting innate immune stimulator. On the one hand, BT bound directly to cGAS and enhanced

[1]Institute of Systems Biomedicine, Department of Immunology, School of Basic Medical Sciences, Beijing Key Laboratory of Tumor Systems Biology, NHC Key Laboratory of Medical Immunology, Peking University Health Science Center, 100191 Beijing, China. [2]These authors contributed equally as first authors: Xingyu Chen, Yang Zhao. ✉E-mail: 2311210031@stu.pku.edu.cn; xwang2015@bjmu.edu.cn; guoxf@stu.pku.edu.cn; fupingyou@hsc.pku.edu.cn

its enzymatic activation, leading to robust cGAMP production and downstream IFN signaling. On the other hand, BT had HDAC inhibitory activity, which derepressed ERVs and generated dsRNA engaging RIG-I/MDA5. Overall, BT induced a host-protective antiviral state through DNA- and RNA-sensing pathways. This dual mechanism positioned BT as a novel broad-spectrum antiviral agent and provided proof-of-principle that repurposing existing drugs can yield systemic innate immune activators.

# Results

## Betrixaban establishes a host-protective antiviral state

During our daily experiments, we repeatedly observed the antiviral potential of Betrixaban (BT), a clinically approved oral Factor Xa inhibitor primarily studied for its anticoagulant activity and prevention of venous thromboembolism. We pre-treated RAW 264.7 cells, bone marrow-derived macrophages (BMDMs), HeLa cells, and HT1080 cells with BT, followed by infection with GFP-tagged vesicular stomatitis virus (VSV-GFP). At 24 h post-infection, we quantified the percentage of GFP-positive cells by flow cytometry (Fig. 1A; Appendix Fig. S1A). The results showed a dramatic reduction in GFP-positive cells in all BT-treated groups, indicating effective inhibition of VSV replication.

To further confirm this antiviral effect and to exclude viral type specificity, we infected BT-treated cells with distinct types of GFP-expressing viruses, including DNA viruses (Herpes simplex virus 1 [HSV-1] and Vaccinia virus [VACV]) and RNA viruses (Newcastle disease virus [NDV] and VSV) to visually assess viral replication. Fluorescence microscopy and flow cytometry analyses consistently demonstrated that BT treatment dramatically decreased GFP expression levels, indicating broad-spectrum suppression viral replication (Figs. 1B and EV1A).

We next tested this virus-resistant effect by adding different concentrations of BT, western blot analysis showed dose-dependent reductions in GFP protein expression across HSV, VACV, VSV, and NDV infections. We observed that RAW 264.7 cells achieved a complete protection at 100 μM against HSV, VSV, and NDV, with VACV exhibiting robust but incomplete inhibition (Fig. 1C). Moreover, we also treated RAW264.7, HT1080, HeLa, BMDMs, and human peripheral blood mononuclear cells (PBMCs) with BT and subsequently infected them with additional viruses, including VSV, HSV-1, Encephalomyocarditis virus (EMCV), Influenza A virus (IAV), Mouse hepatitis virus (MHV), and pseudotyped SARS-CoV-2 variants. We observed a significant decrease in viral replication across these cell types, as evidenced by marked reductions in viral RNA levels compared to untreated controls (Fig. 1D–F and EV1B,C), and proinflammatory cytokines were significantly reduced (Fig. EV1D–F). In some cell types, viral replication was nearly abolished at 100 μM BT, approaching background levels. Specifically in RAW264.7 and HT1080, BT lowered infectious titers by plaque assay (Fig. EV1G). BT alone did not increase *Il1b*, *Il6*, *Tnfα* in cells or in vivo (Fig. EV1H,I). Entry assays were unaffected by BT, whereas time-of-addition tests showed reduced post-entry viral RNA, indicating a host-mediated post-entry restriction (Fig. EV1J–L). Importantly, this antiviral effect was not accompanied by significant cytotoxicity (Fig. EV1M).

Encouraged by these robust in vitro results, we extended our investigation to evaluate the protective effects of BT in vivo using six-week-old C57BL/6 J mice. Animals were infected with MHV, VSV, HSV, IAV or EMCV via intraperitoneal injection, and subsequently treated with BT. Survival analyses indicated significantly improved survival rates in BT-treated groups compared with untreated controls. In addition, RT-qPCR analysis of blood and relevant tissues revealed markedly reduced viral loads across all infections (HSV, MHV, VSV, IAV, and EMCV) after BT treatment (Fig. 1G–K), confirming its potent in vivo antiviral activity. For VSV and HSV, plaque assays of liver homogenates confirmed significant decreases in infectious titers following BT treatment (Fig. EV1N). Remarkably, daily administration of BT for 14 days caused no observable toxicity, as assessed by organ appearance, organ weights, body wight, and histological analysis of major tissues (Fig. EV1O–Q).

Consistent with reduced viral loads, inflammatory cytokines such as *Il-1β*, *Il-6*, and *Tnf-α* were markedly downregulated in BT-treated groups measured by RT-qPCR, further illustrating the potent anti-inflammatory effect accompanying antiviral activity (Figs. 1G–K and EV2A–H). Histological analyses provided additional confirmation of BT-mediated protection, with notably less severe pathological damage observed in liver tissues from mice infected with MHV, VSV, and HSV-1, EMCV, as well as in lung tissues from mice infected with IAV (Fig. 1L).

Collectively, our results clearly demonstrated that BT induced robust antiviral and anti-inflammatory responses, effectively reducing viral replication and virus-induced tissue damage both in vitro and in vivo.

## Betrixaban induces type I-IFN production in the absence of infection

To characterize the transcriptional response to BT, we performed bulk RNA-seq to detect gene expression in BT-stimulated HT1080 cells. The expression of *IFNB1*, as well as a cluster of IFN-stimulated genes (ISGs), including *ISG15*, *OAS2*, *RSAD2*, and *IFIT3* (Fig. 2A), was markedly induced, whereas proinflammatory genes such as *IL1B* were downregulated (Fig. 2B). Then, we performed Gene Set Enrichment Analysis (GSEA) and Over-Representation Analysis (ORA) on the bulk RNA-seq data. Both KEGG and GO analyses identified significant enrichment of multiple innate immunity and antiviral signaling pathways (Fig. 2C).

Next, we used multiple cell lines to confirm these findings, including RAW 264.7, HT1080, HT29, and HeLa. RT-qPCR results demonstrated a marked induction of *IFNB1* mRNA relative to DMSO control. Western blot analysis revealed dose-dependent phosphorylation of TBK1 (p-TBK1) and IRF3 (p-IRF3), as well as the upregulation of Viperin, IFIT3, and OAS2 (Fig. 2D). Across cell types, BT weakly induce apoptosis within our working window. Cleaved caspase-3, cleaved-PARP, cleaved caspase-9 and cleaved caspase-8 were changed a little, and totals were stable (Fig. EV2I).

In human peripheral blood mononuclear cells (PBMCs) and mouse bone marrow-derived macrophages (BMDMs), BT treatment resulted in an obvious increase in *IFNB1* mRNA and *OAS2* mRNA levels (Fig. 2E,F). ELISA confirmed higher secreted IFN-β under the same conditions (Fig. EV2J). C57BL/6 J mice given a single intraperitoneal dose of 50 mg/kg BT showed a significant

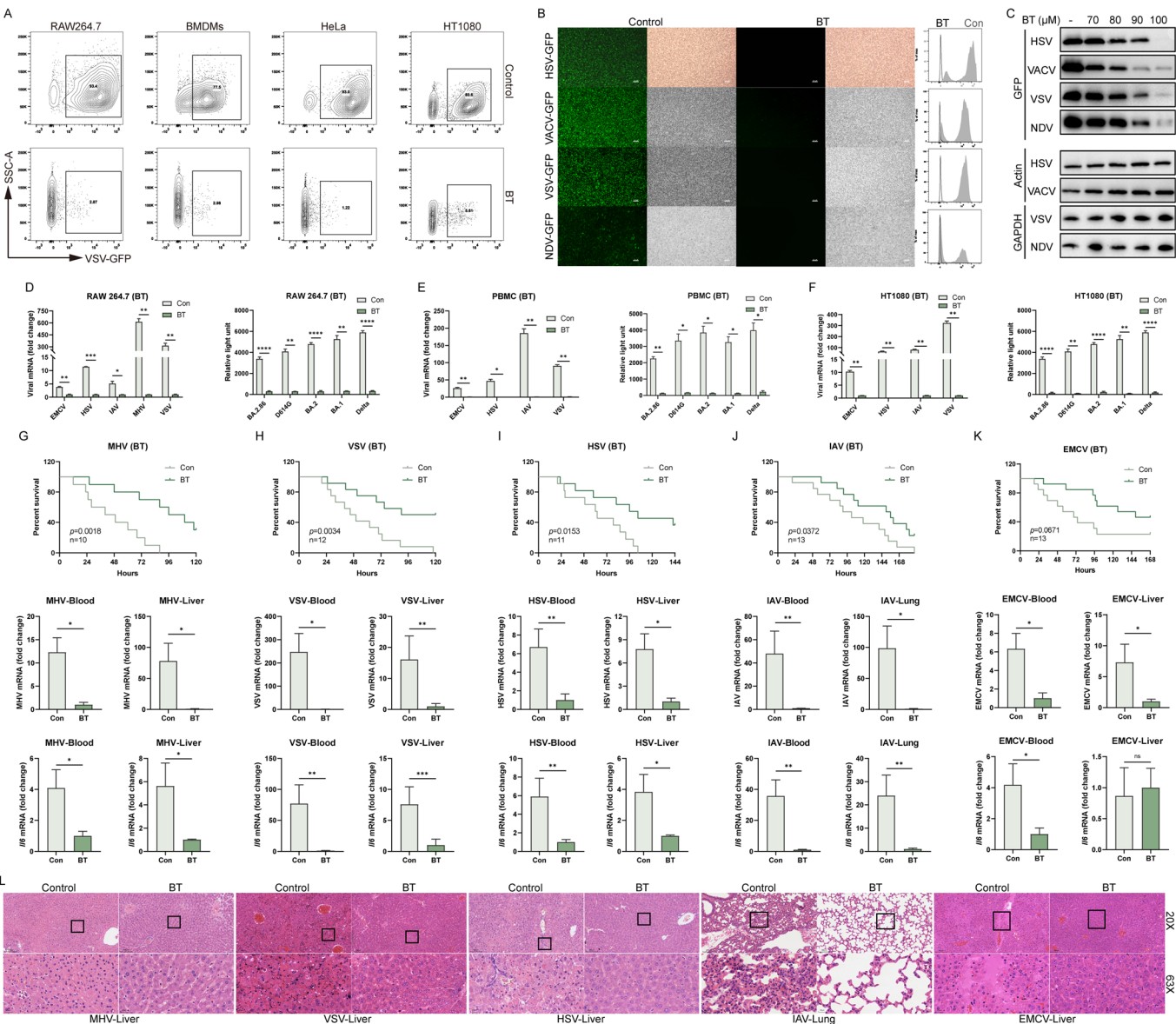

**Figure 1. Betrixaban (BT) establishes a host-protective antiviral state in vitro and in vivo.**

(A) Flow cytometric analysis of VSV-GFP infection in RAW 264.7 cells, bone marrow-derived macrophages (BMDMs), HeLa, and HT1080 cells treated with DMSO (Con) or 100 μM BT and infected with VSV-GFP (MOI = 0.1) for 12 h. Numbers indicated the percentage of GFP-positive cells. The data were representative of three independent experiments. (B) Representative fluorescence (green) and bright-field images of RAW 264.7 cells treated with DMSO (Con) or 100 μM BT and infected with HSV-GFP, VACV-GFP, VSV-GFP or NDV-GFP (MOI = 0.1). Adjacent histograms show flow cytometric quantification of GFP fluorescence (n = 3). (C) Dose-dependent inhibition of viral protein expression by BT. RAW 264.7 cells were treated with the indicated concentrations of BT and infected with HSV-1, VACV, VSV, or NDV (MOI = 0.1) for 12 h, and analyzed by western blot for GFP and actin. (D–F) RT-qPCR quantification of viral RNA (left) and luciferase to detect BA.2.86, D614G, BA.2, BA.1, Delta in RAW 264.7 (D), human PBMCs (E), and HT1080 (F) cells (n = 3; three biological repeats). (G–K) BT protected C57BL/6J mice from lethal viral challenge. Kaplan–Meier survival curves of mice infected intraperitoneally with MHV (G; n = 10), VSV (H; n = 12), HSV-1 (I; n = 11), IAV (J; n = 13), or EMCV (K; n = 13) and treated daily with vehicle or BT (50 mg/kg). Below each survival plot, RT-qPCR quantification of viral RNA and Il-6 mRNA in blood and target organs harvested 24 h post-infection (n = 6; six biological repeats). (L) Histopathological analysis of tissues from MHV-, VSV-, EMCV-, HSV-1-infected livers, and IAV-infected lungs of mice treated as in (G–K). Upper panels, low-magnification (×20); lower panels, high-magnification (×63) views of boxed regions. Scale bars, 100 μm (×20) and 20 μm (×63). Data are shown as mean ± SEM. N.S., not significant, P > 0.05; *P < 0.05; **P < 0.01; ***P < 0.001; ****P < 0.0001. Western blot quantifications are presented in Appendix Fig. S2 and exact P values are in Appendix Tabe S1. Source data are available online for this figure.

systemic upregulation of *Ifnb1* and *Oas2* in heart, liver, lung, spleen, and kidney at 6 h post-injection (Fig. 2G). Consistently, serum and targeted organs IFN-β were elevated in BT-treated mice measured by ELISA (Fig. EV2K).

In RAW 264.7 cells, BT induced a dose-dependent increase in *Ifnb1* mRNA, whereas TBK1 knockout cells showed no induction at any concentration. Consistently, BT failed to upregulate Viperin, IFIT3, or OAS2 proteins in TBK1 knockout cells (Figs. 2H and

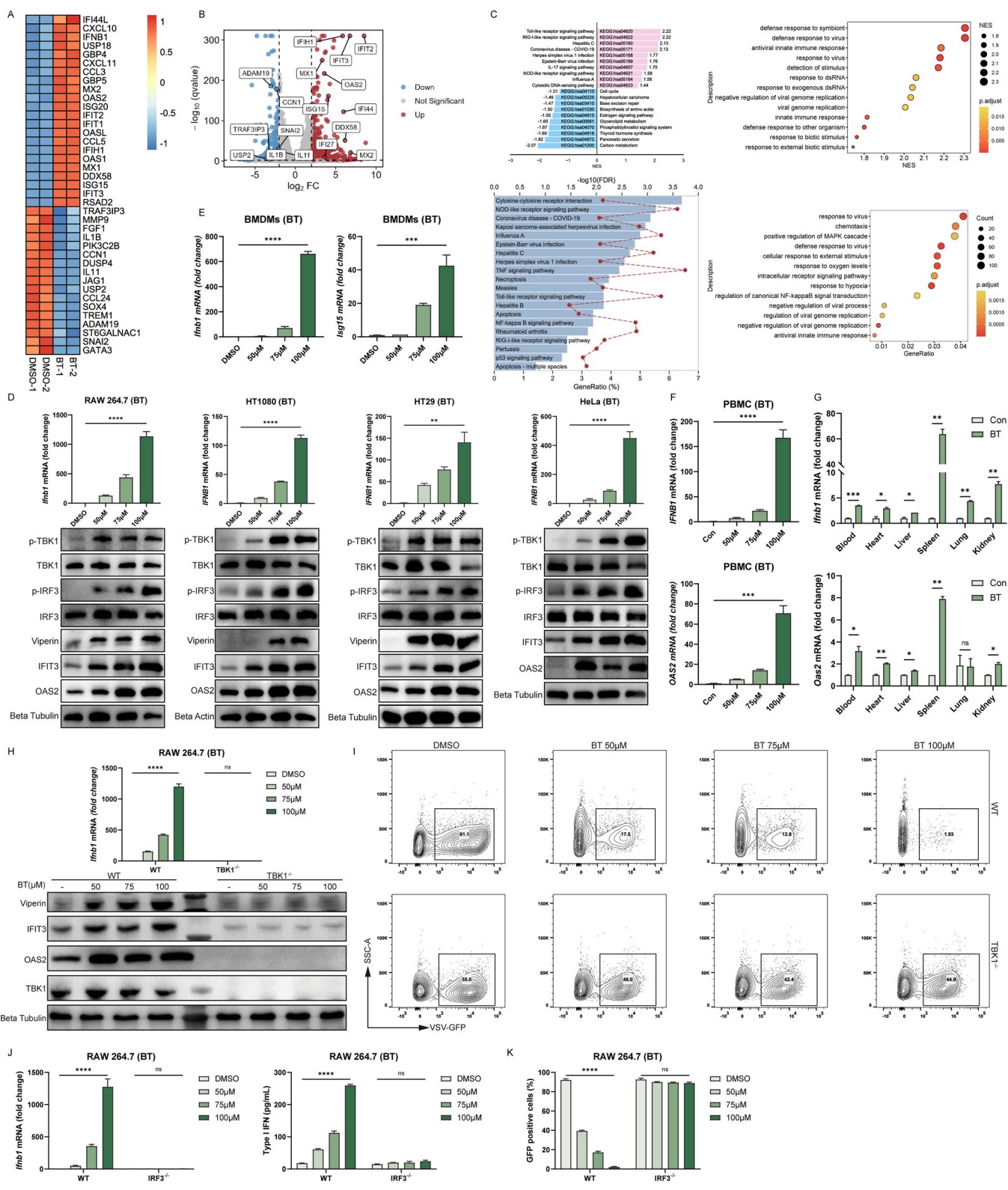

◀ **Figure 2. Betrixaban induces type I-IFN production in the absence of infection.**

(A) Heatmap of bulk RNA-seq in HT1080 cells treated with 100 µM BT versus DMSO for 12 h. (B) Volcano plot of differential expression in BT- versus DMSO-treated HT1080 cells (|log₂FC | >1, FDR < 0.05) (n = 2; two biological repeats). (C) Gene set enrichment analysis (left) and over-representation analysis (right) of BT-regulated genes, highlighting significant enrichment of antiviral and innate immunity pathways (GO and KEGG). (D) Dose-dependent induction of *IFNB1* mRNA (RT-qPR, top panels) and corresponding protein responses (western blots, bottom panels) in RAW 264.7, HT1080, HT29, and HeLa cells treated with the indicated BT concentrations for 12 h (n = 3; three biological repeats). (E) RT-qPCR quantification of *Ifnb1* and *Oas2* mRNA in mouse bone marrow-derived macrophages (BMDMs) treated with indicated concentrations of BT for 12 h (n = 3; three biological repeats). (F) RT-qPCR of *IFNB1* and *OAS2* in human PBMCs (n = 4; four biological repeats). (G) In vivo induction of *Ifnb1* and *Oas2* mRNA in blood, heart, liver, lung, spleen, and kidney of C57BL/6J mice 6 h after a single intraperitoneal injection of BT (50 mg/kg) (n = 3; three biological repeats). (H) RT-qPCR of *Ifnb1* in RAW 264.7 cells following BT treatment (50, 75, 100 µM) for 12 h in wild-type and knockout cells. Western blots showed the absence of Viperin, IFIT3, and OAS2 induction in TBK1 knockout cells (n = 3; three biological repeats). (I) Flow cytometry of VSV-GFP infection in WT and TBK1 knockout RAW 264.7 cells treated with BT (50, 75, 100 µM) and infected (MOI = 0.1) for 12 h. (J) RT-qPCR of *Ifnb1* in RAW 264.7 cells following BT treatment (50, 75, 100 µM) for 12 h in wild-type and knockout cells (left), secreted type I IFN measured by ELISA (right) (n = 3; three biological repeats). (K) Wild-type and knockout cells treated with DMSO (Con) or BT (50, 75, 100 µM) and infected with HSV-GFP or VSV-GFP (MOI = 0.1), measured by flow cytometry. Numbers indicated percentage of GFP-positive cells (n = 3; three biological repeats). Data are shown as mean ± SEM. N.S. not significant, P > 0.05; *P < 0.05; **P < 0.01; ***P < 0.001; ****P < 0.0001. Western blot quantifications are presented in Appendix Fig. S2 and exact P values are in Appendix Tabe S1. Source data are available online for this figure.

EV6A). Then, we treated WT and TBK1 knockout cells with BT (50 µM, 75 µM, 100 µM) for 12 h, simultaneously infected with VSV-GFP (MOI = 0.1). Flow cytometry revealed a concentration-dependent reduction of GFP-positive cells in WT cells, whereas TBK1 knockout cells remained highly permissive at all BT doses (Fig. 2I; Appendix Fig. S1B). These results established that TBK1 was essential for the antiviral action of BT. Furthermore, IRF3 knockout reduced BT-induced IFNB1 and antiviral protection in VSV-GFP and HSV-GFP, establishing IRF3 as a critical downstream effector (Fig. 2J,K). Consistently, knockout of RNASEL did not alter BT-induced *IFNB1* mRNA, while antiviral protection was only modestly reduced by 10–20% in VSV-GFP and HSV-GFP infection assays measured by flow cytometry, indicating that OAS-RNase L functioned as an auxiliary pathway under our conditions (Fig. EV2L,M).

## Betrixaban directly binds and sensitizes cGAS activation

Next, we determined what underlies BT-induced innate immune activation. To identify potential innate immune targets of BT, we applied the DeepAVC prediction model to screen for binding affinity across pattern recognition receptors (PRRs) (Kang et al, 2025). We found that BT was most likely to bind the DNA sensor cGAS, with the highest predicted affinity score of 0.87772816 (Fig. 3A). This computational prediction was validated by surface plasmon resonance (SPR), which demonstrated a direct binding of BT to recombinant human cGAS with a dissociation constant of ~79.78 µM, salmon sperm double-stranded DNA served as a positive control (Figs. 3B,C and EV3A).

To validate the cGAS–STING pathway dependency, we used CRISPR/Cas9 to generate cGAS⁻/⁻ and STING⁻/⁻ HT1080 cell lines (Fig. EV6A). In these knockout cell lines, BT's induction of *IFNB1* mRNA (by RT-qPCR) (Fig. 3D) and secreted IFNB (by ELISA) (Fig. 3E) was dramatically reduced compared to wild-type controls, and likewise in cGAS knockout BMDMs (Fig. EV3B). Consistently, Western blot analysis confirmed that phosphorylation of STING and TBK1 was strongly declined in the absence of cGAS or STING (Fig. 3F). Next, we infected wild-type and knockout cells with VSV-GFP (MOI = 0.1), treated them with BT, and measured infection by flow cytometry (Fig. EV3C). Knockout cells were clearly less protected, but they still showed partial antiviral activity (Fig. 3G).

Furthermore, in order to figure out whether the cGAS-STING pathway was necessary for BT's protective effect against viral

infection in vivo, we treated wild-type and cGAS knockout mice with BT and challenged with lethal doses of DNA viruses HSV-1 or RNA viruses VSV. Unlike in cells, cGAS knockout mice exhibited only a slight delay in mortality and showed no significant survival benefit upon BT treatment (Fig. 3H,I). We quantified blood and liver viral loads by RT-qPCR. In cGAS⁻/⁻ mice, BT caused only minor, non-significant reductions in viral load (Figs. 3H,I and EV3D,E). These in vivo results underscore that the antiviral effect of BT was critically dependent on the integrity of the cGAS-STING.

To understand how BT activated cGAS, we investigated cellular changes after treatment through confocal microscopy. The dsDNA immunostaining under digitonin-only permeabilization revealed numerous extra-nuclear dsDNA-positive puncta in BT-treated cells (Fig. 3J), which were largely absent in untreated controls. Moreover, the MitoTracker Deep Red FM (Invitrogen M22426) imaging showed a change from a tubular mitochondrial network to disorganized and fragmented structures with increased per-cell labeling (Fig. 3K), indicating mitochondrial network remodeling. To further validate these structural changes at higher resolution, we performed STED super-resolution microscopy (Fig. 3L) and transmission electron microscopy (TEM) (Fig. 3M). Both methods confirmed the presence of highly fragmented mitochondria and loss of cristae integrity in BT-treated cells, consistent with mitochondrial damage and the leakage of mitochondrial DNA into the cytosol. To functionally test the contribution of dsDNA, when mitochondrial DNA availability was curtailed, BT-induced *IFNB1* mRNA expression and antiviral protection were reduced but not abolished (Fig. EV3F,G). These findings suggested that BT induced mitochondrial stress, leading to the release of endogenous dsDNA, a likely upstream trigger for cGAS activation.

We further used in vitro reconstitution assays to test the direct effect of BT on cGAS enzymatic activity. Purified recombinant human cGAS was incubated with its DNA substrate in the presence or absence of BT, and the production of cGAMP was quantified by LC-MS. The addition of BT alongside dsDNA significantly enhanced cGAMP production by cGAS compared to dsDNA alone. Notably, BT alone, even without any added DNA, stimulated cGAS to produce a low but detectable amount of cGAMP, whereas cGAS normally remains inactive without a DNA trigger (Fig. 3N). For comparison, Mn²⁺ a known weak cGAS activator, produced more cGAMP than BT alone, whereas in the presence of dsDNA the potentiation achieved by BT was comparable to that of Mn²⁺ (Fig. EV3H). These results indicated that BT potentiated cGAS

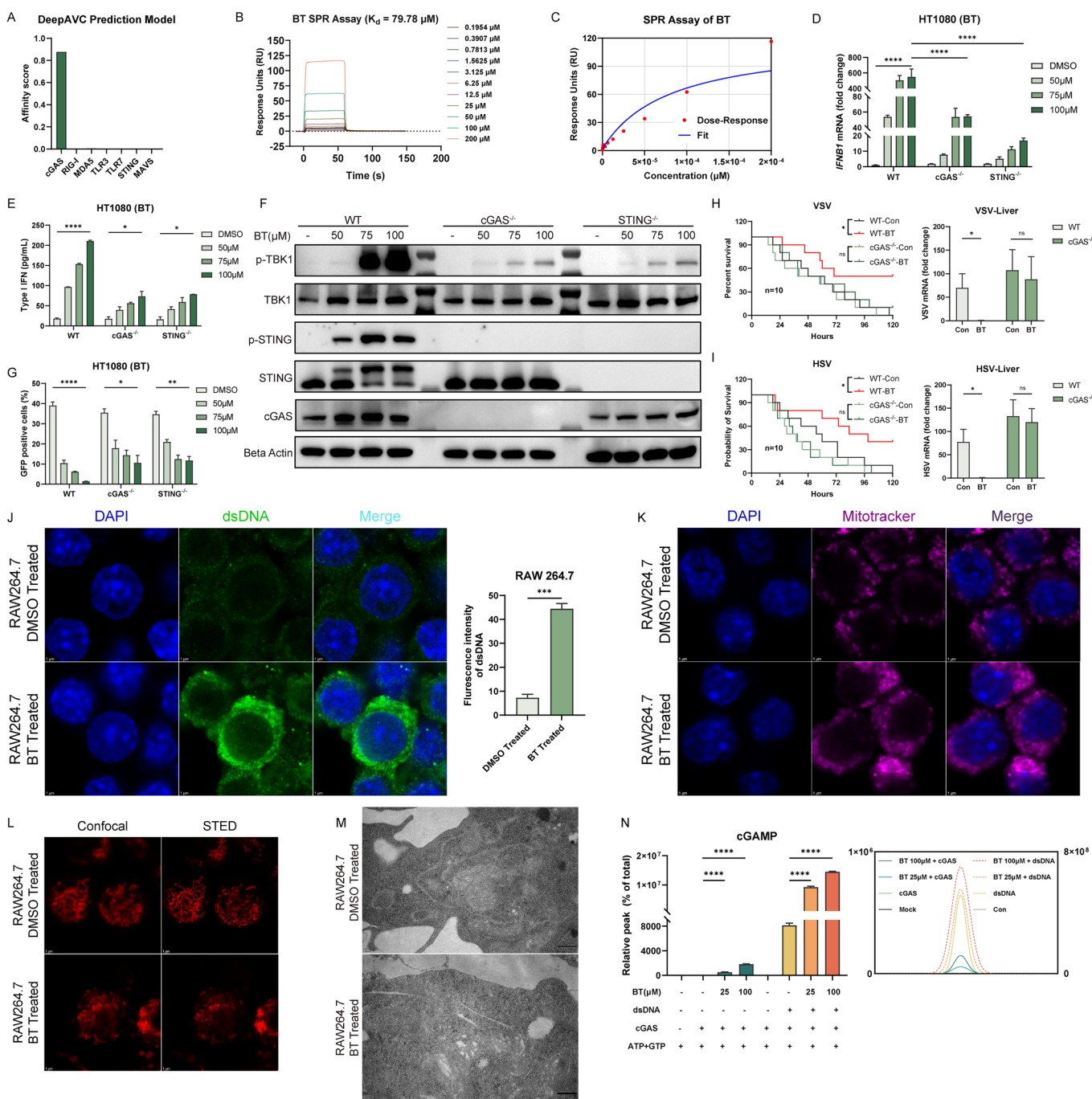

activity in generating cGAMP, and even partially activated cGAS in the absence of DNA.

## Betrixaban remodels chromatin and reactivates ERVs

Bulk RNA-seq analysis revealed downregulation of multiple HDAC family transcripts upon treatment. Therefore, we performed RT-qPCR in HT1080 cells to assess HDAC expression. RT-qPCR analysis was consistent with bulk RNA-seq findings (Fig. 4A). Moreover, a fluorometric assay of HDAC enzymatic activity showed an obvious reduction in treated cells compared to controls

(Fig. 4B), indicating effective HDAC inhibition. To evaluated global histone acetylation levels, we conducted western blots to detect a marked increase in H3K27ac (histone H3 lysine 27 acetylation) in drug-treated cells relative to untreated cells (Fig. 4C). These observations aligned with the expected outcome of HDAC blockade, as HDAC inhibitors were known to cause widespread hyperacetylation of histones (Lane and Chabner, 2009; Shimazu et al, 2013).

To verify the condition of acetylation of histones, we conducted CUT&Tag profiling of H3K27ac. The results revealed widespread gains in this active histone mark across the genome in BT-treated

◀    **Figure 3.  Betrixaban directly binds and sensitizes cGAS activation.**

(A) Predicted binding affinity scores of BT against pattern recognition receptors using the DeepAVC model. (B) SPR sensorgrams of BT binding to recombinant human cGAS at the indicated concentrations (0.1954–200 μM). (C) Dose–response curve fitted from SPR data in (B). (D) RT-qPCR of *IFNB1* mRNA in wild-type (WT), cGAS knockout, and STING knockout HT1080 cells treated with BT (50, 75, or 100 μM) or DMSO for 12 h ($n = 3$; three biological repeats). (E) Secreted type I IFN measured by ELISA in the same cell lines and treatments as in (D) ($n = 3$; three biological repeats). (F) Western blots of WT, cGAS knockout, and STING knockout HT1080 cells treated as in (D). (G) Flow cytometry of VSV-GFP infection in WT, cGAS knockout and STING knockout HT1080 cells treated with BT (50, 75, 100 μM) and infected (MOI = 0.1) for 12 h ($n = 3$; three biological repeats). (H, I) In vivo VSV and HSV-1 challenges in wild-type and cGAS knockout mice. Left panel is the survival curves of mice ($n = 10$; ten biological repeats), right panel is RNA levels of viruses in the liver measured by RT-qPCR ($n = 5$; five biological repeats). (J) Confocal micrographs and quantification of the immunofluorescence of RAW 264.7 cells treated with DMSO or 50 μM BT for 12 h, stained for DAPI (blue) and cytosolic dsDNA (green) ($n = 3$; three biological repeats). Scale bars, 5 μm. (K) Confocal images of RAW 264.7 cells treated as in (J), stained with DAPI (blue) and MitoTracker to visualize mitochondrial morphology. Scale bars, 5 μm. (L) Comparison of mitochondrial structure by confocal versus STED super-resolution microscopy in RAW 264.7 cells treated with DMSO or BT and stained with PK Mito. Scale bars, 2 μm. (M) Transmission electron micrographs of mitochondria in RAW 264.7 cells treated with DMSO or BT. (N) In vitro cGAS enzymatic assays: LC-MS quantification of cGAMP production by recombinant cGAS incubated with dsDNA in the presence of BT (25 or 100 μM), and cGAS incubated with BT (25 or 100 μM) in the absence of exogenous DNA ($n = 3$; three biological repeats). Data are shown as mean ± SEM. N.S. not significant, $P > 0.05$; *$P < 0.05$; **$P < 0.01$; ****$P < 0.0001$. Western blot quantifications are presented in Appendix Fig. S2 and exact $P$ values are in Appendix Tabe S1. Source data are available online for this figure.

cells (Fig. 4D), particularly at promoters (Fig. EV4A–C), indicating that many regulatory regions transitioned to an epigenetically active state. In parallel, ATAC-seq analysis demonstrated a global increase in chromatin accessibility upon treatment. The intensity of ATAC-seq peaks were higher in BT-treated cells (Figs. 4E and EV4D–F), often co-localizing with regions of increased H3K27ac (Fig. 4F). For example, the promoters of canonical interferon-stimulated genes (ISGs) such as MX1, IFIT1, and IFNB1 showed minimal H3K27ac and accessibility in the control group but displayed strong H3K27ac enrichment and open chromatin peaks after treatment (IGV snapshots in Figs. 4G and EV4G). This corresponded with pronounced upregulation of these genes in the BT-treated group in bulk RNA-seq (Fig. 4F), and exemplified how the drug induced chromatin relaxation at specific loci involved in immune response. Overall, our results indicated that the BT treatment resulted in robust HDAC inhibition and consequent chromatin decondensation.

Given the extensive chromatin changes, we next examined TRIM28 (Bacon et al, 2020; Goodarzi et al, 2011). CUT&Tag profiling for TRIM28 demonstrated a genome-wide loss of TRIM28 binding in treated cells (Figs. 4H and EV4H–J). The intensity of TRIM28 enrichment at its target sites was greatly reduced compared to control cells, indicating that TRIM28 was lost from many chromatin regions by the drug. Notably, regions that lost TRIM28 occupancy corresponded frequently to those that gained H3K27ac, suggesting an inverse relationship between TRIM28 binding and acetylation levels (Figs. 4I and EV4K).

To determine the sequence of events, we performed a time-course analysis around the onset of chromatin opening. We found that TRIM28 dissociation preceded the accumulation of H3K27ac. In a representative TRIM28 target locus, CUT&Tag-qPCR revealed that TRIM28 enrichment dropped significantly by 4 h after treatment, whereas H3K27ac at the same site did not show a significant increase until 8-12 h (Fig. 4J). This timing indicated that TRIM28 removal was an early event that likely facilitated subsequent histone acetylation. Together, our data indicated that the drug removed TRIM28-mediated repression, thereby enabling H3K27ac deposition and increased chromatin accessibility.

An important consequence of the chromatin de-repression was the reactivation of transposable elements, particularly endogenous retroviruses (ERVs) (Asimi et al, 2022). Consistently, bulk RNA-seq analysis revealed robust upregulation of numerous ERV transcripts in BT-treated cells (Fig. 4K), indicating that the drug

treatment unleashed a broad set of previously silenced retro-elements. We next validated whether the activated ERVs showed concordant chromatin changes. A metaprofile of H3K27ac, Trim28, and accessibility signals centered on upregulated ERVs showed that BT-treatment induced strong H3K27ac enrichment, increased ATAC-seq accessibility and near-complete loss of TRIM28 binding (Figs. 4L–N and EV4L–O). This indicated that these elements transitioned to an active chromatin state as they were transcriptionally activated. A representative example was provided by the HERVK14-int locus. At this locus, genome browser tracks show that H3K27ac enrichment was greatly elevated and TRIM28 binding was lost after treatment, and a new ATAC-seq peak emerged overlapping the ERV sequence (Fig. 4O). Interestingly, we also identified further candidate ERVs that may function in antiviral immunity (Fig. 4P). Collectively, these transcriptomic and epigenomic data supported that BT, via potent HDAC inactivation and widespread histone acetylation, disrupted TRIM28-mediated epigenetic silencing of ERVs.

## Reactivated ERVs mediate the cytosolic dsRNA-sensing pathway

Annotation of upregulated TEs showed that long terminal repeat (LTR) retrotransposons dominated the reactivated elements. Over 400 distinct ERV/LTR loci were transcriptionally derepressed by BT, which far exceeded the induction of non-LTR elements, including LINEs, SINEs, or DNA transposons (Fig. 5A). This result indicated that widespread ERV reactivation was a source of immunostimulatory dsRNA. Supporting this, the GO annotation of these ERV loci identified DDX58/IFIH1-mediated induction of interferon alpha and beta as the top-enriched association (Fig. 5B). The Reactome annotation of these ERV loci revealed functions like regulation of type I interferon-mediated signaling pathway (Fig. EV5A). Furthermore, GSEA of bulk RNA-seq confirmed that BT-treated cells were highly enriched for gene signatures of RIG-I-like receptor signaling and cellular responses to double-stranded RNA (Fig. 5C), suggesting engagement of cytosolic RNA-sensing pathways. We also noticed that IFNB1 induction was not completely abolished in cGAS or STING knockout cells, suggesting that additional nucleic acid sensors may partially compensate in the absence of cGAS-STING. Together, these results strongly indicated the activation of the RIG-I/MDA5-MAVS antiviral RNA-sensing pathway in BT-treated cells.

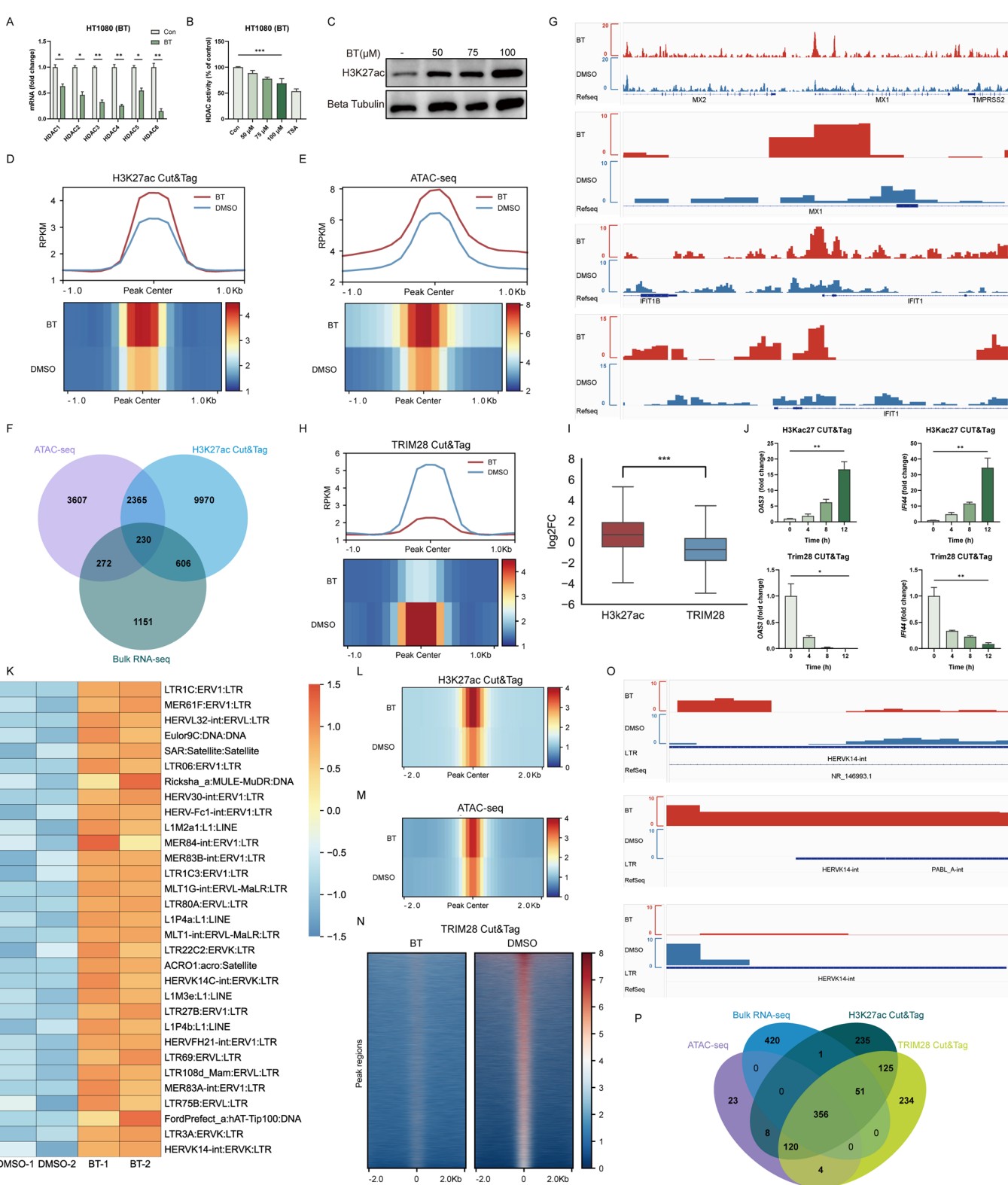

◀

**Figure 4.   Betrixaban remodels chromatin and reactivates ERVs.**

(A) RT-qPCR of *HDAC* mRNA in HT1080 cells treated with 100 μM BT or DMSO for 12 h (*n* = 3; three biological repeats). (B) Fluorometric HDAC activity assay in HT1080 cells after 12 h treatment with BT (50, 75, 100 μM) or DMSO. Enzymatic activity is shown as percentage of DMSO control (*n* = 3; three biological repeats). (C) Western blot of global H3K27ac levels in HT1080 cells. (D) CUT&Tag metaprofile (top) and heatmap (bottom) of H3K27ac signal ±1 kb around called peaks. (E) ATAC-seq metaprofile (top) and heatmap (bottom) of chromatin accessibility ±1 kb around peak centers. (F) Venn diagram of overlapping regions showing increased H3K27ac, increased accessibility, and increased gene expression in BT-treated cells. (G) Genome browser (IGV) snapshots at MX1 and IFIT1 loci showing H3K27ac CUT&Tag and ATAC-seq. (H) CUT&Tag metaprofile (top) and heatmap (bottom) of TRIM28 occupancy ±1 kb around called peaks, illustrating widespread loss of TRIM28 binding. (I) Boxplots in CUT&Tag signal for H3K27ac (left) and TRIM28 (right) at consensus peak regions (*n* = 1; one biological repeat). (J) Time-course CUT&Tag-qPCR at an OAS2 and IFNB1 regulatory locus, measured at 0, 4, 8, and 12 h after BT (100 μM) treatment (*n* = 3; three biological repeats). (K) Heatmap of the top 40 upregulated ERV families in bulk RNA-seq. (L–N) Meta-profiles of chromatin changes at the set of BT-reactivated ERV loci (±2 kb from center). (L) H3K27ac CUT&Tag; (M) TRIM28 CUT&Tag; (N) ATAC-seq heatmaps. (O) Representative IGV tracks at the HERVK14-int locus showing BT-induced gain of H3K27ac (top), loss of TRIM28 binding (middle), and emergence of an ATAC-seq peak (bottom). (P) Four-way Venn diagram of ERV loci identified by bulk RNA-seq upregulation, H3K27ac CUT&Tag gain, TRIM28 CUT&Tag loss, and increased ATAC-seq accessibility. Data are shown as mean ± SEM. N.S. not significant, *P* > 0.05; *\**P* < 0.05; \*\**P* < 0.01; \*\*\**P* < 0.001. Western blot quantifications are presented in Appendix Fig. S2 and exact *P* values are in Appendix Tabe S1. Source data are available online for this figure.

Initially, we used dsRNA-specific antibody to conduct confocal microscopy in order to confirm the existence of ERVs (Fig. 5D). Immunofluorescence imaging of BT-treated cells revealed a marked accumulation of dsRNA, whereas the DMSO-treated group showed only basal levels of diffuse dsRNA staining. This result was consistent with the transcriptional derepression of ERVs and the generation of immunogenic viral RNA mimics that can activate RIG-I and MDA5.

To directly test the role of RIG-I/MDA5-MAVS signaling in BT-induced interferon responses, we utilized CRISPR/Cas9 to generate knockout cell lines for RIG-I, MDA5, or MAVS (Fig. EV6A). We found that BT-induced interferon responses were severely blunted in all three knockout cell lines through RT-qPCR and ELISA (Fig. 5E,F), and likewise in MAVS knockout BMDMs and under RIG-I inhibition (Figs. EV3B and EV5B). Consistently, Western blot analysis demonstrated that BT triggered strong phosphorylation of TBK1 in wild-type cells, whereas phospho-TBK1 was greatly diminished in knockout cells (Fig. 5G). These results indicated that BT's activation of downstream interferon signaling was dependent on an intact RIG-I/MDA5-MAVS pathway.

Correspondingly, BT treatment was markedly less effective at protecting these knockout cell lines from VSV-GFP infection (Figs. 5H and EV5C). This dependence was recapitulated in vivo using MAVS knockout mice derived no significant survival benefit from BT treatment following lethal VSV and HSV-1 challenge (Fig. 5I,J). We quantified blood and liver viral loads by RT-qPCR. In MAVS$^{-/-}$ mice, BT caused only minor, non-significant reductions in viral load. While BT-treated wild-type mice exhibited significantly improved survival and reduced viral loads, underscoring that MAVS is essential for BT's protective effect in vivo (Figs. 5I,J and EV5D).

Moreover, our previous multi-omics analysis identified HERVK14-int, which has been reported to produce immunogenic dsRNA sensed by RIG-I and MDA5(Berkhout et al, 1999; Roulois et al, 2015). Consistent with this, RIP-qPCR showed that BT treatment increased HERVK14-int RNA association with RIG-I and MDA5, respectively (Fig. 5K), confirming that ERV-derived dsRNA was directly bound by these cytosolic RNA sensors.

Given our earlier results demonstrating cGAS-STING activation by BT, we generated MAVS/STING double-knockout cells. Strikingly, double knockout cells completely lost BT-induced interferon production (Fig. 5L) and antiviral protection against VSV-GFP infection (Fig. 5M). These findings strongly suggested

that BT concurrently activated both RNA-sensing (RIG-I/MDA5-MAVS) and DNA-sensing (cGAS-STING) pathways, and that dual-pathway activation was critical for the full antiviral response induced by BT.

## Discussion

In this study, we demonstrated that Betrixaban (BT) as the first small-molecule directly bound cGAS and sensitized DNA-mediated activation to induce a broad antiviral state by linking epigenetic reprogramming to innate nucleic acid sensing. BT treatment caused an accumulation of double-stranded DNA (dsDNA), likely due to BT-induced mitochondrial stress and leakage of mitochondrial DNA (mtDNA), then the DNA sensor cGAS was activated. Strikingly, BT also directly bound and activated cGAS to boost cGAMP production, lowering its DNA activation threshold.

Simultaneously, BT acted as a potent histone deacetylase inhibitor (HDACi), leading to histone hyperacetylation, chromatin relaxation, and displacement of the transcriptional co-repressor TRIM28 from chromatin. This effectively lifted TRIM28-mediated silencing of endogenous retroviruses (ERVs), allowing their transcription. The reactivated ERV transcripts formed cytosolic double-stranded RNA (dsRNA) that activated the RIG-I/MDA5-MAVS RNA-sensing pathway, triggering robust type I interferon (IFN) production and interferon-stimulated gene (ISG) expression.

Meanwhile, our work provided proof-of-concept that an oral small molecule can systemically boost host antiviral defenses by harnessing innate nucleic acid sensors rather than targeting pathogens directly. We identified BT as a dual-pathway innate immune activator that may help circumvent viral evasion strategies, as many viruses target a single sensing mechanism. As an FDA-approved orally available drug, BT has clear advantages in delivery and dosing precision compared to large DNA or unstable cyclic dinucleotide agonists. Repurposing an existing drug like BT provides a translationally attractive path to broad-spectrum antivirals and immunotherapies.

Our findings extended and refined the viral mimicry model of innate immunity. Normally, mammalian cells safeguard genomic integrity and avoid spurious antiviral responses by tightly silencing transposable elements via DNA methylation and repressive histone modifications(Dahlet et al, 2020; Deniz et al, 2019; Kidder et al, 2017; Wu et al, 2024). TRIM28 recruits HDACs and H3K9

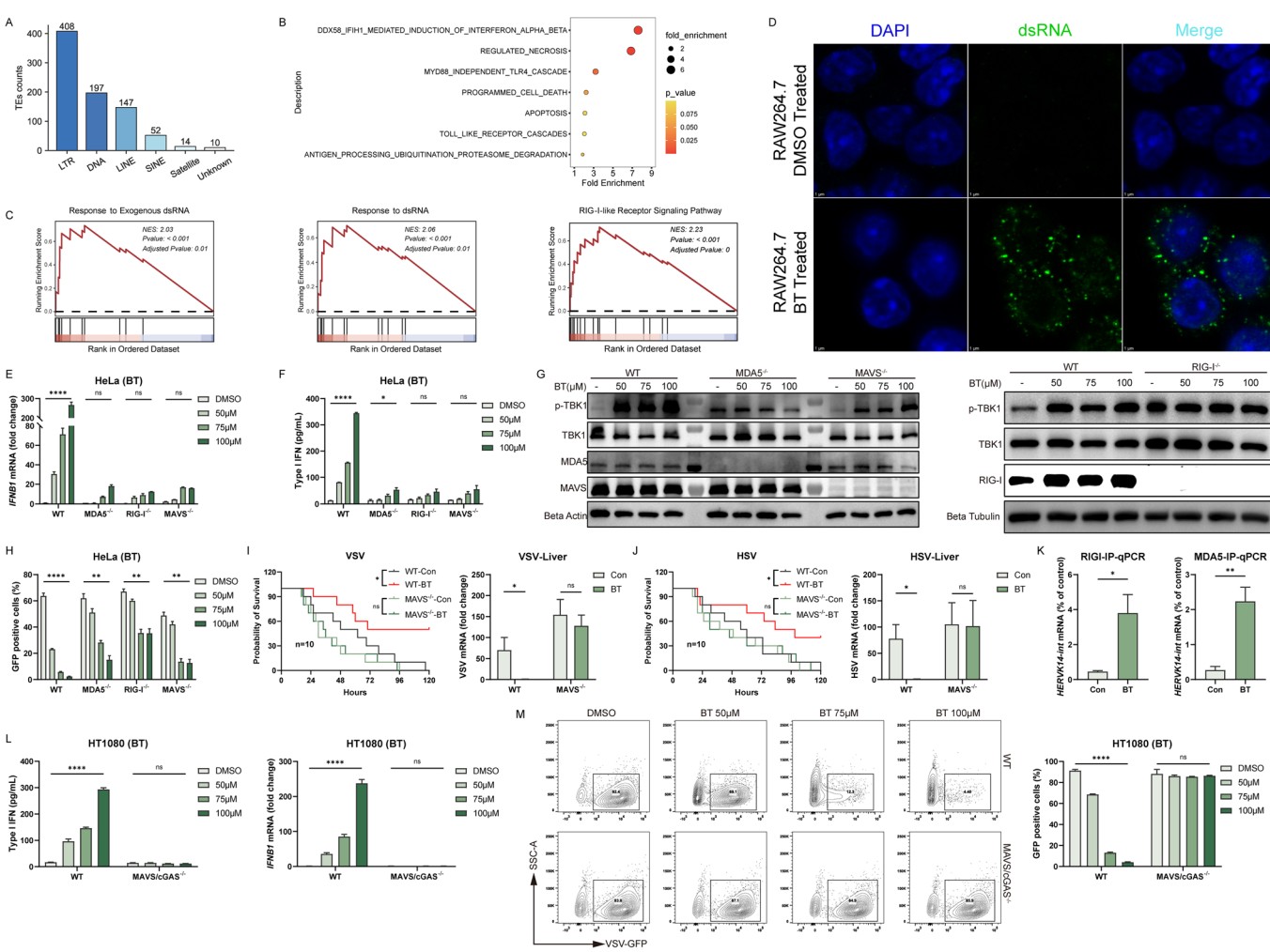

**Figure 5. Reactivated ERVs mediate the cytosolic dsRNA-sensing pathway.**

(A) Distribution of transposable element (TE) classes among loci upregulated by BT in bulk RNA-seq (counts of LTR, DNA, LINE, SINE, Satellite, and Unknown elements). (B) Reactome terms enriched among BT-reactivated ERV loci. (C) GSEA enrichment plots in BT- versus DMSO-treated HT1080 cells. (D) Confocal micrographs of RAW 264.7 cells treated with DMSO or 50 μM BT for 12 h, stained for dsRNA (green) and nuclei (DAPI, blue). BT treatment induces punctate cytosolic dsRNA accumulations. Scale bars, 5 μm. (E) RT-qPCR of *IFNB1* mRNA in HeLa wild-type (WT), MDA5 knockout, RIG-I knockout, and MAVS knockout cells treated with BT (*n* = 3; three biological repeats). (F) Secreted type I IFN measured by ELISA in the same HeLa cell lines and treatments as in (E) (*n* = 3; three biological repeats). (G) Western blot of p-TBK1, total TBK1, MDA5, MAVS, and RIG-I, the samples were the same as (E). (H) Flow cytometric quantification of VSV-GFP infection in HeLa wild-type (WT), MDA5 knockout, RIG-I knockout, and MAVS knockout cells treated with BT (*n* = 3; three biological repeats). (I, J) Left are survival curves of wild-type and MAVS knockout C57BL/6 J mice challenged with lethal VSV and HSV-1 after a single i.p. dose of BT (50 mg/kg) (*n* = 10; ten biological repeats). Right are viruses RNA levels in livers measured by RT-qPCR (*n* = 5; five biological repeats). (K) RIP-qPCR detection of *HERVK14-int* RNA associated with RIG-I or MDA5 in RAW 264.7 cells treated with DMSO or BT for 12 h (*n* = 3; three biological repeats). (L) ELISA quantification of secreted type I IFN and RT-qPCR of *IFNB1* mRNA (*n* = 3; three biological repeats). (M) Flow cytometry of VSV-GFP infection (*n* = 3; three biological repeats). Data are shown as mean ± SEM. N.S., not significant, *P* > 0.05; *P < 0.05; **P < 0.01; ****P < 0.0001. Western blot quantifications are presented in Appendix Fig. S2 and exact *P* values are in Appendix Tabe S1. Source data are available online for this figure.

methyltransferases to enforce ERV silencing(Enriquez-Gasca et al, 2023; Guo et al, 2024a; Xu et al, 2021; Zhao et al, 2024). Consistent with prior studies, we found that inhibiting HDACs with BT or losing TRIM28 lead to derepression of over 400 ERV loci and induced a robust type I IFN response. This reinforced observations from cancer epigenetic therapy that reawakening ERVs can trigger innate immunity, but our data highlighted TRIM28 displacement as the critical upstream trigger in this process.

We also identified the endogenous retrovirus HERVK14-int as a principal source of immunogenic dsRNA that was bound by RIG-I and MDA5, consistent with earlier reports linking this element to

innate activation (Mao et al, 2024; Roulois et al, 2015). These results strengthened the concept that the innate immune system can detect reawakened retroviral relics as if they were invading viruses.

Beyond ERV reactivation, an unexpected discovery was that BT directly targeted the cGAS-STING DNA-sensing pathway. To our knowledge, BT was the first small molecule reported to bind and activate cGAS. We showed that BT bound cGAS and allosterically lowered the DNA threshold required for its activation, resulting in greater production of cGAMP and downstream IFN signaling than DNA alone. This contrasts with classical STING agonists, such as

cyclic dinucleotides or diABZI, which bypass cGAS to activate STING (Ramanjulu et al, 2018; Sun et al, 2021b). In contrast, BT engaged the pathway at the top. BT's unique mode of action means it could avoid some limitations of DNA- or STING-targeted agonists, such as the delivery challenges of large DNA ligands or the excessive inflammation caused by direct STING agonists. Classically, cGAS-STING and RLRs can induce TNF-α/IL-6 via NF-κB. Our findings supported selective activation of TBK1-IRF3 with limited NF-κB activation. Future studies will define the upstream basis of this selectivity.

While our data demonstrated that BT directly bound cGAS and functionally lowered its DNA-activation threshold, the precise structural basis of this allostery remains to be defined. In particular, the binding site and conformational consequences on the cGAS catalytic core are not yet resolved, and whether BT modulates cGAS-DNA phase behavior or oligomerization is unknown. Likewise, BT-induced mitochondrial stress and cytosolic dsDNA could be mechanistically coupled to HDAC inhibition, but we cannot exclude parallel BT-specific effects. The residual antiviral activity observed in single-pathway knockouts indicates minor contributions from additional sensors or stress pathways that warrant mapping. Future work will combine SAR with orthogonal biophysics (SPR/ITC across BT analogs, cryo-EM with cGAS mutants), targeted mutagenesis outside the DNA-binding surface, and comparative profiling against benchmark HDAC inhibitors to disentangle HDAC-ERV and cGAS-centric contributions. These studies will refine how BT achieves dual-axis activation while favoring TBK1-IRF3 over broad NF-κB outputs.

BT's concurrent activation of cGAS-STING and RIG-I/MDA5 also made it an attractive immunostimulatory adjuvant. In oncology, BT could heat up immunologically cold tumors by provoking a strong IFN-driven antitumor response (Sun et al, 2021a). ERVs de-repression and cGAS activation together are able to increase tumor immunogenicity and may sensitize cancer cells to T cell-based therapies, especially when paired with immune checkpoint blockade (Jiang et al, 2025; Ng et al, 2023; O'Donnell et al, 2019). Likewise, BT's dual-axis engagement thereby can boost dendritic cells activation and antigen presentation via type I IFNs and other cytokines(Duong et al, 2022; Guo et al, 2024b). Unlike single-pathway agonists, BT engaged both axes with a single orally available compound, potentially achieving a more potent adjuvant effect. Furthermore, BT's direct activation of cGAS might complement existing STING agonists, reinforcing downstream interferon signaling and thereby enhancing the efficacy of vaccines or oncolytic virotherapy(Jiang et al, 2020; Liu et al, 2022).

Another point to be mentioned is safety considerations. It will be important when translating these findings. BT's immunostimulatory effect will likely require controlled, short-term dosing to avoid unintended autoinflammation. Since BT is already FDA-approved (as an anticoagulant) with a well-characterized pharmacokinetic and safety profile could accelerate its repurposing for antiviral or immunotherapy applications (Chi et al, 2018). Strategies such as localized delivery or single-dose regimens might maximize immune benefits while minimizing systemic inflammation.

However, our study also raised several questions that warrant further investigation. First, which specific ERV families are the dominant drivers of the dsRNA that triggers RIG-I/MDA5? We observed broad upregulation of ERVs, with LTR retroelements predominating, but it remained unclear whether a select subset of ERVs accounts for most of the immunostimulatory dsRNA. Future studies could employ RIG-I or MDA5 RNA immunoprecipitation followed by sequencing to pinpoint which ERV-derived transcripts are bound by these sensors. Identifying these primary drivers would clarify how broadly the viral mimicry effect spans the repeatome and whether targeting their specific repressors or repeat classes could further boost the immune response.

Second, is the mitochondrial dsDNA release caused by BT a direct consequence of HDAC inhibition or an off-target effect unique to BT? Comparing BT with other HDAC inhibitors for their ability to induce mitochondrial stress, mtDNA leakage, and dual-pathway activation could help pinpoint the critical structural or biochemical features required. If BT is unique in combining these properties, HDAC inhibition and direct cGAS binding, it defines a novel molecular mechanism and could guide the design of next-generation dual-pathway immunostimulatory agents.

Another important area is the durability of the antiviral state induced by BT. How long after treatment do cells or organisms remain refractory to infection, and does repeated dosing sustain protection or lead to innate immune tolerance? These issues are critical if BT is to be used as a prophylactic or vaccine adjuvant. Ideally, one would want a rapid, transient spike of innate immunity that jump-starts defenses without causing exhaustion. Thus, optimizing dosing regimens will be important. It will also be valuable to test co-administration of BT with vaccines to determine if heightened innate activation translates into stronger adaptive immune responses.

In summary, our work revealed a novel mechanism by which an existing drug can induce an antiviral state through epigenetic derepression and dual innate immune activation. It bridged chromatin biology and immunology, demonstrating that endogenous retroelements can be harnessed as allies in immune defense. Addressing the questions raised above will deepen our understanding of viral mimicry and innate immunity, and will inform the development of BT or next-generation epigenetic modulators as promising antiviral and immunotherapeutic agents.

## Methods

**Reagents and tools table**

| Reagent/resource | Reference or source | Identifier or catalog number |
|---|---|---|
| **Experimental models** | | |
| C57BL6/J (*M. musculus*) | Department of Laboratory Animal Science, Peking University Health Science Center | N/A |
| C57BL6/J (*M. musculus*) cGAS$^{-/-}$ | Dr. Zhengfan Jiang | Peking University |
| C57BL6/J (*M. musculus*) MAVS$^{-/-}$ | Dr. Zhengfan Jiang | Peking University |
| RAW 264.7 (*M. musculus*) | ATCC | RRID: CVCL_0493 |
| HeLa (*H. sapiens*) | ATCC | RRID: CVCL_0030 |

| Reagent/resource | Reference or source | Identifier or catalog number |
|---|---|---|
| HT1080 (*H. sapiens*) | ATCC | RRID: CVCL_0316 |
| HT29 (*H. sapiens*) | ATCC | RRID: CVCL_0320 |
| 17-Cl1 (*M. musculus*) | ATCC | RRID: CVCL_VT75 |
| Vero (*C. aethiops*) | ATCC | RRID: CVCL_0059 |
| hPBMC (*H. sapiens*) | YAYUBIO | N/A |
| **Antibodies** | | |
| Rabbit anti-TBK1 (1:500 dilution) | Proteintech | 28397-1-AP |
| Rabbit anti-Beta Tubulin (1:4000 dilution) | Proteintech | 10094-1-AP |
| Rabbit anti-RIG-I/DDX58 (1:500 dilution) | Proteintech | 25068-1-AP |
| Rabbit anti-MAVS (1:500 dilution) | Proteintech | 14341-1-AP |
| Rabbit anti-IFIT3 (1:500 dilution) | Proteintech | 15201-1-AP |
| Mouse anti-Beta Actin (1:4000 dilution) | Proteintech | 66009-1-Ig |
| Mouse anti-GFP tag (1:8000 dilution) | Proteintech | 50430-2-AP |
| Goat Anti-Rabbit IgG (1:8000 dilution) | Proteintech | SA00001-2 |
| Goat Anti-Mouse IgG (1:8000 dilution) | Proteintech | SA00001-1 |
| Rabbit anti-pTBK1 (Ser172) (1:1000 dilution) | Cell Signaling Technology | 5483 |
| Rabbit anti-pIRF3 (Ser396) (1:1000 dilution) | Cell Signaling Technology | 29047 |
| Rabbit anti-IRF3 (1:1000 dilution) | Cell Signaling Technology | 11904 |
| Rabbit anti-cGAS (1:1000 dilution) | Cell Signaling Technology | 15102 |
| Rabbit anti-STING (1:1000 dilution) | Cell Signaling Technology | 13647 |
| Rabbit anti-MDA5 (1:1000 dilution) | Cell Signaling Technology | 5321 |
| Rabbit anti-OAS2 (1:1000 dilution) | Cell Signaling Technology | 24344 |
| Rabbit anti-Viperin (1:1000 dilution) | Cell Signaling Technology | 75654 |
| Rabbit anti-Acetyl-Histone H3 (Lys27) (1:1000 dilution) | Cell Signaling Technology | 8173 |
| Mouse anti-DNA (1:200 dilution) | Sigma | CBL186 |
| Mouse anti-dsRNA (1:200 dilution) | SCICON | 10010200 |
| **Oligonucleotides and other sequence-based reagents** | | |
| RT-qPCR primers | This study | Table EV1 |

| Reagent/resource | Reference or source | Identifier or catalog number |
|---|---|---|
| **Chemicals, enzymes, and other reagents** | | |
| GM-CSF | ABclonal | RP01206 |
| Betrixaban | MedChemExpress | HY-10268 |
| Recombinant human cGAS protein | MedChemExpress | HY-P72337 |
| DMEM | EallBio | 03.1002C |
| RPMI 1640 | EallBio | 03.4001C |
| VSV Indiana strain | Dr. J. Rose | Yale University |
| VSV-GFP | Dr. J. Rose | Yale University |
| HSV-1 strain 17 | Dr. Zhengfan Jiang | Peking University |
| HSV-GFP | Dr. Zhengfan Jiang | Peking University |
| VACV-GFP | Dr. Zhengfan Jiang | Peking University |
| NDV-GFP | Dr. Zhengfan Jiang | Peking University |
| MHV-A59 strain | ATCC | VR-764 |
| EMCV | ATCC | VR-129B |
| IAV strain PR8 | Dr. Qiang Feng | Fudan University |
| Pseudotyped SARS-CoV-2 variants | Dr. Xuesen Zhao | Peking University |
| HiScript II RT SuperMix | Vazyme | R223-01 |
| SYBR Green qMix | Vazyme | Q311 |
| RIPA Lysis Buffer (Strong) | MedChemExpress | HY-K1001 |
| EDTA-Free Protease Inhibitor Cocktail | MedChemExpress | HY-K0010 |
| Phosphatase Inhibitor Cocktail II | MedChemExpress | HY-K0022 |
| Phosphatase Inhibitor Cocktail III | MedChemExpress | HY-K0023 |
| Nitrocellulose membrane | Beyotime | FFN08 |
| Enhanced chemiluminescence | EallBio | 07.10009-50 |
| human IFNB1 ELISA Kit | BOSTER | EK2286 |
| CCK-8 reagent | YEASEN | 40203ES60 |
| Ethidium bromide | MedChemExpress | HY-D0021 |
| High-throughput RNA extraction kit | TIANGEN | A0123A01 |
| Hyperactive ATAC-Seq Library Prep Kit | Vazyme Biotech | TD711 |
| CUT&Tag Library Prep Kit | KP172 | Transgen |
| **Software** | | |
| ImageJ | https://imagej.net/ij/index.html | N/A |
| GraphPad Prism 10 | www.graphpad-prism.cn | N/A |

| Reagent/resource | Reference or source | Identifier or catalog number |
|---|---|---|
| DESeq2 R package (v1.38.3) | https://bioconductor.org/packages//release/bioc/html/DESeq2.html | N/A |
| Trim-Galore (v0.6.4) | https://www.bioinformatics.babraham.ac.uk/projects/trim_galore/ | N/A |
| Bowtie2 aligner (v2.3.5.1) | https://github.com/BenLangmead/bowtie2 | N/A |
| Samtools (v1.10) | https://sourceforge.net/projects/samtools/files/samtools/1.10/ | N/A |
| MACS3 (v3.0.0a5) | https://github.com/macs3-project/MACS/releases/tag/v3.0.0 | N/A |
| Bedtools merge (v2.31.1) | https://github.com/arq5x/bedtools2/releases | N/A |
| BamCoverage (v3.3.2) | https://deeptools.readthedocs.io/en/latest/content/tools/bamCoverage.html | N/A |
| Csaw R package (v1.38.0) | https://new.bioconductor.org/packages/release/bioc/vignettes/csaw/inst/doc/csaw.html | N/A |
| ChIP seeker R package (v1.34.1) | https://bioc.r-universe.dev/ChIPseeker/doc/ChIPseeker.html | N/A |
| Integrative Genomics Viewer (IGV) (v2.17.4) | https://igv.org/doc/desktop/#DownloadPage/ | N/A |
| Other | | |
| Flow cytometer | BD Biosciences | N/A |
| Cryostat Microtome | Leica | N/A |
| Fluorescence microscope | Nikon | N/A |
| Transmission electron microscope | JEOL | JEM-1400 |

## Reagents and antibodies

Rabbit antibodies against TBK1 (catalog no. 28397-1-AP), Beta Tubulin (catalog no. 10094-1-AP), RIG-I/DDX58 (catalog no. 25068-1-AP), MAVS (catalog no. 14341-1-AP), and IFIT3 (catalog no. 15201-1-AP), along with mouse antibodies targeting Beta Actin (catalog no. 66009-1-Ig) and the GFP tag (catalog no. 50430-2-AP), were procured from Proteintech. Additional antibodies, including rabbit phospho-TBK1/NAK (Ser172) (D52C2) (catalog no. 5483), phospho-IRF-3 (Ser396) (D6O1M) (catalog no. 29047), IRF-3 (D6I4C) (catalog no. 11904), cGAS (D1D3G) (catalog no. 15102), STING (D2P2F) (catalog no. 13647), MDA5 (D74E4) (catalog no. 5321), OAS2 (E2G4K) (catalog no. 24344), Viperin (F7T8D) (catalog no. 75654) and Acetyl-Histone H3 (Lys27) (D5E4) (catalog no. 8173) were obtained from Cell Signaling Technology. Secondary antibody utilized HRP-conjugated Affinipure Goat Anti-Rabbit IgG (H + L) (catalog no. SA00001-2) and HRP-conjugated Affinipure Goat Anti-Mouse IgG (H + L) (catalog no. SA00001-1), were also from Proteintech. ABclonal supplied active recombinant mouse granulocyte-macrophage colony-stimulating

factor (GM-CSF) protein (catalog no. RP01206). Recombinant human cGAS protein were purchased from MedChemExpress (catalog no. HY-P72337). We also used anti-DNA mouse monoclonal antibody (Sigma, CBL186) and anti-dsRNA antibody (J2) (SCICON, 10010200). Betrixaban was bought from MedChemExpress (catalog no. HY-10268).

## Cells

The RAW264.7, HeLa, HT1080, HT29, 17Cl-1, and Vero cell lines were sourced from the American Type Culture Collection (ATCC). Human peripheral blood mononuclear cells (PBMCs) were bought from YAYUBIO. Bone marrow-derived macrophages (BMDMs) were isolated from the femurs and tibiae of 8-week-old C57BL/6J mice. The bone marrow was cultured with GM-CSF for 7 days to induce macrophage differentiation. RAW264.7, HT1080, HT29, 17Cl-1, Vero, and BMDMs were cultured in Dulbecco's Modified Eagle's Medium (DMEM) (03.1002 C, EallBio), while HeLa and HT29 were maintained in RPMI 1640 (03.4001 C, EallBio). All culture media were supplemented with 10% fetal bovine serum (FBS) and 1% penicillin-streptomycin for optimal cell growth and maintenance.

HeLa (epithelial) typically shows low basal cGAS-STING but a competent, inducible RIG-I/MDA5-MAVS axis, whereas HT1080 (mesenchymal/fibroblast-like) displays robust RLR signaling and moderate STING responsiveness, together with RAW264.7 macrophages, this spans distinct innate-immune wirings across species and lineages.

## Virus infection and propagation

Cells at 70–80% confluence were infected with the following viruses at the indicated multiplicity of infection (MOI): vesicular stomatitis virus (VSV, MOI = 0.1), herpes simplex virus 1 (HSV-1, F strain, MOI = 0.5), encephalomyocarditis virus (EMCV, MOI = 0.1), mouse hepatitis virus (MHV, A59 strain, MOI = 0.1), influenza A virus (IAV, PR8 strain, MOI = 0.1) and GFP-expressing viruses, including VSV (MOI = 0.1), HSV (MOI = 0.1) and vaccinia virus (VACV, MOI = 0.1) and newcastle disease virus (NDV, MOI = 0.1).

Virus stocks were prepared as follows: VSV Indiana strain and VSV-GFP, generously provided by J. Rose (Yale University), were propagated in Vero cells. HSV-1 strain 17, VSV-GFP, HSV-GFP, VACV-GFP, NDV-GFP were gifts from Zhengfan Jiang (Peking University), and were cultured in the same cell line for viral amplification. MHV-A59 strain (ATCC, VR-764) and EMCV (ATCC, VR-129B) were obtained from ATCC and propagated in 17Cl-1 and Vero cells, respectively. IAV strain PR8, kindly provided by Qiang Feng (Fudan University), was propagated in fertile chicken eggs under standard conditions. Pseudotyped SARS-CoV-2 variants were provided by Xuesen Zhao (Peking University).

## Mice and in vivo virus infection

Wild-type (WT) C57BL/6J mice were purchased from the Department of Laboratory Animal Science, Peking University Health Science Center. cGAS$^{-/-}$ and MAVS$^{-/-}$ mice, all on a C57BL/6J background, were gifts from Zhengfan Jiang (Peking University).

All animal care and procedures were conducted in accordance with the Guide for the Care and Use of Laboratory Animals by the

Chinese Association for Laboratory Animal Science. The protocols were approved by the Animal Care Committee of Peking University Health Science Center (permit number: LA 2016240). Mice were bred and housed under specific pathogen-free conditions at the Laboratory Animal Center of Peking University. Only mice aged 6–8 weeks were used in the experiments.

Age- and sex-matched C57BL/6 J littermates were used for all in vivo experiments. Six-week-old mice were infected with different viruses at a dose of $1 \times 10^8$ plaque-forming units (PFU) per mouse via intraperitoneal injection (IP). Survivals were monitored daily.

## Hematoxylin–eosin (H&E) staining

The tissues were quickly placed in cold saline solution and rinsed after they were collected, then fixed in 4% paraformaldehyde, dehydrated, and embedded in paraffin prior to sectioning at 5 mm, and sections were stained with hematoxylin and eosin.

## Luciferase assay

After the infection of pseudotyped SARS-CoV-2 variants for 24 h, cells were lysed and measured by the luciferase reporter assay system (Transgen).

## RNA extraction, reverse transcription, and real-time quantitative PCR

RNA extraction from cells or following various treatments or infections was performed using TRIzol reagent (TIANGEN, A0123A01). The purified RNA was then reverse-transcribed into cDNA using HiScript II RT SuperMix (Vazyme, R223-01). Target gene expression was quantified using SYBR Green qMix (Vazyme, Q311) in a quantitative reverse transcription PCR (RT-qPCR) assay. The relative expression levels of target mRNAs were normalized to the housekeeping gene Actb. Detailed primer sequences used in this study are provided in Table EV1.

## Total protein extraction and western blot analysis

Cells were lysed using RIPA Lysis Buffer (Strong) (MedChemExpress, HY-K1001), which was supplemented with protease inhibitors, including an EDTA-Free Protease Inhibitor Cocktail, Phosphatase Inhibitor Cocktail II (100× in DMSO), and Phosphatase Inhibitor Cocktail III (100× in DMSO). Clarified cell extracts (10–30 μg) were resolved on SDS-polyacrylamide gels and then transferred onto nitrocellulose membranes (Beyotime, FFN08). After blocking, the membranes were incubated with specific primary antibodies. The bound secondary antibodies were detected using the enhanced chemiluminescence (ECL) method (EallBio, 07.10009-50).

## Enzyme-linked immunosorbent assay (ELISA) for IFNB1

Cell culture supernatants were collected after 12 h of BT (100 μM) or DMSO treatment. Secreted IFN-β levels were measured using a commercial human IFNB1 ELISA Kit (BOSTER, EK2286) per the manufacturer's protocol. The absorbance was read at 450 nm (with 570 nm reference) on a microplate reader. IFNB1 concentrations were calculated from a standard curve generated with known recombinant IFNB1.

## Cell cytotoxicity assay

We quantified cell viability to exclude non-specific cytotoxicity within the antiviral working range of Betrixaban (MedChemExpress, HY-10268) and to ensure that reduced viral readouts were not driven by loss of viable cells. Cells were seeded in 96-well plates at $2 \times 10^3$ cells per well in 100 μL complete medium and allowed to adhere overnight. The next day, cells were treated with BT at the indicated concentrations (0–x μM) with a final DMSO ≤ 0.1% (v/v); vehicle controls received DMSO only. After 48 h incubation at 37 °C and 5% $CO_2$, 10 μL CCK-8 reagent (YEASEN, 40203ES60) was added to each well, mixed gently, and incubated for 1 h. Absorbance at 450 nm (reference 570 nm, when available) was measured on a microplate reader. For each plate, blank wells containing medium plus CCK-8 without cells were used for background subtraction. Each condition was measured in technical triplicate and repeated in ≥3 independent experiments.

We used the CCK-8 assay to exclude overt cytotoxicity within the antiviral dose range and to clarify that BT-mediated restriction is host-driven rather than a consequence of reduced cell number. Because CCK-8 primarily reports cellular dehydrogenase activity, which can be influenced by metabolic reprogramming, we complemented it with apoptosis readouts and observed no induction across the working concentrations and time windows. In parallel, entry assays were unaffected, and post-entry viral RNA loads and infectious titers declined, supporting a host-mediated post-entry block. Together, these results indicated that BT's antiviral phenotype was not attributable to cytotoxicity or cytostasis but to innate immune activation. We have clarified this interpretation in the text and now explicitly state that viability measurements were used to rule out non-specific toxicity while viral readouts were normalized to housekeeping controls, ensuring that reductions reflect antiviral responses rather than cell loss.

## Fluorescence assay

Cells were infected with GFP-tagged viruses at an MOI of 0.1 for 12 h. After treatment with small molecules, cells were visualized under a Nikon fluorescence microscope equipped with a 488 nm excitation filter and a 510-nm emission filter. Images were captured at a magnification of ×100.

## Flow cytometry analysis

After infection, cells were washed twice with phosphate-buffered saline (PBS) to remove residual virus, followed by trypsinization to generate a single-cell suspension. The cells were collected by centrifugation at 1600 rpm for 5 min, and the supernatant was discarded. The resulting cell pellets were resuspended in PBS and transferred to flow cytometry tubes for subsequent analysis. To evaluate the infection efficiency, the percentage of GFP-positive cells was quantified using a flow cytometer (BD Biosciences).

## CRISPR-Cas9 system

All knockout cell lines were generated using the CRISPR-Cas9 system. Guide RNAs (gRNAs) with high efficiency and specificity were designed and selected using an online CRISPR design tool.

The oligos were annealed and cloned into the lentiCRISPR v2 vector, which had been digested with the BsmBI enzyme (NEB).

For lentivirus production, 293T cells were transfected with the following plasmids: lentiCRISPR v2 (2400 ng), packaging plasmid psPAX2 (800 ng; Addgene 12260), envelope plasmid VSV-G (800 ng; Addgene 8454), and PEI (1600 ng). The transfection mixture was incubated at 37°C for 72 h. Viral supernatants were then collected and used to infect cells in the presence of polybrene (Beyotime Biotechnology, China). 48 h post-infection, cells were refreshed with fresh culture medium and selected using 3 µg/mL puromycin. Successful knockout of cGAS or STING was validated by western blots.

## Surface plasmon resonance (SPR) assay

SPR experiments were performed using the S-Class label-free molecular interaction analysis system, with the assistance of Polar Life Sciences, to evaluate the interaction between cGAS protein and Betrixaban. cGAS protein was immobilized onto a C5 chip via amine coupling using a protein stock solution prepared at 10 µg/mL in 1× PBST (pH 4.5), with an immobilization time of 600 s. Small molecules were initially dissolved in 100% DMSO at a concentration of 10 mM and diluted in 1× PBST containing 5% DMSO before injection. Each analyte was flowed over the immobilized cGAS at a rate of 30 µL/min, with association and dissociation phases set to 60 s and 120 s, respectively. Binding data were collected and analyzed to determine the kinetics and affinities (Kd) of the interactions.

## cGAS-cGAMP activity assay

Purified recombinant human full-length hcGAS was incubated in 50 µL reaction containing: 1 µM hcGAS, 1 mM ATP, 1 mM GTP, 100 mM NaCl, 40 mM Tris-HCl pH 7.5, and 10 mM $MgCl_2$ at 37 °C for 30 min, with or without $5 \times 10^{-3}$ mg/mL dsDNA (3.85 nM). Duplex dsDNA was salmon sperm DNA from Oncorhynchus keta supplied as high molecular weight linear DNA with a molecular mass of $1.3 \times 10^6$ Da. BT was added at 25 µM or 100 µM as indicated. The dsDNA:BT ratios of about $1:6.5 \times 10^3$ and $1:2.6 \times 10^4$ and a cGAS:dsDNA ratio of about 260:1. Reactions were heated at 99 °C and added 200 µL acetonitrile to denature proteins and centrifuged at 14,000 rpm for 40 min. The supernatants were analyzed by LC-MS/MS.

## 2′-3′cGAMP measurement by LC-MS/MS

Mass spectrometric detection was performed on Xevo TQ-S. Samples were analyzed by liquid chromatography–tandem mass spectrometry on a triple-quadrupole instrument equipped with an electrospray ionization source operating in positive-ion mode. Chromatographic separation was achieved on a reverse-phase C18 column, using mobile phase A (water/0.1% formic acid) and mobile phase B (acetonitrile/0.1% formic acid) at a flow rate of 0.3 mL/min. The gradient was held at 2% B for 1 min, ramped to 98% B over 4 min, held for 1 min, then returned to 2% B over 0.5 min. Data were acquired in multiple-reaction-monitoring (MRM) mode. The parent ion ($m/z$ 675.1) was monitored with two transitions: $m/z$ 675.1, 136.0 (adenine fragment) for quantification and $m/z$ 675.1, 152.0 (guanine fragment) for confirmation.

## Immunofluorescence microscopy

Cells on coverslips were washed once with pre-warmed phosphate-buffered saline (PBS) and fixed in 4% (wt/vol) paraformaldehyde at room temperature for 15 min. After three washes in PBS, cells were permeabilized with 0.5% (vol/vol) Triton X-100 for 5 min. After three washes in PBS, cells were blocked in PBS containing 5% (wt/vol) bovine serum albumin (BSA) for 30 min, and incubated with indicated antibodies in PBS containing 3% (wt/vol) BSA for 4 h at room temperature. After three washes, cells were incubated with secondary antibodies or Alexa Fluor 555-conjugated secondary antibodies for 1 h, and then with DAPI for 15 min. The coverslips were washed extensively and mounted onto slides. Imaging of the cells was carried out using a Leica microscope.

## TEM imaging

Imaging was performed on a JEOL JEM-1400 transmission electron microscope operated at 80 kV. Representative fields containing mitochondria were captured to document cristae organization, outer and inner membrane integrity, and other ultrastructural features.

## Mitochondrial DNA depletion

To acutely reduce endogenous mitochondrial DNA (mtDNA), we followed Wang et al (Wang et al, 2018). Briefly, HT1080 cells were cultured in DMEM supplemented with 10% FBS, 4 mM L-glutamine, 4.5 g/L glucose, 100 µg/mL sodium pyruvate, 50 µg/mL uridine, and 100 ng/mL ethidium bromide (EtBr) for 6 days. Depletion was quantified by qPCR as the ratio of mtDNA to nuclear DNA (mtDNA/nDNA).

## RNA immunoprecipitation

After treatments, cells were washed twice with ice-cold PBS, covered with 1 mL PBS, and UV-crosslinked at 254 nm. Cross-linked cells were scraped into PBS, pelleted at 1600 rpm for 5 min at 4 °C, and lysed in 500 µL ice-cold NT2 buffer (50 mM HEPES–KOH pH 7.5, 150 mM KCl, 1 mM $MgCl_2$, 0.05% NP-40) supplemented with protease inhibitors, 1 mM NaF, 0.5 mM DTT, 1 U/µL RNase inhibitor and 2 U/µL DNase I. Lysates were cleared at 14,000 rpm for 15 min at 4 °C. An aliquot (50 µL) was reserved as Input. The remainder was incubated overnight at 4 °C with 2 µg anti-RIG-I or anti-MDA5 antibody pre-bound to 30 µL Protein A/G magnetic beads. Beads were washed four times with high-salt NT2 buffer (50 mM HEPES–KOH pH 7.5, 300 mM KCl, 0.05% NP-40), then treated on-bead with DNase I and RNase inhibitor (37 °C, 15 min). Complexes were eluted by adding 100 µL NT2 buffer containing 0.1% SDS and 0.5 mg/mL Proteinase K, and incubating at 56 °C for 15 min. RNA was extracted from IP and Input fractions using TRIzol, reverse-transcribed with random hexamers, and HERVK14-int levels quantified by qPCR. Enrichment was calculated as percent of Input.

## RNA-seq and data analysis

Total RNA was extracted using the high-throughput RNA extraction kit (TIANGEN, A0123A01). Quality control, library

preparation, sequencing, and data analysis were performed by Suzhou GENEWIZ Biotechnology Company (https://www.genewiz.com.cn/), following established standard protocols. The gene count matrix was generated using featureCounts (v2.0.0). The count data was normalized using the Fragments Per Kilobase Million (FPKM) formula. Differential expression analysis was conducted using the DESeq2 R package (v1.38.3), with the thresholds set at | log2(FoldChange)| >2 and p-value < 0.05 to identify significantly differentially expressed genes (DEGs).

## ATAC-seq

We employed a high-throughput sequencing methodology based on transposase-mediated chromatin accessibility profiling (ATAC-seq). This technique utilized the specificity of Tn5 transposase to selectively cleave accessible regions of chromatin. The transposase, preloaded with DNA sequence adapters, was incubated with isolated cell nuclei, enabling the targeted insertion of adapters into open chromatin regions. Subsequently, indexed primers were utilized for PCR amplification to construct sequencing libraries. The resultant libraries, following sequencing, provided comprehensive insights into DNA regions associated with chromatin accessibility. This experiment was performed using the Hyperactive ATAC-Seq Library Prep Kit (TD711, Vazyme Biotech), with sequencing services conducted by GENEWIZ Biotechnology Company (Suzhou, China).

## CUT&Tag

In this study, cells were immobilized using Concanavalin A-coated Magnetic Beads Pro (ConA Beads Pro) and permeabilized with the nonionic detergent digitonin. The target proteins were then detected using a specific primary antibody, followed by a corresponding secondary antibody and Protein A/G, forming an immunocomplex. To tether the Tn5 transposase to the target protein, it was fused to Protein A/G and guided by the antibody complex. Upon activation with $Mg^{2+}$, the Tn5 transposase catalyzed targeted DNA cleavage, simultaneously inserting adapter sequences at both ends of the fragmented DNA. These DNA fragments were then amplified via PCR for library construction. The entire process was performed using the CUT&Tag Library Prep Kit (Transgen, KP172), with sequencing services provided by GENEWIZ Biotechnology Company (Suzhou, China).

## Data analysis of ATAC-seq and CUT&Tag

For ATAC-seq and CUT&Tag high-throughput data, sequenced by GENEWIZ Biotechnology, raw fastq data underwent quality control using Trim-Galore (v0.6.4). Filtered reads were then aligned and quantified against the mouse genome mm10 using the bowtie2 aligner (v2.3.5.1), generating SAM or BAM files. The bowtie2 alignment parameters were: bowtie2 --verysensitive -X 2000 -x $Bowtie2Index -1 $cln_res/${sample}_1_val_1.fq.gz -2 $cln_res/${sample}_2_val_2.fq.gz -p $PPN | samtools view -buSh -@ $PPN | samtools sort -@ $PPN -O BAM -o $aln_res/${sample}.sorted.bam. Sorting and indexing of SAM/BAM files were completed using samtools (v1.10) with the command: samtools index -@ $PPN $aln_res/${sample}.sorted.bam.

Peak calling was performed using MACS3 (v3.0.0a5) with default parameters, yielding peak files in BED format for each sample. For downstream analysis, different strategies were applied to ATAC-seq and CUT&Tag data. ATAC-seq peaks were merged across all samples using the bedtools merge (v2.31.1) function, followed by normalization with bamCoverage (v3.3.2) in the deepTools suite (49) using the command: bamCoverage --bam $var -o ${var%.*}.bw --binSize 100 --normalizeUsing RPKM --effectiveGenomeSize 2864785220 --ignoreForNormalization chrM –extendReads. For CUT&Tag data, reads in peak regions were normalized by RPKM using the same bamCoverage parameters. Comparisons between treatment and input samples were done using bamCompare with the following command: bamCompare -b1 idx-MUTMSCLACH.bam -b2 idx-MUTMSCLAIN.bam -o bwResIpInput/MUTIPratioInput --binSize 145 --normalizeUsing RPKM --effectiveGenomeSize 2864785220 --ignoreForNormalization chrM --extendReads -p 22 --scaleFactorsMethod None. Differential analysis for both ATAC-seq and CUT&Tag was conducted using the csaw R package (v1.38.0), while peak annotation was performed using the ChIP seeker R package (v1.34.1). Finally, peak visualization was achieved using Integrative Genomics Viewer (IGV) (v2.17.4).

## GREAT analysis

GREAT (Genomic Regions Enrichment of Annotations Tool) was a powerful resource for functional enrichment analysis of unannotated genomic regions, especially in non-coding areas. By associating these regions with nearby coding genes, GREAT allowed researchers to infer potential biological functions. This approach was beneficial for studying cis-regulatory elements, such as the differentially accessible regions (DARs) identified through techniques like ATAC-seq, ChIP-seq, or CUT&Tag. By leveraging GREAT, we can gain insights into the functional roles of these unannotated regions, enhancing our understanding of genome regulation and gene expression control.

## Graphics

Synopsis Graphic was created with BioRender.com.

## Quantification and statistical analysis

The Student's $t$ test was used to analyze data. The two compared groups were labeled by square brackets. The ANOVA test was used to analyze experiments involving multiple tests and variables. Survival curves were compared using the Mantel–Cox test. For the bar graph, one representative experiment of at least three independent experiments is shown, and each was done in triplicate. Data are shown as mean ± SEM ($n \geq 3$). N.S. not significant, $P > 0.05$; *$P < 0.05$; **$P < 0.01$; ***$P < 0.001$; ****$P < 0.0001$.

# Data availability

The sequencing datasets produced in this study are available in the following databases: RNA-Seq data, ATAC-Seq data and CUT&Tag-Seq data: Sequence Read Archive PRJNA1261098.

## The paper explained

### Problem

Most antivirals act on a single viral target, which can limit spectrum and enable resistance. A complementary approach is to boost the patient's own innate antiviral sensors, but current tools either stimulate only one pathway (for example, STING alone) or face delivery and safety hurdles. There is also no widely used small-molecule that directly activates cGAS, the DNA sensor that initiates interferon responses. An oral, systemically deliverable agent that engages multiple innate sensors could provide broad protection and clinical flexibility.

### Results

We discovered that Betrixaban (BT)—an FDA-approved oral drug—activates innate immunity through two mechanisms. First, BT directly binds and allosterically sensitizes cGAS, increasing cGAMP production and downstream interferon signaling. Second, BT shows HDAC-inhibitory activity that de-represses endogenous retroviruses (ERVs), generating double-stranded RNA that engages RIG-I/MDA5. Across cell types and in mice, BT induces IFN programs and restrains diverse DNA and RNA viruses; genetic loss of TBK1/IRF3 or MAVS/STING abrogates protection, confirming pathway dependence.

### Impact

BT exemplifies a host-directed, dual-pathway antiviral strategy that could broaden coverage and reduce resistance risk. Because BT is oral and already has a human safety/PK record, repurposing may accelerate clinical translation—as a prophylactic during outbreaks or as an adjuvant to enhance vaccine or oncolytic responses. Its upstream action at cGAS plus ERV-mediated RIG-I/MDA5 engagement suggests strong interferon benefits with potential to limit excessive inflammation when appropriately dosed; this provides a feasible path toward clinically practical immunostimulation.

The source data of this paper are collected in the following database record: biostudies:S-SCDT-10_1038-S44321-025-00356-7.

## Peer review information

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

## Acknowledgements

This work was supported by the National Key Research and Development Program of China (2021YFC2302602), the Beijing Natural Science Foundation (Z210014), and the National Natural Science Foundation of China (NSFC 8235071080, 3247080250, 31570891, 31872736, 32022028, 81991505, 82201928), awarded to Prof. Fuping You. The funders had no role in study design, data collection and analysis, decision to publish, or preparation of the manuscript.

## Author contributions

**Xingyu Chen**: Funding acquisition. **Yang Zhao**: Conceptualization; Resources; Software; Formal analysis; Supervision; Investigation; Visualization; Methodology; Writing—original draft; Project administration; Writing—review and editing. **Yunfei Xie**: Data curation; Formal analysis. **Tianyi Liu**: Investigation. **Haocheng Wang**: Investigation. **Xiao Wang**: Funding acquisition. **Xuefei Guo**: Investigation. **Fuping You**: Funding acquisition.

Source data underlying figure panels in this paper may have individual authorship assigned. Where available, figure panel/source data authorship is listed in the following database record: biostudies:S-SCDT-10_1038-S44321-025-00356-7.

## Disclosure and competing interests statement

The authors declare no competing interests.

# Expanded View Figures

**Figure EV1. Betrixaban (BT) establishes a host-protective antiviral state in vitro and in vivo.**

(A) Representative density contour plot images of flow cytometry. RAW 264.7 cells treated with DMSO (Con) or 100 μM BT and infected with HSV-GFP, VACV-GFP, VSV-GFP or NDV-GFP (MOI = 0.1). (B) RT-qPCR quantification of viral RNA, luciferase to detect BA.2.86, D614G, BA.2, BA.1, Delta and fluorescence (green) and bright-field images in HeLa cells. (C, D) RT-qPCR quantification of viral RNA, *Il1b, Il6, Tnfa* mRNA treated with DMSO (Con) or 100 μM BT and infected with different viruses for 12 h in BMDMs. (E, F) RT-qPCR and luciferase to detect *Il1b, Il6, Tnfa* mRNA treated with DMSO (Con) or 100 μM BT and infected with different viruses for 12 h in RAW 264.7 cells. (G) Virus titers in RAW264.7 and HT1080 cells infected with viruses following the treatment of BT. (H) RT-qPCR of *Il1b, Il6* and *Tnfa* in RAW 264.7 cells following BT treatment (50, 75, 100 μM) for 12 h. (I) In vivo detection of *Il1b, Il6* and *Tnfa* mRNA in blood, heart, liver, lung, spleen and kidney of C57BL/6 J mice 6 h after a single intraperitoneal injection of BT (50 mg/kg). (J) Time-of-addition assay. After VSV or HSV-1 infection (37 °C 1 h; washes), BT (50, 75, 100 μM) was added at 0, +1, +2, +4, or +6 h post-infection. We quantified viruses mRNA using RT-qPCR. (K) 4 °C attachment assay. Cells were pre-chilled and exposed to VSV or HSV-1 at 4 °C for 1 h, followed by ice-cold washes. Viral genomes bound to the cell surface were quantified by RT-qPCR. (L) 37 °C internalization assay. After 4 °C binding and washes, cells were shifted to 37 °C for 60 min to allow uptake, then subjected to a brief trypsin wash to remove surface-bound virions. Intracellular viral RNA was measured by RT-qPCR. (M) Cell viability measured by CCK-8 assay in RAW 264.7, HeLa, HT1080, HT29, PBMCs and BMDMs (top). *IFNB1* (RT-qPCR) and VSV-GFP (%GFP, flow cytometry) dose-responses in RAW264.7 cells, EC50/IC50 indicated (bottom). (N) Virus titers in the livers of mice infected with a lethal dose of VSV, HSV-1 with or without the treatment of BT (as described in the "Methods"). (O–Q) Mice received daily intraperitoneal injections for 14 consecutive days of either no treatment (MOCK), solvent alone (PBS), or the test compound (BT, 50 mg/kg/day). At the end of the dosing period, major organs were harvested for analysis. Body weight and organ weights (heart, liver, spleen, lung, kidney) (O), representative photographs of dissected organs (P), histological examination of organs. Scale bars, 100 μm (×20) (Q). Data are shown as mean ± SEM. N.S., not significant, $P > 0.05$; *$P < 0.05$; **$P < 0.01$; ***$P < 0.001$.

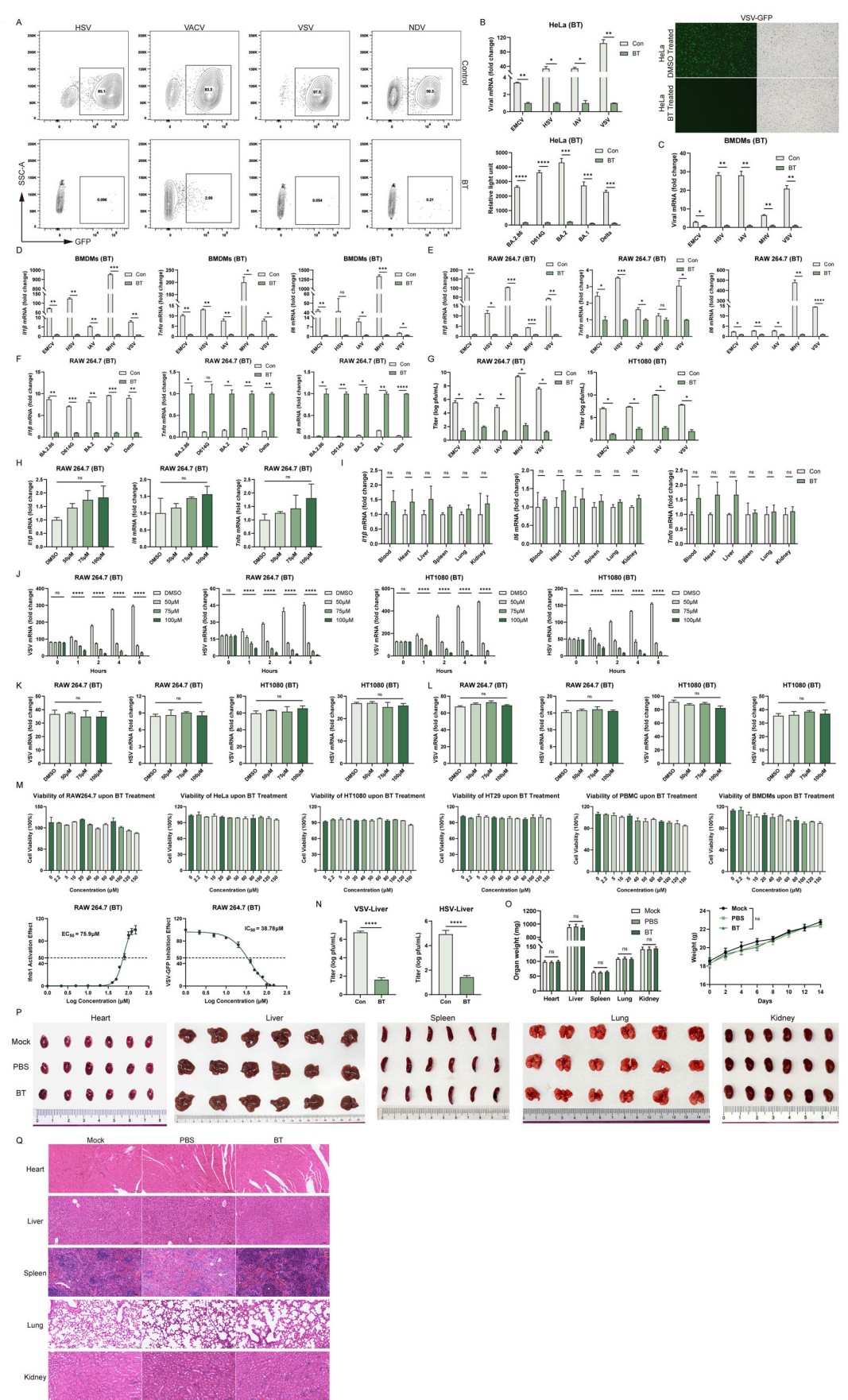

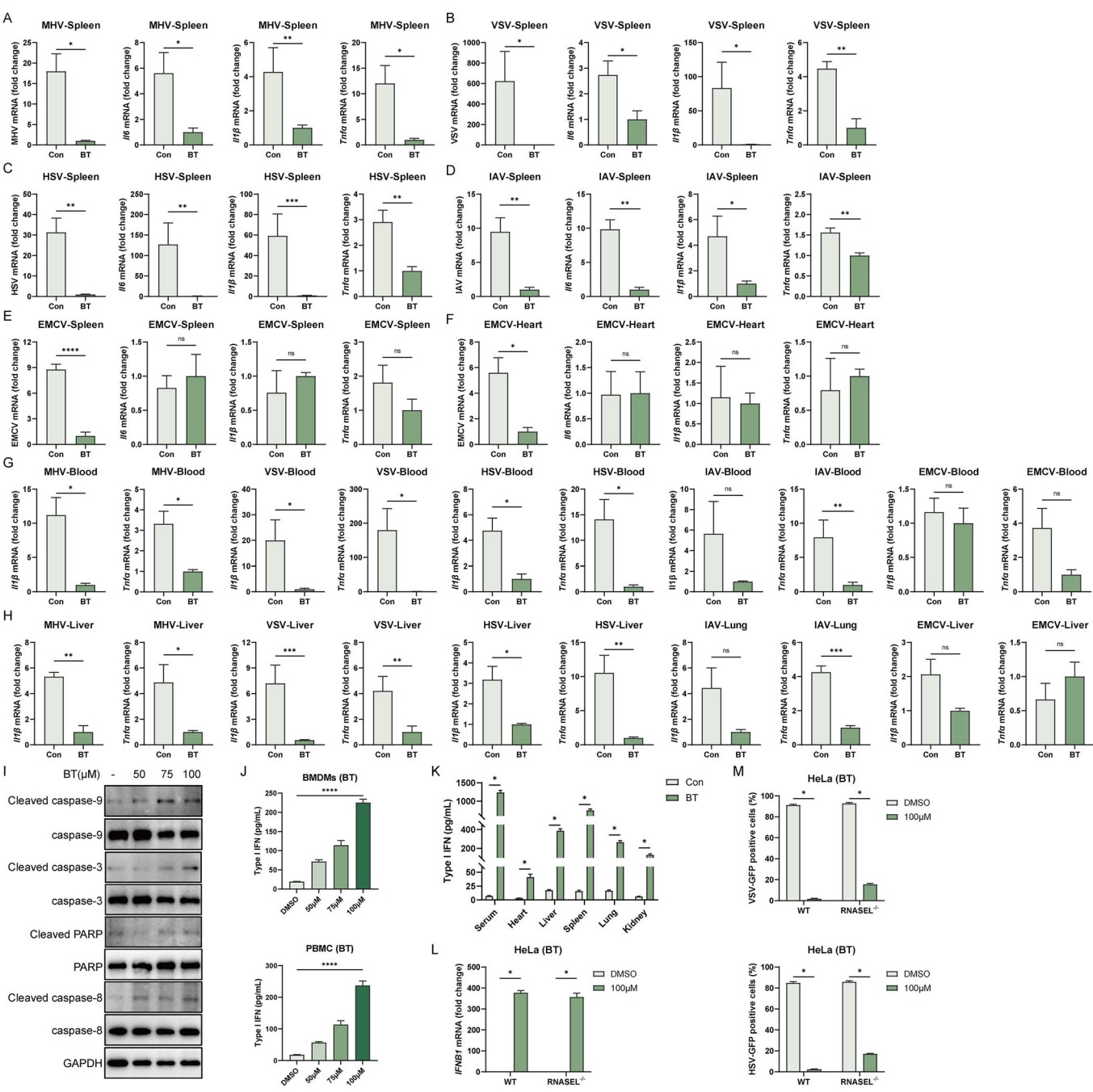

**Figure EV2. Betrixaban (BT) establishes a host-protective antiviral state in vivo.**

(A–H) BT protected C57BL/6 J mice from lethal viral challenge. Mice infected intraperitoneally with MHV (n = 10), VSV (n = 12), HSV-1 (n = 11), IAV (n = 13) or EMCV (n = 13) and treated daily with vehicle or BT (50 mg/kg). RT-qPCR quantification of viral RNA and *Il-6, Il1b, Tnfa* mRNA in target organs harvested 24 h post-infection. (I) Western blot of total and cleaved caspase-3, total and cleaved PARP, total and cleaved caspase-9, and total and cleaved caspase-8. GAPDH served as a control, the samples were same to Fig. 2D. (J) Secreted type I IFN measured by ELISA in the same BMDMs and PBMCs and treatments as in Fig. 2E,F. (K) Secreted type I IFN measured by ELISA in mice serum and targeted organs. Six-week-old C57BL/6J mice 6 h after a single intraperitoneal dose of BT (50 mg/kg). (L) RT-qPCR quantification of *IFNB1* mRNA in HeLa cells treated with indicated concentrations of BT for 12 h. (M) HeLa cells treated with DMSO (Con) or 100 μM BT and infected with HSV-GFP or VSV-GFP (MOI = 0.1), measured by flow cytometry. Data are shown as mean ± SEM. N.S., not significant, P > 0.05; *P < 0.05; **P < 0.01; ***P < 0.001; ****P < 0.0001. Western blot quantifications are presented in Appendix Fig. S2.

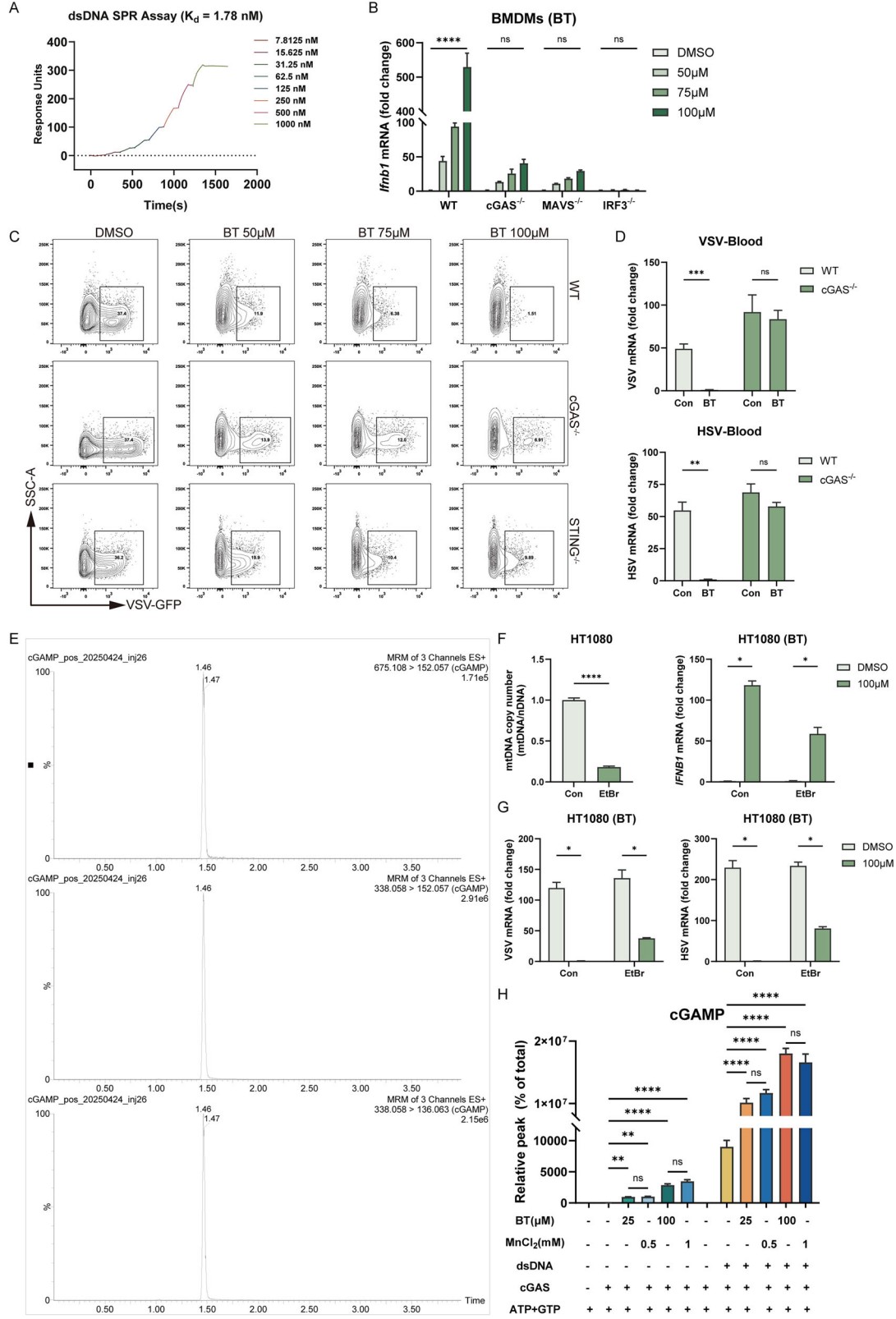

◄ **Figure EV3. Betrixaban directly binds and sensitizes cGAS activation.**

(A) SPR sensorgrams of salmon sperm double-stranded DNA binding to recombinant human cGAS at the indicated concentrations. (B) RT-qPCR of *Ifnb* in BMDMs from WT, cGAS$^{-/-}$, MAVS$^{-/-}$, and IRF3$^{-/-}$ mice following 100 μM BT treatment for 12 h. (C) Representative density contour plot images of flow cytometry of VSV-GFP infection in WT, cGAS knockout and STING knockout HT1080 cells treated with BT (50, 75, 100 μM) and infected (MOI = 0.1) for 12 h. (D) Viruses RNA levels in blood measured by RT-qPCR, the treatment is the same to Fig. 3H,I. (E) LC-MS/MS characterization of cGAMP standard and monitored ion transitions. (F) HT1080 cells were cultured with or without 100 ng/ml ethidium bromide for 6 days. Depletion of mtDNA was measured by Quantitative PCR of mtDNA versus genomic DNA (left). Cells were treated with 100 μM BT for 12 h, *IFNB* mRNA level were monitored by RT-qPCR (right). (G) After the depletion of mtDNA, cells were infected with VSV or HSV following the treatment of 100 μM BT or DMSO, RT-qPCR quantification of viral RNA 12 h post-infection. (H) In vitro cGAS enzymatic assays: LC-MS quantification of cGAMP production by recombinant cGAS incubated with BT or Mn$^{2+}$. Data are shown as mean ± SEM. N.S., not significant, $P > 0.05$; *$P < 0.05$; **$P < 0.01$; ***$P < 0.001$; ****$P < 0.0001$.

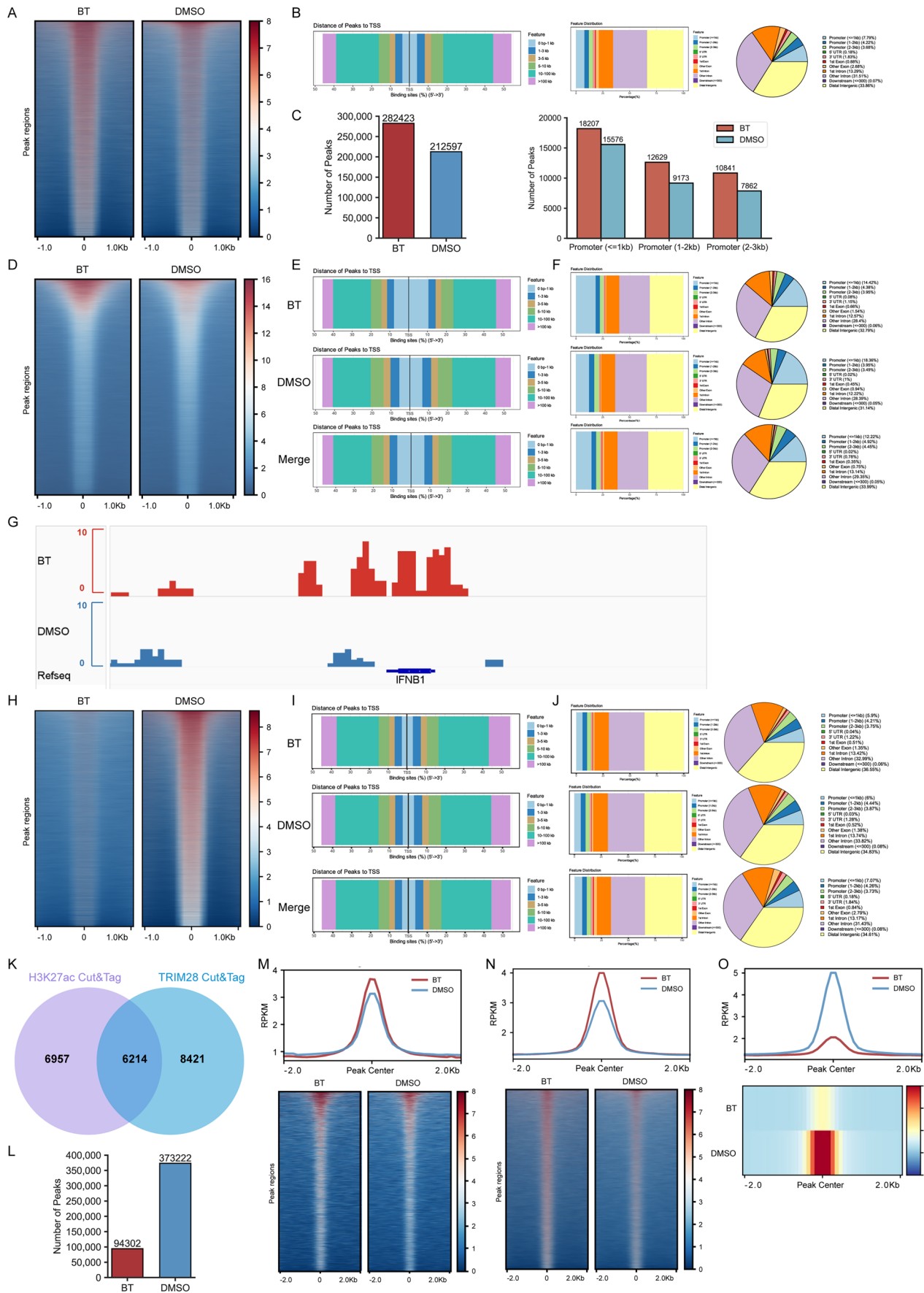

◀ **Figure EV4. Betrixaban remodels chromatin and reactivates ERVs.**

(A) Heatmaps of H3K27ac CUT&Tag signal intensity (RPKM) ±1 kb around all called H3K27ac peak centers in BT- versus DMSO-treated cells. (B) Genomic annotation of H3K27ac peaks. distance to the nearest transcription start site (left), bar plot of feature distribution (middle), and pie chart summarizing the proportion of peaks in each genomic feature (right). (C) Total number of H3K27ac peaks called in BT and DMSO (left), and number of H3K27ac peaks mapping to promoter regions stratified by distance from the TSS ( ≤1 kb, 1–2 kb, 2–3 kb; right). (D) Heatmaps of ATAC-seq signal intensity ±1 kb around all ATAC peak centers in BT- versus DMSO-treated cells. (E) Distance-to-TSS annotation for ATAC-seq peaks in BT, DMSO and merged samples. (F) Genomic feature distribution of ATAC peaks in BT, DMSO and merged datasets, shown as bar plots and pie charts. (G) Genome browser (IGV) snapshots at IFNB1 loci showing H3K27ac CUT&Tag. (H) Heatmaps of TRIM28 CUT&Tag signal intensity ±1 kb around all TRIM28 peak centers in BT- versus DMSO-treated cells. (I) Distance-to-TSS annotation for TRIM28 peaks in BT (top), DMSO (middle) and merged samples (bottom). (J) Genomic feature distribution of TRIM28 peaks in BT, DMSO and merged datasets, shown as bar plots and corresponding pie charts. (K) Venn diagram showing overlap between H3K27ac gain and TRIM28 loss at ERV loci upregulated by BT. (L) Total number of ATAC-seq peaks called in BT versus DMSO samples at ERV loci. (M) Meta-profile (top) and heatmap (bottom) of H3K27ac CUT&Tag signal ±2 kb around the centers of ERV loci that were upregulated by BT. (N) Meta-profile (top) and heatmap (bottom) of TRIM28 CUT&Tag signal at the same set of BT-reactivated ERV loci, illustrating loss of TRIM28 binding. (O) Meta-profile (top) and heatmap (bottom) of ATAC-seq signal at BT-reactivated ERV loci, showing increased chromatin accessibility upon BT treatment.

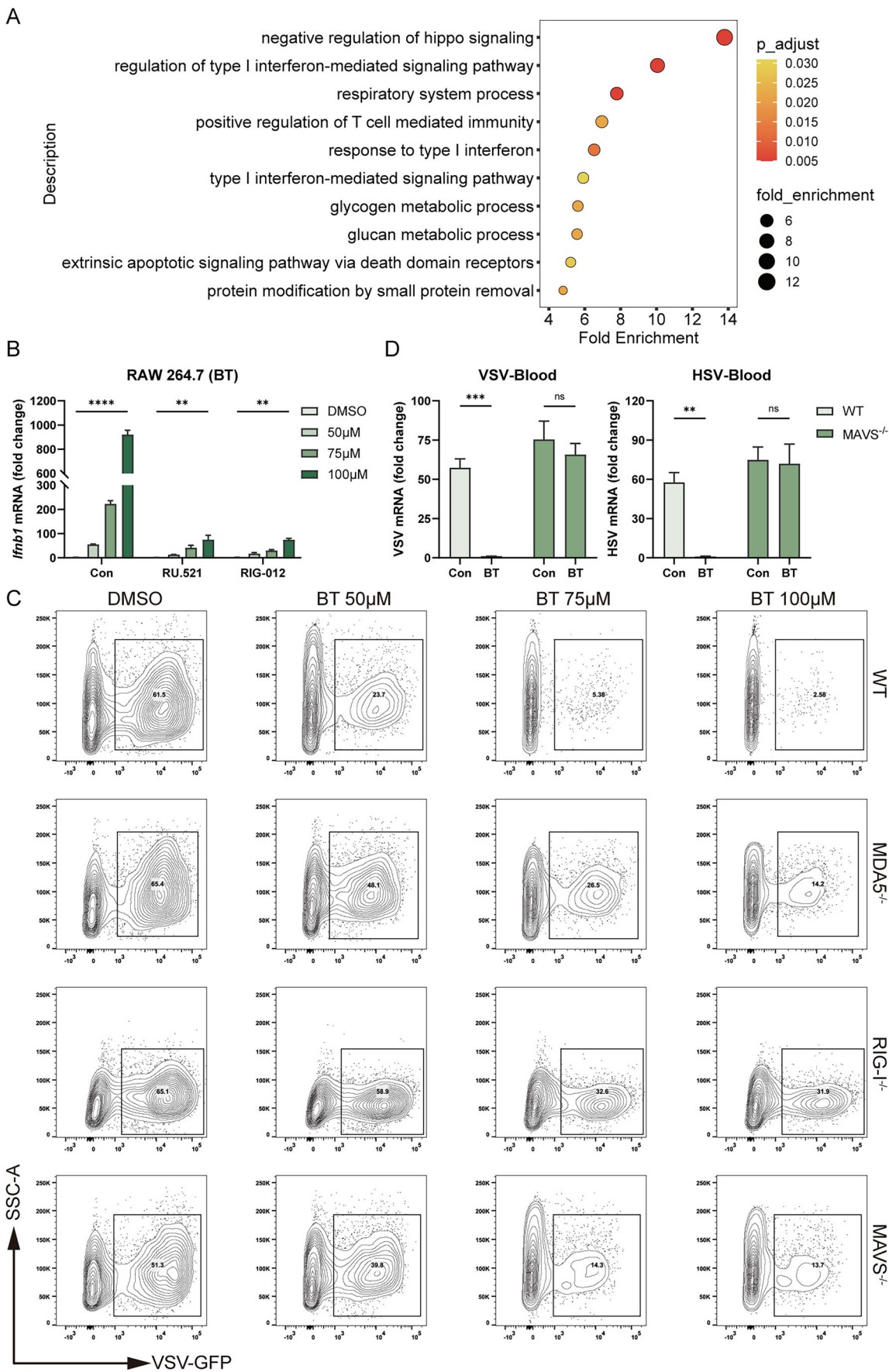

◀

**Figure EV5. Reactivated ERVs mediate the cytosolic dsRNA-sensing pathway.**

(A) Gene Ontology terms enriched among BT-reactivated ERV loci. (B) RT-qPCR of *Ifnb1* in RAW 264.7 cells after BT treatment (100 μM, 12 h) with or without cGAS inhibition (RU.521) or RIG-I inhibition (RIG-012). (C) Representative density contour plot images of flow cytometry, the treatment was the same to Fig. 5H. (D) Viruses RNA levels in blood measured by RT-qPCR, the treatment is the same to Fig. 5I,J. Data are shown as mean ± SEM. N.S., not significant, $P > 0.05$; *$P < 0.05$; **$P < 0.01$; ***$P < 0.001$; ****$P < 0.0001$.

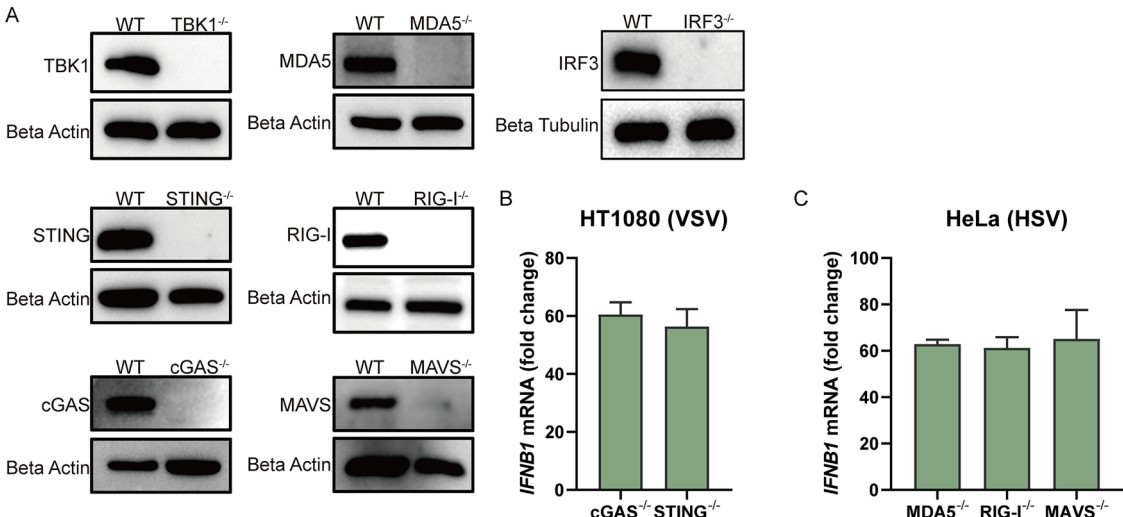

**Figure EV6. Knockout cell lines.**

(A) Western blot analyses of all knockout cell lines used in this study. (B, C) RT-qPCR to test the functions of knockout cell lines. Data are shown as mean ± SEM. N.S., not significant, $P > 0.05$; *$P < 0.05$; **$P < 0.01$; ***$P < 0.001$; ****$P < 0.0001$. Western blot quantifications are presented in Appendix Fig. S2.

