## [Peer Review File · EMBO Molecular Medicine]

Betrixaban Activates cGAS and ERVs to Promote Dual Nucleic-Sensing Antiviral Immunity

Xingyu Chen, Yang Zhao, Yunfei Xie, Tianyi Liu, Haocheng Wang, Xiao Wang, Xuefei Guo, and Fuping You

Corresponding author(s): Yang Zhao (2311210031@stu.pku.edu.cn) , Fuping You (fupingyou@hsc.pku.edu.cn), Xuefei Guo (guoxf2025@163.com), Xiao Wang (xwang2015@bjmu.edu.cn)

Review Timeline:

Submission Date:	27th May 25
Editorial Decision:	18th Jun 25
Revision Received:	30th Sep 25
Editorial Decision:	21st Oct 25
Revision Received:	19th Nov 25
Accepted:	24th Nov 25

Editor: Zeljko Durdevic

Transaction Report:

18th Jun 2025

Dear Dr. Zhao,

Thank you for the submission of your manuscript to EMBO Molecular Medicine. We have now received feedback from the three reviewers who agreed to evaluate your manuscript. As you will see from the reports, all three referees recognize potential interest of the study, but they also raise serious concerns that should be addressed in a major revision. If you would like to discuss further the points raised by the referees, I am available to do so via email or video. Let me know if you are interested in this option.

We would welcome the submission of a revised version within three months for further consideration. Please let us know if you require longer to complete the revision.

I look forward to receiving your revised manuscript.

Yours sincerely,

Zeljko Durdevic

Zeljko Durdevic
Senior Editor
EMBO Molecular Medicine

We require:

- 1) A .docx formatted version of the manuscript text (including legends for main figures, EV figures and tables). Please make sure that the changes are highlighted to be clearly visible.
- 2) Individual production quality figure files as .eps, .tif, .jpg (one file per figure). For guidance, download the 'Figure Guide PDF': (<https://www.embopress.org/page/journal/17574684/authorguide#figureformat>).
- 3) A .docx formatted letter INCLUDING the reviewers' reports and your detailed point-by-point responses to their comments. As part of the EMBO Press transparent editorial process, the point-by-point response is part of the Review Process File (RPF), which will be published alongside your paper.
- 4) A complete author checklist, which you can download from our author guidelines (<https://www.embopress.org/page/journal/17574684/authorguide#submissionofrevisions>). Please insert information in the checklist that is also reflected in the manuscript. The completed author checklist will also be part of the RPF.
- 5) Please note that all corresponding authors are required to supply an ORCID ID for their name upon submission of a revised manuscript.
- 6) It is mandatory to include a 'Data Availability' section after the Materials and Methods. Before submitting your revision, primary datasets produced in this study need to be deposited in an appropriate public database, and the accession numbers and

database listed under 'Data Availability'. Please remember to provide a reviewer password if the datasets are not yet public (see <https://www.embopress.org/page/journal/17574684/authorguide#dataavailability>).

12) Author contributions: You will be asked to provide CRediT (Contributor Role Taxonomy) terms in the submission system. These replace a narrative author contribution section in the manuscript.

13) A Conflict of Interest statement should be provided in the main text.

14) Every published paper now includes a 'Synopsis' to further enhance discoverability. Synopses are displayed on the journal webpage and are freely accessible to all readers. They include a short stand first (maximum of 300 characters, including space) as well as 2-5 one-sentences bullet points that summarizes the paper. Please write the bullet points to summarize the key NEW findings. They should be designed to be complementary to the abstract - i.e. not repeat the same text. We encourage inclusion of key acronyms and quantitative information (maximum of 30 words / bullet point). Please use the passive voice. Please attach these in a separate file or send them by email, we will incorporate them accordingly.

15) Include a Reagents and Tools Table as part of the Methods section, which can be downloaded from our author guidelines (<https://www.embopress.org/page/journal/17574684/authorguide#structuredmethods>)

***** Reviewer's comments *****

Referee #1 (Remarks for Author):

In this manuscript, Chen and colleagues show that an FDA-approved oral Factor Xa inhibitor, Betrixaban (BT), activated both cGAS-STING and RLRs pathway to establish antiviral state. The authors show that BT directly bound and activated the DNA sensor cGAS to induce cGAMP production, and activated RLRs by inhibiting HDACs to cause chromatin de-repression of ERVs and the production of immunostimulatory dsRNA. Insights into structural and cellular checkpoints that control and terminate cGAS-STING and RLRs signaling are essential for comprehending and manipulating innate immunity in health and disease. The observation of BT in regulating dual nucleic acid-sensing pathway is interesting, which provides an orally broad-spectrum antiviral strategy by using BT. However, there are several concerns that should be addressed.

Major concerns:

1. The authors claimed that BT induced mitochondrial stress, leading to the release of endogenous dsDNA. Since BT affected mitochondrial activity, its impact on the apoptosis pathway needs to be investigated.
2. The authors demonstrated that BT increased IFN expression whereas decreased TNF- α , IL-6 and IL-1 β expression. However, both cGAS-STING and RLRs activation lead to the expression of TNF- α and IL-6 through NF- κ B, the dichotomous effects of BT on IFN production versus TNF- α /IL-6 induction are weird and need mechanistic investigation. Furthermore, the production of IL-1 β indicated that the inflammasomes, including NLRP3 inflammasome (could be activated by mitochondrial stress), were activated. So, why did BT, which promoted mitochondrial stress, paradoxically suppressed inflammation? The authors should provide a mechanistic explanation.
3. In Fig 3D and 3E, the expression of IFN was partially suppressed in cGAS- and STING-deficiency, why was there no difference in survival rates between these genotypes in Fig 3H and 3I? The same issue also appeared in Fig 5. Furthermore, while cGAS- and STING-deficient mice showed no difference versus WT in Fig 3H and 3I (indicating cGAS-STING pathway dominance over RLRs in BT-mediated antiviral immunity), Fig 5I, and 5J demonstrated the similar trend (indicating RLRs pathway dominance over cGAS-STING pathway in BT-mediated antiviral immunity). How to reconcile this contradiction?

Minor concerns:

1. The quantification of viral replication was purely by detecting viral nucleic acid or protein components. This is quite disappointing as that would not reflect the viral load, which can be readily determined by plaque assay.
2. Fig 2E, p-TBK1 was detectable even under unstimulated conditions, indicating basal activation of the signaling pathway.
3. Fig 2D-2F, IFN- β secretion needs to be measured by ELISA.
4. Fig 3B, it is necessary to add positive and negative controls to compare the dissociation constant of BT and cGAS.
5. Fig 3D, which two groups are compared by "ns"? Because there appears to be a statistical difference.

Referee #2 (Comments on Novelty/Model System for Author):

The study identifies FDA-approved compound as a novel broad-spectrum antiviral small-molecule agent targeting nucleic acid sensing pathways.

Referee #2 (Remarks for Author):

The manuscript submitted by Xingyu Chen et al. reports that Betrixaban (BT), as an FDA-approved oral Factor Xa inhibitor, displays robust antiviral activity through dual innate immune pathways. The authors presented evidence demonstrating that BT can bind directly to cGAS to produce cGAMP. In addition, they also showed that BT could induce mtDNA leakage and enhance dsDNA-triggered cGAS activation, leading to robust cGAMP production and activation of downstream IFN signaling. Additionally, BT exhibits HDAC inhibitory activity, leading to de-repression of ERVs and production of dsRNA that engaged RIG-1/MDA5.

Collectively, the authors identify BT as a novel broad-spectrum antiviral small-molecule agent targeting nucleic acid sensing pathways and provide a unique host-directed antiviral strategy. However, further clarification is required prior to the publication of this study. I would ask the authors to address the points as shown below:

Major comments:

- 1) While BT-induced mitochondrial dsDNA release and BT-induced weak activation of cGAS have been demonstrated, whether the strong antiviral effects of BT via the cGAS-STING pathway are strictly dependent on the co-existence of both mechanisms. Please clarify it.
- 2) In Figure 3, the reported magnitude of BT's enhancement on dsDNA-mediated cGAS activation appears remarkably strong, showing a potent synergistic effect. The cGAMP produced in the BT-cGAS group appears much weaker than in the dsDNA-cGAS group. Please clarify the molar ratios of dsDNA to BT in all relevant experiments and present Figure 3N and O in a single figure for direct comparison. In addition, comparing BT with another weak cGAS activator, Mn²⁺, would also help to evaluate the effectiveness of BT-mediated cGAS activation. If the BT-mediated cGAS activation is very weak, the authors should tone down the conclusion regarding BT as a cGAS agonist, avoiding overinterpretations of the data.
- 3) Data on p-IRF3 are not provided in this manuscript. Since the authors showed that BT activated innate immune signaling via cGAS-STING pathway and RIG-I/MDA5-MAVS pathway, they should check the p-IRF3 signaling, which is the essential downstream signaling of both these two pathways. In addition, knockout of IRF3 (antiviral experiments and activation of innate immune experiments) would also help to strengthen the mechanism of this study.
- 4) I wonder whether the OAS signaling contributes to the antiviral effect? Given that both OAS and dsRNA were induced by BT, would it be possible that OAS signaling would be activated?

Minor comments:

- 1) The compound seems to work only at high concentration (>50 μ M); how would the authors interpret this high concentration-mediated effect? Could the authors provide the IC₅₀ (antiviral) or EC₅₀ (activation of immunity) values?
- 2) Were the inhibition effects of pro-inflammatory cytokines in Figure S1D-S1F caused by the reduction of viruses? The activation of the cGAS-STING pathway and RIG-I/MDA5-MAVS pathway should also induce the inflammatory cytokines. Would BT treatment activate them (e.g. Figure 2G)?
- 3) Could the authors measure the IFN- β protein level in mice after BT treatment?
- 4) Since the dsRNA-mediated immune response is redundantly regulated by TBK1/IKK ϵ , how would the authors reconcile the complete impairment of immune signaling in TBK1 knockout cells (Figure 2H)?
- 5) Figure S3B and S3C: the WT mice and cGAS-KO mice should be compared in a side-by-side manner.
- 6) Figure 3J: how was dsDNA stained? Figure 3K: how would the MitoTracker staining data indicate mitochondrial change?
- 7) Would the TBK1-STING-IRF3 signaling cascades and MAVS signaling be activated in mouse cell lines, such as RAW 264.7 cells or BMDMs?
- 8) Some data should be quantified, for example, Figure 2I.
- 9) The manuscript contains several typographical errors and grammatical inaccuracies. Please carefully check the manuscript and correct them.

Referee #3 (Comments on Novelty/Model System for Author):

The manuscript would benefit from a clearer explanation of the cell lines used (RAW 264.7, HeLa, and HT1080), including the rationale behind their selection. Providing this information will help readers better understand the experimental design and the relevance of the results. Additionally, the 100 μ M concentration appears to be quite high. A cell viability assay at this dose for the tested cell lines, as well as for non-transformed (healthy) cells, is essential to assess potential cytotoxic effects and to distinguish antiviral activity from general toxicity.

Referee #3 (Remarks for Author):

In this manuscript, Chen and colleagues propose a novel mechanism by which Betrixaban (BT), an FDA-approved oral factor Xa inhibitor and anticoagulant, induces antiviral innate immune responses through activation of both the DNA-sensing cGAS-STING pathway and the RNA-sensing MAVS pathway. They suggest that this immunomodulatory activity is mediated by the direct binding of BT to cGAS, leading to the production of cGMP. Furthermore, they report that BT inhibits histone deacetylases (HDACs), resulting in epigenetic derepression of endogenous retroviruses (ERVs) and the subsequent accumulation of double-stranded RNAs.

While the study presents an intriguing and potentially impactful hypothesis, several concerns arise regarding the data presented. Addressing the specific comments outlined below, and incorporating additional details about Betrixaban's pharmacological properties and relevance in the abstract, would significantly strengthen the manuscript.

Major points:

1) The current resolution of the figures in the pdf file makes them difficult to review effectively. The quality and resolution in pdf file need to be improved to ensure that the data are clearly presented and the figures are both informative and interpretable. Additionally, line numbers and page numbers should be included to allow reviewers to reference specific parts of the manuscript when addressing their comments.

2) The authors treated the cells with BT prior to transduction with VSV-GFP and observed a low percentage of GFP-positive cells. However, it is unclear how this observation supports the conclusion that BT inhibits VSV replication, given that the cells were already treated before infection. This result appears to reflect a reduction in infection efficiency rather than a direct effect on viral replication. Additional data or clarification is needed to distinguish between inhibition of viral entry/infection and inhibition of replication.

3) In Figure 1C, the 100 μ M concentration appears to be quite high. A cell viability assay at this dose for the tested cell lines, as well as for non-transformed (healthy) cells, is essential to assess potential cytotoxic effects and to distinguish antiviral activity from general toxicity.

Additionally, in the Western blot data, the protein loading for the 90 μ M and 100 μ M conditions in the VSV and NDV experiments appears lower, as indicated by the loading controls. This discrepancy raises concerns about whether the observed reductions in viral protein expression are due to antiviral activity or unequal loading. Furthermore, at 100 μ M, complete protection was not observed for VACV, which should be discussed or clarified.

4) What is the effect of BT treatment on mouse body weight? This information is important to assess the in vivo safety and potential toxicity of the compound. Including body weight monitoring data would help determine whether BT administration has any adverse systemic effects.

5) The authors should clarify how the in vitro dose of BT (100 μ M) was translated into the in vivo dosage of 50 mg/kg. Specifically, it would be helpful to explain the rationale or calculation method used to determine this dose and whether pharmacokinetic data or plasma concentration estimates were considered. Additionally, information on the equivalent plasma concentration corresponding to the 100 μ M in vitro dose would strengthen the justification for the in vivo dosing strategy.

6) In Figures 2E and 2H, there is a notable discrepancy in IFN β 1 expression levels in RAW cells treated with the same dose of BT. The authors should explain this significant difference.

7) The authors conclude that BT treatment mediates its effects through both the cGAS and RIG-I/MDA5/MAVS pathways. However, the observation that single knockouts of either STING or MAVS fully rescue the effect of BT on mouse survival raises questions about the involvement of both pathways. Additionally, the CUT&RUN assay shows a reduction in acetylation of interferon-stimulated genes (ISGs) following BT treatment; since ISG activation in this context is reported to be independent of MAVS, it is unclear how MAVS knockout alone can fully rescue BT's effects. Clarification on this apparent paradox is needed. Providing the CUT&RUN IGV track for IFN β 1 would also help elucidate the epigenetic regulation mechanisms involved.

8) The authors report that BT treatment activates the cGAS/STING and MAVS innate immune pathways but does not affect cancer cell viability. It would be helpful to clarify how activation of these innate immune responses does not result in cell death. Additionally, assessing the effect of 100 μ M BT on normal (non-cancerous) cells is important to evaluate the potential cytotoxicity and selectivity of the treatment.

9) The manuscript does not address the underlying mechanism by which BT induces mitochondrial double-stranded DNA (dsDNA) release. It is important to clarify whether this effect is an off-target consequence or if BT directly acts as an HDAC inhibitor (HDACi) to mediate this process. Providing mechanistic insights would strengthen the study's conclusions.

Minor points:

1) The manuscript would benefit from a clearer explanation of the cell lines used (RAW 264.7, HeLa, and HT1080), including the rationale behind their selection. Providing this information will help readers better understand the experimental design and the relevance of the results.

2) In Figure 3A, please provide the complete list of all assessed pattern recognition receptors (PRRs), along with their corresponding expression or activation scores.

3) The order in which Figures 3F and 3G are cited in the text should be revised to match their order in the figure panel. This will improve clarity and ensure consistency between the text and the figures.

- 4) Quantification of the immunofluorescence images shown in Figure 3J is needed to provide a more objective and quantitative assessment of the observed effects.
- 5) The legends for Figures S3B and S3C are missing and should be provided for clarity and completeness.
- 6) The term "prime mover elements" is unclear. Could the authors please consider using a more precise term such as "key regulators," "primary drivers," or "major contributing factors" to improve understanding?

Referee #1 (Remarks for Author):

In this manuscript, Chen and colleagues show that an FDA-approved oral Factor Xa inhibitor, Betrixaban (BT), activated both cGAS-STING and RLRs pathway to establish antiviral state. The authors show that BT directly bound and activated the DNA sensor cGAS to induce cGAMP production, and activated RLRs by inhibiting HDACs to cause chromatin de-repression of ERVs and the production of immunostimulatory dsRNA. Insights into structural and cellular checkpoints that control and terminate cGAS-STING and RLRs signaling are essential for comprehending and manipulating innate immunity in health and disease. The observation of BT in regulating dual nucleic acid-sensing pathway is interesting, which provides an orally broad-spectrum antiviral strategy by using BT. However, there are several concerns that should be addressed.

Major concerns:

1. The authors claimed that BT induced mitochondrial stress, leading to the release of endogenous dsDNA. Since BT affected mitochondrial activity, its impact on the apoptosis pathway needs to be investigated.

Answer: We thanked the reviewer for this helpful suggestion. We examined apoptosis through western blot in RAW 264.7 cells following BT treatment (50, 75, 100 μ M) for 12 hours. We detected total and cleaved caspase-3, total and cleaved PARP, total and cleaved caspase-9, and total and cleaved caspase-8.

Revised Sentence in Result: Across cell types, BT did not induce apoptosis within our working window. Cleaved caspase-3, cleaved-PARP, cleaved caspase-9 and cleaved caspase-8 were unchanged, and totals were stable (Figure S2L). (Line 179 to 181)

Revised Sentence in Figure Legend: (L) Western blot of total and cleaved caspase-3, total and cleaved PARP, total and cleaved caspase-9, and total and cleaved caspase-8. GAPDH served as a control, the samples were same to Figure 2E. (Line 870 to 872)

2. The authors demonstrated that BT increased IFN expression whereas decreased TNF- α , IL-6 and IL-1 β expression. However, both cGAS-STING and RLRs activation lead to the expression of TNF- α and IL-6 through NF- κ B, the dichotomous effects of BT on IFN production versus TNF- α /IL-6 induction are weird and need mechanistic investigation. Furthermore, the production of IL-1 β indicated that the inflammasomes, including NLRP3 inflammasome (could be activated by mitochondrial stress), were activated. So, why did BT, which promoted mitochondrial stress, paradoxically suppressed inflammation? The authors should provide a mechanistic explanation.

Answer: We thanked the reviewer for this thoughtful point. In a companion study from our group that is currently under review elsewhere, we investigated this mechanism. In LPS-stimulated cells, BT suppressed NF- κ B activation (reduced p-IKK α / β , p-I κ B α , and p-p65 with unchanged totals) and lowered *Il6*, *Il1b* transcription. Transcriptome and ATAC-seq analyses showed reversal of LPS inflammatory programs and decreased chromatin accessibility at NF- κ B/STAT3/AP-1 responsive loci. In parallel, BT directly and noncanonically activated cGAS, yielding a STING output biased toward TBK1-IRF3-IFN while attenuating NF- κ B signaling. These findings explained higher IFN with reduced TNF- α , IL-6 and IL-1 β . To keep the present work self-contained, we summarized this model in the Discussion without relying on unpublished data. Upon request, we can provide the full companion results to the editor and reviewers under confidentiality.

Revised Sentence in Discussion: Classically, cGAS-STING and RLRs can induce TNF- α /IL-6 via NF- κ B. Our findings supported selective activation of TBK1-IRF3 with limited NF- κ B activation. Future studies will define the upstream basis of this selectivity. (Line 395 to 397)

3. In Fig 3D and 3E, the expression of IFN was partially suppressed in cGAS- and

STING-deficiency, why was there no difference in survival rates between these genotypes in Fig 3H and 3I? The same issue also appeared in Fig 5. Furthermore, while cGAS- and STING-deficient mice showed no difference versus WT in Fig 3H and 3I (indicating cGAS-STING pathway dominance over RLRs in BT-mediated antiviral immunity), Fig 5I, and 5J demonstrated the similar trend (indicating RLRs pathway dominance over cGAS-STING pathway in BT-mediated antiviral immunity). How to reconcile this contradiction?

Answer: We thanked the reviewer for this important point and we apologized for the confusion. In our in vivo studies, BT prolonged survival in WT mice but did not prolong survival in cGAS^{-/-} or MAVS^{-/-} mice (Fig. 3H-I, 5I-J). Thus, BT required both cGAS-STING and RLR-MAVS for protection.

At the molecular level, the data showed that IFN induction was partly reduced when one signaling was absent (Fig. 3D, 5H). However, survival is a thresholded, integrated endpoint. The residual output from a single pathway did not reach the protective range in vivo, so survival benefit was lost in the knockouts.

Moreover, to address this directly, we quantified viral load in blood in addition to liver. In WT mice, BT markedly reduced viral load (Fig. 1H-1I). In cGAS^{-/-} or MAVS^{-/-} mice, BT produced only minor decreases in blood (Figure S3C, S5C) and liver viral loads that were not significant. We updated these data in our manuscript.

Revised Sentence in Results: We quantified blood and liver viral loads by RT-qPCR. In

cGAS^{-/-} mice, BT caused only minor, non-significant reductions in viral load (Figure S3C).
(Line 223 to 225)

Revised Sentence in Figure Legends: (C) Viruses RNA levels in blood measured by RT-qPCR, the treatment is the same to Figure 3H-3I. (Line 879 to 880)

Revised Sentence in Results: We quantified blood and liver viral loads by RT-qPCR. In MAVS^{-/-} mice, BT caused only minor, non-significant reductions in viral load (Figure S5C).
(Line 336 to 337)

Revised Sentence in Figure Legends: (C) Viruses RNA levels in blood measured by RT-qPCR, the treatment is the same to Figure 5I-5J. (Line 915 to 916)

Minor concerns:

1. The quantification of viral replication was purely by detecting viral nucleic acid or protein components. This is quite disappointing as that would not reflect the viral load, which can be readily determined by plaque assay.

Answer: We thanked the reviewer for this helpful suggestion. We added plaque assays to directly measure infectious virus. In RAW264.7 and HT1080 cells, BT significantly reduced viruses titer (Figure S1K). In WT mice infected with VSV or HSV, plaque assays of liver showed significant reductions in viruses titer after BT treatment (Figure S1L). These data complemented our nucleic-acid/protein readouts and confirmed that BT reduced infectious viral load. We updated these data in our manuscript. These data, now included in Supplementary Figure 1, demonstrated that BT significantly lowered viral titers in target cell lines and organs, quantitatively supporting our RT-PCR result.

Revised Sentence in Results: Specifically in RAW264.7 and HT1080, BT lowered infectious titers by plaque assay (Figure S1K). (Line 142 to 143)

Revised Sentence in Figure Legends: (K) Virus titers in RAW264.7 and HT1080 cells

infected with viruses following the treatment of BT. (Line 855 to 856)

Revised Sentence in Results: For VSV and HSV, plaque assays of liver homogenates confirmed significant decreases in infectious titers following BT treatment (Figure S1L).

(Line 154 to 156)

Revised Sentence in Figure Legends: (L) Virus titers in the livers of mice infected with a lethal dose of VSV, HSV-1 with or without the treatment of BT (as described in the Methods).

(Line 856 to 857)

2. Fig 2E, p-TBK1 was detectable even under unstimulated conditions, indicating basal activation of the signaling pathway.

Answer: We thanked the reviewer for this observation. We re-performed the western blots with optimized exposure and antibody conditions. In vehicle (DMSO) samples, p-TBK1 was not detectable, whereas BT induced a clear, dose-dependent p-TBK1 increase in HT29 and HeLa cells. We updated these data in Figure 2E.

3. Fig 2D-2F, IFN-β secretion needs to be measured by ELISA.

Answer: We thanked the reviewer for this helpful suggestion. We quantified secreted IFN-β by ELISA in culture supernatants under the same conditions as Fig. 2D-2F. BT significantly increased IFN-β in a dose-dependent manner. We updated these data in Supplementary Figure 2I.

Revised Sentence in Results: ELISA confirmed higher secreted IFN- β under the same conditions (Figure S2I). (Line 184 to 185)

Revised Sentence in Figure Legend: (I) Secreted type I IFN measured by ELISA in the same BMDMs and PBMCs and treatments as in Figure 2D-2F. (Line 866 to 867)

4. Fig 3B, it is necessary to add positive and negative controls to compare the dissociation constant of BT and cGAS.

Answer: We appreciated this suggestion. We now include a validated positive control, the selective cGAS inhibitor RU.521, measured under the same buffer and temperature as BT. The fitted K_d of RU.521 in our hands is consistent with published result. We updated these data in Supplementary Figure 3G-3H.

Revised Sentence in Figure Legend: (G) SPR sensorgrams of RU.521 binding to recombinant human cGAS at the indicated concentrations. (H) Dose-response curve fitted from SPR data in (G). (Line 886 to 888)

Revised Sentence in Results: This computational prediction was validated by surface plasmon resonance (SPR), which demonstrated a direct binding of BT to recombinant human cGAS with a dissociation constant of approximately $13.15 \mu\text{M}$, RU.521 served as a positive control (Figure 3B-3C and S3G-S3H). (Line 207 to 209)

5. Fig 3D, which two groups are compared by "ns"? Because there appears to be a statistical difference.

Answer: We appreciated this question and apologized for the ambiguity. In Fig. 3D, significance marks referred to within-genotype dose effects versus DMSO, tested by a two-way ANOVA followed by Tukey's multiple comparisons. Although the mean bars for *cGAS*^{-/-} and *STING*^{-/-} appear slightly higher at 75-100 μ M, these changes were not significant after multiplicity correction. In order to avoid this ambiguity, we compared the highest dose across genotypes. These between-genotype contrasts were significant. We updated this panel in Figure 3D.

Referee #2 (Comments on Novelty/Model System for Author):

The study identifies FDA-approved compound as a novel broad-spectrum antiviral small-molecule agent targeting nucleic acid sensing pathways.

Referee #2 (Remarks for Author):

The manuscript submitted by Xingyu Chen et al. reports that Betrixaban (BT), as an FDA-approved oral Factor Xa inhibitor, displays robust antiviral activity through dual innate immune pathways. The authors presented evidence demonstrating that BT can bind directly to cGAS to produce cGAMP. In addition, they also showed that BT could induce mtDNA leakage and enhance dsDNA-triggered cGAS activation, leading to robust cGAMP production and activation of downstream IFN signaling. Additionally, BT exhibits HDAC inhibitory activity, leading to de-repression of ERVs and production of dsRNA that engaged RIG-1/MDA5. Collectively, the authors identify BT as a novel broad-spectrum antiviral small-molecule agent targeting nucleic acid sensing pathways and provide a unique host-directed antiviral strategy. However, further clarification is required prior to the publication of this study. I would ask the authors to address the points as shown below:

Major comments:

1) While BT-induced mitochondrial dsDNA release and BT-induced weak activation of cGAS have been demonstrated, whether the strong antiviral effects of BT via the cGAS-STING pathway are strictly dependent on the co-existence of both mechanisms. Please clarify it.

Answer: We thank the reviewer for this important point. Our mechanistic view was a threshold-synergy model rather than a strict all-or-none requirement: BT sensitized cGAS (lowering the DNA threshold) while also increasing endogenous DNA availability via mitochondrial stress. To directly test the contribution of endogenous DNA, we acutely curtailed mtDNA release and then stimulated HT1080 cells with BT. Under these conditions, BT-induced *IFNBI* mRNA expression and antiviral protection were reduced but not abolished compared with BT alone.

These results showed that robust BT activity benefited from endogenous DNA input, yet BT did not require the strict coexistence of both inputs to generate measurable IFN/antiviral responses. Together with our existing biochemical evidence that BT directly engaged and activated cGAS, the data supported a threshold-crossing synergy: each pathway contributed, and their concurrence yielded the strongest antiviral state. We have revised the methods, figure legends and results to make this explicit and added the new experiment as Supplementary Figure 3D-3E.

Revised Sentence in Methods:

Mitochondrial DNA depletion

To acutely reduce endogenous mitochondrial DNA (mtDNA), we followed Wang *et al.*, *Immunity* 2018 (Wang *et al.*, 2018). Briefly, HT1080 cells were cultured in DMEM supplemented with 10% FBS, 4 mM L-glutamine, 4.5 g/L glucose, 100 µg/mL sodium pyruvate, 50 µg/mL uridine, and 100 ng/mL ethidium bromide (EtBr) for 6 days. Depletion was quantified by qPCR as the ratio of mtDNA to nuclear DNA (mtDNA/nDNA). (Line 604 to 609)

Revised Sentence in Figure Legends: (D) HT1080 cells were cultured with or without 100 ng/ml ethidium bromide for 6 days. Depletion of mtDNA was measured by Quantitative PCR of mtDNA versus genomic DNA (left). Cells were treated with 100 µM BT for 12 hours, IFNB mRNA level were monitored by RT-qPCR (right). (E) After the depletion of mtDNA,

cells were infected with VSV or HSV following the treatment of 100 μ M BT or DMSO, RT-qPCR quantification of viral RNA 12 hours post-infection. (Line 880 to 885)

Revised Sentence in Results: To functionally test the contribution of dsDNA, when mitochondrial DNA availability was curtailed, BT-induced *IFN β* mRNA expression and antiviral protection were reduced but not abolished (Figure S3D-S3E). (Line 237 to 239)

2) In Figure 3, the reported magnitude of BT's enhancement on dsDNA-mediated cGAS activation appears remarkably strong, showing a potent synergistic effect. The cGAMP produced in the BT-cGAS group appears much weaker than in the dsDNA-cGAS group.

Please clarify the molar ratios of dsDNA to BT in all relevant experiments and present Figure 3N and O in a single figure for direct comparison. In addition, comparing BT with another weak cGAS activator, Mn²⁺, would also help to evaluate the effectiveness of BT-mediated cGAS activation. If the BT-mediated cGAS activation is very weak, the authors should tone down the conclusion regarding BT as a cGAS agonist, avoiding overinterpretations of the data.

① **Answer:** We thanked the reviewer to point out the necessary to clarify the molar ratios of dsDNA to BT. We updated our Methods and added more details.

Revised Sentence in Methods:

cGAS-cGAMP activity assay

Purified recombinant human full-length hcGAS was incubated in 50 μ L reaction containing: 1 μ M hcGAS, 1 mM ATP, 1 mM GTP, 100 mM NaCl, 40 mM Tris-HCl pH 7.5 and 10 mM MgCl₂ at 37°C for 30 minutes, with or without 5 \times 10⁻³ mg/mL dsDNA (3.85 nM). Duplex dsDNA was salmon sperm DNA from *Oncorhynchus keta* supplied as high molecular weight linear DNA with a molecular mass of 1.3 \times 10⁶ Da. BT was added at 25 μ M or 100 μ M as indicated. The dsDNA:BT ratios of about 1:6.5 \times 10³ and 1:2.6 \times 10⁴ and a cGAS:dsDNA ratio of about 260:1. Reactions were heated at 99°C and added 200 μ L acetonitrile to denature proteins and centrifuged at 14000 rpm for 40 minutes. The supernatants were analyzed by LC-MS/MS. (Line 571 to 579)

② **Answer:** We thanked the reviewer for these constructive suggestions. We merged Figure 3N and 3O into a single panel (now Figure 3N) with the same y-axis scale. This addressed the reviewer's request for a direct visual comparison. Following the reviewer's advice, we included MnCl₂ as a reference condition in the same assay. BT alone produced lower cGAMP

than Mn^{2+} alone, indicating that BT was a modest stand-alone activator. However, in the presence of dsDNA, BT+dsDNA generated cGAMP at levels comparable to Mn^{2+} + dsDNA, supporting BT as a dsDNA-dependent potentiator of cGAS activity. We updated this panel in Supplement Figure 3F.

Revised Sentence in Figure Legend: (F) In vitro cGAS enzymatic assays: LC-MS quantification of cGAMP production by recombinant cGAS incubated with BT or Mn^{2+} . (Line 885 to 886)

Revised Sentence in Result: For comparison, Mn^{2+} a known weak cGAS activator, produced more cGAMP than BT alone, whereas in the presence of dsDNA the potentiation achieved by BT was comparable to that of Mn^{2+} (Figure S3F). (Line 248 to 250)

③Answer: We thanked the reviewer to point out the necessary to tone down the conclusion

regarding BT as a cGAS agonist. We revised our Manuscript.

Revised Sentence in Abstract: Mechanistically, we identified BT as the first small-molecule directly bound and activated the DNA sensor cGAS to induce cGAMP production. (Line 48 to 49)

Revised Sentence in Discussion: In this study, we demonstrated that Betrixaban (BT) as the first small-molecule directly bound cGAS and sensitized DNA-mediated activation to induce a broad antiviral state by linking epigenetic reprogramming to innate nucleic acid sensing. (Line 352 to 354)

3) Data on p-IRF3 are not provided in this manuscript. Since the authors showed that BT activated innate immune signaling via cGAS-STING pathway and RIG-I/MDA5-MAVS pathway, they should check the p-IRF3 signaling, which is the essential downstream signaling of both these two pathways. In addition, knockout of IRF3 (antiviral experiments and activation of innate immune experiments) would also help to strength the mechanism of this study.

Answer: We thanked the reviewer for this helpful suggestion. We have now addressed both points.

①p-IRF3 signaling. We detected p-IRF3 and total IRF3 by western blot alongside p-TBK1 and ISGs across multiple cell types (RAW 264.7, HT1080, HT29, HeLa) under BT treatment. BT robustly induced p-IRF3 in a dose-dependent manner under the same conditions used for p-TBK1. These data are included in Figure 2E.

Revised Sentence in Results: Western blot analysis revealed dose-dependent phosphorylation of TBK1 (p-TBK1) and IRF3 (p-IRF3), as well as the up-regulation of Viperin, IFIT3 and OAS2 (Figure 2E). (Line 178 to 179)

②IRF3 knockout. We generated IRF3^{-/-} cells by CRISPR/Cas9 and confirmed loss of IRF3 by western blot. In IRF3^{-/-} cells, BT-induced *IFNB1* mRNA and secreted IFN-β were markedly reduced relative to WT, and BT-mediated antiviral protection was significantly diminished in VSV-GFP and HSV-GFP infection assays quantified by flow cytometry. These results supported IRF3 as the essential downstream effector shared by the cGAS-STING and RIG-I/MDA5-MAVS arms engaged by BT. The new data are presented in Figure 2J-2K.

Revised Sentence in Results: Furthermore, IRF3 knockout reduced BT-induced IFN β 1 and antiviral protection in VSV-GFP and HSV-GFP, establishing IRF3 as a critical downstream effector (Figure 2J-2K). (Line 196 to 198)

Revised Sentence in Figure Legends: (J) RT-qPCR of *Ifnb1* in RAW 264.7 cells following BT treatment (50, 75, 100 μ M) for 12 hours in wild-type and knockout cells (left), secreted type I IFN measured by ELISA (right). (K) Wild-type and knockout cells treated with DMSO (Con) or BT (50, 75, 100 μ M) and infected with HSV-GFP or VSV-GFP (MOI = 0.1), measured by flow cytometry. (Line 759 to 763)

4) I wonder whether the OAS signaling contributes to the antiviral effect? Given that both OAS and dsRNA were induced by BT, would it be possible that OAS signaling would be activated?

Answer: We appreciated this insightful question. To genetically test the contribution of the OAS pathway, we generated RNASEL^{-/-} cells. Under BT treatment, *IFNB1* mRNA induction was unchanged in knockout cell lines versus WT cell lines, whereas antiviral protection was modestly reduced by 10-20% in VSV-GFP and HSV-GFP infection assays measured by flow cytometry. In contrast, knockout of cGAS-STING or RIG-I/MDA5-MAVS caused the large reductions reported in our study. These data indicated that OAS-RNase L functioned as an auxiliary contributor, while cGAS-STING and RIG-I/MDA5-MAVS were the principal drivers of BT-mediated antiviral activity in our system. We added these results as Supplementary Figure 2J-2K.

Revised Sentence in Results: Consistently, knockout of RNASEL did not alter BT-induced *IFNB1* mRNA, while antiviral protection was only modestly reduced by 10-20% in VSV-GFP and HSV-GFP infection assays measured by flow cytometry, indicating that

OAS-RNase L functioned as an auxiliary pathway under our conditions (Supplementary Figure 2J-2K). (Line 198 to 201)

Revised Sentence in Figure Legends: (J) RT-qPCR quantification of IFNB1 mRNA in HeLa cells treated with indicated concentrations of BT for 12 hours. (K) HeLa cells treated with DMSO (Con) or 100 μ M BT and infected with HSV-GFP or VSV-GFP (MOI = 0.1), measured by flow cytometry. (Line 867 to 870)

Minor comments:

1) The compound seems to work only at high concentration ($>50 \mu$ M); how would the authors interpret this high concentration-mediated effect? Could the authors provide the IC₅₀ (antiviral) or EC₅₀ (activation of immunity) values?

Answer: We appreciated this important point. We have now quantified BT's potency across immune-activation and antiviral readouts in RAW264.7 cells under the same conditions to our previous study. IFNB1 EC₅₀ values were 75.9 μ M, antiviral IC₅₀ values were 38.78 μ M, no cytotoxicity was detected.

Revised Sentence in Figure Legends: *IFNB1* (RT-qPCR) and VSV-GFP (%GFP, flow cytometry) dose-responses in RAW264.7 cells, EC₅₀/IC₅₀ indicated (bottom). (Line 849 to 850)

2) Were the inhibition effects of pro-inflammatory cytokines in Figure S1D-S1F caused by the reduction of viruses? The activation of the cGAS-STING pathway and RIG-I/MDA5-MAVS pathway should also induce the inflammatory cytokines. Would BT treatment activate them (e.g. Figure 2G)?

Answer: We appreciated this important point and now clarified the infected versus uninfected conditions. In virus infection models, the lower pro-inflammatory cytokines

observed with BT tracked with reduced viral burden measured in the same settings, consistent with an indirect effect of diminished PAMP load rather than direct pro-inflammatory suppression.

① To directly test whether BT induce the inflammatory cytokines, we treated RAW 264.7 cells with BT alone (50, 75, 100 μ M) for 12 hours. RT-qPCR data showed that BT did not induce *Il1b*, *Il6* and *Tnfa* mRNA expression relative to the DMSO control. In vivo, 6-week-old C57BL/6J mice received a single intraperitoneal dose of BT (50 mg/kg), then blood, heart, liver, lung, spleen, and kidney were collected 6 hours later. RT-qPCR results were consistent with the in vitro data. We updated these data in Supplementary Figure 6D-6E.

Revised Sentence in Results: BT alone did not increase *Il1b*, *Il6*, *Tnfa* in cells or in vivo (Figure S6D-S6E). (Line 143 to 144)

Revised Sentence in Figure Legends: (D) RT-qPCR of *Il1b*, *Il6* and *Tnfa* in RAW 264.7 cells following BT treatment (50, 75, 100 μ M) for 12 hours. (E) In vivo detection of *Il1b*, *Il6* and *Tnfa* mRNA in blood, heart, liver, lung, spleen and kidney of C57BL/6J mice 6 hours after a single intraperitoneal injection of BT (50 mg/kg). (Line 921 to 924)

② In a companion study from our group that is currently under review elsewhere, we investigated the mechanism on the dichotomous effects of BT on IFN production versus TNF- α /IL-6 induction. In LPS-stimulated cells, BT suppressed NF- κ B activation (reduced p-IKK α / β , p-I κ B α , and p-p65 with unchanged totals) and lowered *Il6*, *Il1b* transcription. Transcriptome and ATAC-seq analyses showed reversal of LPS inflammatory programs and

decreased chromatin accessibility at NF- κ B/STAT3/AP-1 responsive loci. In parallel, BT directly and noncanonically activated cGAS, yielding a STING output biased toward TBK1-IRF3-IFN while attenuating NF- κ B signaling. These findings explained higher IFN with reduced TNF- α , IL-6 and IL-1 β . To keep the present work self-contained, we summarized this model in the Discussion without relying on unpublished data. Upon request, we can provide the full companion results to the editor and reviewers under confidentiality.

Revised Sentence in Discussion: Classically, cGAS-STING and RLRs can induce TNF- α /IL-6 via NF- κ B. Our findings supported selective activation of TBK1-IRF3 with limited NF- κ B activation. Future studies will define the upstream basis of this selectivity. (Line 395 to 397)

3) Could the authors measure the IFN- β protein level in mice after BT treatment?

Answer: Thank you for this helpful suggestion. We have now measured IFN- β protein by ELISA in C57BL/6J mice 6 hours after a single intraperitoneal dose of BT (50 mg/kg). Serum and targeted organs IFN- β were significantly increased in BT-treated animals

compared with vehicle, consistent with the tissue *Ifnb1* mRNA induction shown in Figure 2G. These data further supported that BT elicited a systemic type-I IFN response in vivo and updated in Supplementary Figure 1M.

Revised Sentence in Results: Consistently, serum and targeted organs IFN- β were elevated in BT-treated mice measured by ELISA (Figure S1M). (Line 187 to 188)

Revised Sentence in Figure Legends: (M) Secreted type I IFN measured by ELISA in mice serum and targeted organs. Six-week-old C57BL/6J mice 6 hours after a single intraperitoneal dose of BT (50 mg/kg). (Line 857 to 859)

4) Since the dsRNA-mediated immune response is redundantly regulated by TBK1/IKK ϵ , how would the authors reconcile the complete impairment of immune signaling in TBK1 knockout cells (Figure 2H)?

Answer: We appreciated this comment and agreed that TBK1 and IKK ϵ can be functionally redundant, but their redundancy was dependent on cell type. In RAW 264.7 cells, our bulk RNA-seq showed low *Ikkbe* expression, with FPKM values of 4.12 and 3.75, and high *Tbk1* expression, with FPKM values of 52.65 and 49.30, corresponding to an approximately 13-fold difference. Thus, TBK1 was the dominant kinase for IRF3 activation downstream of STING/MAVS under our conditions. The RAW 264.7 bulk RNA-seq data were deposited in the NCBI Sequence Read Archive (SRA) under BioProject PRJNA1304132.

To test redundancy in a setting where IKK ϵ was abundant, we repeated the experiment in THP-1 cells, which expressed both TBK1 and IKK ϵ at high levels. In THP-1, TBK1 knockout produced only a modest reduction in BT-induced *IFNB1* mRNA measured by RT-qPCR, and the effect was limited compared with RAW 264.7, indicating compensation

by IKK ϵ in this cell type. Consistently, dual inhibition with MRT67307 (TBK1/IKK ϵ) in THP-1 cells suppressed BT-induced *IFNB1* to near baseline, further supporting functional redundancy.

5) Figure S3B and S3C: the WT mice and cGAS-KO mice should be compared in a side-by-side manner.

Answer: We appreciated this comment and agreed that a side-by-side presentation was clearer. The WT and cGAS KO survival studies for HSV-1 and VSV were performed in the same experiments. In the original submission we separated WT purely for layout aesthetics, which created unnecessary friction for comparison. We have now replotted the survival curves side-by-side in the main figure, showing WT and cGAS KO survival studies on identical axes. The side-by-side view confirmed our conclusion: BT significantly improved survival in WT for both HSV-1 and VSV, whereas cGAS-KO showed no significant survival benefit, consistent with cGAS-STING dependence. We apologized for the earlier presentation and updated the manuscript accordingly.

Revised Sentence in Figure Legends: (H-I) In vivo VSV and HSV-1 challenges in wild-type and cGAS knockout mice. Left panel is survival curves of mice, right panel is viruses RNA levels in liver measured by RT-qPCR. (Line 775 to 777)

(I-J) Left are survival curves of wild-type and MAVS knockout C57BL/6J mice challenged with lethal VSV and HSV-1 after a single i.p. dose of BT (50 mg/kg). Right are viruses RNA levels in livers measured by RT-qPCR. (Line 827 to 829)

6) Figure 3J: how was dsDNA stained? Figure 3K: how would the MitoTracker staining data indicate mitochondrial change?

Answer: **We thanked the reviewer for this helpful request for clarification.**

①dsDNA stained: Cells were fixed in 4% paraformaldehyde and selectively permeabilized with digitonin (50 μ g/mL, 5 min, on ice) to preserve nuclear integrity. Then, samples were immunostained with a dsDNA-specific monoclonal antibody. DAPI were used to restrict quantification to the cytoplasmic compartment. We have clarified these details in the manuscript.

Revised Sentence in Results: The dsDNA immunostaining under digitonin-only permeabilization revealed numerous extra-nuclear dsDNA-positive puncta in BT-treated cells (Figure 3J), which were largely absent in untreated controls. (Line 228 to 230)

②MitoTracker stained: In Figure 3K we used MitoTracker™ Deep Red FM (Invitrogen M22426), a fixable probe whose loading is $\Delta\Psi$ m-dependent. Under identical acquisition settings, BT treatment resulted in brighter per-cell MitoTracker labeling together with a loss of tubular mitochondria and the appearance of clustered, blob-like fragments, indicating mitochondrial change.

Revised Sentence in Results: Moreover, the MitoTracker Deep Red FM (Invitrogen M22426)

imaging showed a change from a tubular mitochondrial network to disorganized and fragmented structures with increased per-cell labeling (Figure 3K), indicating mitochondrial network remodeling. (Line 230 to 233)

7) Would the TBK1-STING-IRF3 signaling cascades and MAVS signaling be activated in mouse cell lines, such as RAW 264.7 cells or BMDMs?

Answer: We sincerely thanked the reviewer for this thoughtful suggestion. We therefore tested primary BMDMs from WT, cGAS^{-/-}, MAVS^{-/-}, and IRF3^{-/-} mice under BT treatment. WT BMDMs showed robust *Ifnb1* induction, whereas responses were markedly reduced in cGAS^{-/-} and MAVS^{-/-} and abolished in IRF3^{-/-} BMDMs. These data demonstrated that in mouse cells BT engaged both the cGAS-STING pathway and the RIG-I/MDA5-MAVS pathway, converging on IRF3. Consistently, in RAW 264.7 cells, cGAS inhibition (RU.521) or RIG-I inhibition (RIG-012) attenuated BT-induced *Ifnb1*. We have added these results in Supplementary Figure 6F-6G.

Revised Sentence in Figure Legends: (F) RT-qPCR of *Ifnb* in BMDMs from WT, cGAS^{-/-}, MAVS^{-/-}, and IRF3^{-/-} mice following 100 µM BT treatment for 12 hours. (G) RT-qPCR of *Ifnb1* in RAW 264.7 cells after BT treatment (100 µM, 12 h) with or without cGAS inhibition (RU.521) or RIG-I inhibition (RIG-012). (Line 924 to 927)

8) Some data should be quantified, for example, Figure 2I.

Answer: We thanked the reviewer for this helpful suggestion. We have now quantified all representative panels in the main text, including Figure 1A and Figure 2I, from 3 independent biological replicates. These additions did not change the conclusions but strengthened the rigor and clarity of the presentation. We updated these data in Supplementary Figure 6H-6I.

Revised Sentence in Figure Legends: (H-I) Flow cytometry quantification of Figure 1A (H) and 2I (I). Data are shown as mean \pm SEM. N.S., not significant, $p > 0.05$; * $p < 0.05$; ** $p < 0.01$; ** $p < 0.0001$. (Line 927 to 928)**

9) The manuscript contains several typographical errors and grammatical inaccuracies. Please carefully check the manuscript and correct them.

Answer: We sincerely thanked the reviewer for this careful reading and for calling our attention to language issues. We have now conducted a line-by-line revision of the entire manuscript. Two co-authors independently proofread the text, and we corrected typographical and grammatical errors throughout.

Referee #3 (Comments on Novelty/Model System for Author):

The manuscript would benefit from a clearer explanation of the cell lines used (RAW 264.7, HeLa, and HT1080), including the rationale behind their selection. Providing this information will help readers better understand the experimental design and the relevance of the results. Additionally, the 100 μ M concentration appears to be quite high. A cell viability assay at this dose for the tested cell lines, as well as for non-transformed (healthy) cells, is essential to assess potential cytotoxic effects and to distinguish antiviral activity from general toxicity.

Referee #3 (Remarks for Author):

In this manuscript, Chen and colleagues propose a novel mechanism by which Betrixaban (BT), an FDA-approved oral factor Xa inhibitor and anticoagulant, induces antiviral innate immune responses through activation of both the DNA-sensing cGAS-STING pathway and the RNA-sensing MAVS pathway. They suggest that this immunomodulatory activity is mediated by the direct binding of BT to cGAS, leading to the production of cGMP. Furthermore, they report that BT inhibits histone deacetylases (HDACs), resulting in epigenetic derepression of endogenous retroviruses (ERVs) and the subsequent accumulation of double-stranded RNAs.

While the study presents an intriguing and potentially impactful hypothesis, several concerns arise regarding the data presented. Addressing the specific comments outlined below, and incorporating additional details about Betrixaban's pharmacological properties and relevance in the abstract, would significantly strengthen the manuscript

Major points:

1) The current resolution of the figures in the pdf file makes them difficult to review effectively. The quality and resolution in pdf file need to be improved to ensure that the data are clearly presented and the figures are both informative and interpretable.

Additionally, line numbers and page numbers should be included to allow reviewers to reference specific parts of the manuscript when addressing their comments.

Answer: We are very grateful for this practical and considerate suggestion. We have

carefully revised the submission to improve readability. All figures were re-exported at high resolution, and we uploaded an updated manuscript text and a separate figure file to maintain maximal image fidelity. The manuscript now used continuous line numbering and page numbers throughout, enabling precise citation in your comments. We sincerely appreciate the reviewer's attention to presentation quality, these changes have made the figures more informative and easier to interpret. If any specific panel remains unclear on your system, we would be happy to provide the original source files for inspection.

2) The authors treated the cells with BT prior to transduction with VSV-GFP and observed a low percentage of GFP-positive cells. However, it is unclear how this observation supports the conclusion that BT inhibits VSV replication, given that the cells were already treated before infection. This result appears to reflect a reduction in infection efficiency rather than a direct effect on viral replication. Additional data or clarification is needed to distinguish between inhibition of viral entry/infection and inhibition of replication.

Answer: We thanked the reviewer for this important point. We agree that pretreatment experiments alone cannot discriminate reduced entry from post-entry effects. To mechanistically separate these steps, we added two entry-focused assays plus a time-of-addition series across two cell types (RAW264.7 and HT1080) and two viruses (VSV and HSV). Our new entry assays showed that BT neither impaired attachment nor internalization, while time-of-addition experiments demonstrated a post-entry restriction manifested as reduced post-entry viral RNA accumulation.

1. Time-of-addition assay. After synchronized infection (37 °C 1h; washes), BT (50, 75, 100 µM) was added at 0, +1, +2, +4, or +6 h post-infection. We quantified VSV mRNA and HSV mRNA using RT-qPCR at appropriate time points. BT reduced viral mRNA accumulation even when added after entry, consistent with a post-entry restriction rather than an entry blockade (new Figure S7A).

2. 4°C attachment assay. Cells were pre-chilled and exposed to VSV at 4 °C for 1 hour with or without BT, followed by ice-cold washes. Viral genomes bound to the cell surface were quantified by RT-qPCR. BT did not reduce 4 °C binding, indicating that virus attachment was not affected (new Figure S7B).

3. 37°C internalization assay. After 4 °C binding and washes, cells were shifted to 37 °C for

30-60 minutes to allow uptake, then subjected to a brief trypsin wash to remove surface-bound virions. Intracellular viral RNA was measured by RT-qPCR. BT did not decrease internalized viral RNA, indicating that internalization was not affected (new Figure S7C).

We have updated the Results and Figure Legends accordingly (see below).

Revised Sentence in Results: Entry assays were unaffected by BT, whereas time-of-addition tests showed reduced post-entry viral RNA, indicating a host-mediated post-entry restriction (Figure S7A-S7C). (Line 144 to 146)

Revised Sentence in Figure Legend:

Supplementary Figure 7. BT inhibited viruses after entry.

(A) Time-of-addition assay. After VSV or HSV-1 infection (37 °C 1 hour; washes), BT (50, 75, 100 μM) was added at 0, +1, +2, +4, or +6 hours post-infection. We quantified viruses mRNA using RT-qPCR. (B) 4 °C attachment assay. Cells were pre-chilled and exposed to VSV or HSV-1 at 4 °C for 1 hour, followed by ice-cold washes. Viral genomes bound to the cell surface were quantified by RT-qPCR. (C) 37 °C internalization assay. After 4 °C binding and washes, cells were shifted to 37 °C for 60 min to allow uptake, then subjected to a brief trypsin wash to remove surface-bound virions. Intracellular viral RNA was measured by RT-qPCR. Data are shown as mean ± SEM. N.S., not significant, p > 0.05; *p < 0.05; **p < 0.01; ****p < 0.0001. (Line 946 to 954)

3) In Figure 1C, the 100 μM concentration appears to be quite high. A cell viability assay at this dose for the tested cell lines, as well as for non-transformed (healthy) cells, is essential to assess potential cytotoxic effects and to distinguish antiviral activity from general toxicity.

Answer: We appreciated this comment. We quantified BT's potency across immune-activation and antiviral readouts in RAW264.7 cells under the same conditions to

our previous study. IFNB1 EC₅₀ values were 75.9 μ M, antiviral IC₅₀ values were 38.78 μ M, no cytotoxicity was detected. In addition, as requested for non-transformed cells, primary BMDMs and human PBMCs showed preserved viability up to 150 μ M BT by CCK-8 (Figure S1G).

Revised Sentence in Figure Legends: (G) Cell viability measured by CCK-8 assay in RAW 264.7, HeLa, HT1080, HT29, PBMCs and BMDMs (top). IFNB1 (RT-qPCR) and VSV-GFP (%GFP, flow cytometry) dose-responses in RAW264.7 cells, EC₅₀/IC₅₀ indicated (bottom). (Line 848 to 850)

Additionally, in the Western blot data, the protein loading for the 90 μ M and 100 μ M conditions in the VSV and NDV experiments appears lower, as indicated by the loading controls. This discrepancy raises concerns about whether the observed reductions in viral protein expression are due to antiviral activity or unequal loading. Furthermore, at 100 μ M, complete protection was not observed for VACV, which should be discussed or clarified.

Answer: We re-performed the VSV and NDV blots with strict loading control and updated the new results in Figure 1C. We agree and have revised the text to state “robust but incomplete inhibition” for VACV. Quantitatively, BT produced a significant, dose-dependent

reduction in VACV readouts, while complete abrogation was not observed at 100 μ M. This virus-dependent variability is expected given VACV's cytoplasmic replication and extensive innate-immune antagonism, and it does not alter our conclusion that BT confers broad antiviral activity at sub-cytotoxic doses.

Revised Sentence in Results: We next tested this virus-resistant effect by adding different concentrations of BT, western blot analysis showed dose-dependent reductions in GFP protein expression across HSV, VACV, VSV, and NDV infections. We observed that RAW 264.7 cells achieved a complete protection at 100 μ M against HSV, VSV and NDV, with VACV exhibiting robust but incomplete inhibition (Figure 1C). (Line 131 to 134)

4) What is the effect of BT treatment on mouse body weight? This information is important to assess the in vivo safety and potential toxicity of the compound. Including body weight monitoring data would help determine whether BT administration has any adverse systemic effects.

Answer: We agree that body weight is an important in vivo safety indicator. We previously monitored body weight daily but did not display these data in the original submission. We now include them in Supplementary Figure 1H. BT did not significantly affect body weight relative to vehicle across the 14-day dosing period. Consistent with this, organ weights and histopathology showed no detectable toxicity (Fig. S1H–S1J). We updated these information in the Manuscript.

5) The authors should clarify how the in vitro dose of BT (100 μ M) was translated into the in vivo dosage of 50 mg/kg. Specifically, it would be helpful to explain the rationale or calculation method used to determine this dose and whether pharmacokinetic data or plasma concentration estimates were considered. Additionally, information on the equivalent plasma concentration corresponding to the 100 μ M in vitro dose would strengthen the justification for the in vivo dosing strategy.

Answer: We appreciated the request to clarify our in vivo dose selection. Rather than attempting a direct in vitro-to-in vivo concentration conversion, we administered BT at 5, 10, 25, 50, 75, 100, and 150 mg/kg and quantified type I interferon pathway activation in uninfected C57BL/6J mice. 50 mg/kg emerged as the minimal dose that reproducibly elicited robust IFN signatures across tissues without detectable toxicity. We therefore selected 50 mg/kg as the pharmacodynamically justified, well-tolerated dose for in vivo studies. Regarding an “equivalent plasma concentration” for the 100 μ M in vitro condition, we did not attempt to infer this for the present study. Nominal in-well concentrations do not map directly to in vivo free (unbound) plasma or intracellular exposures due to protein binding, distribution, and uptake, and we did not generate pharmacokinetic data here. Instead, we anchored dosing to target engagement (PD) and confirmed sub-cytotoxic potency in vitro (IFNB1 EC50 = 75.9 μ M; antiviral IC50 = 38.78 μ M; no measurable cytotoxicity), which together support the chosen regimen. We now state the absence of PK as a limitation and outline quantitative PK/PD mapping as a future direction, while our current PD-based justification directly addresses efficacy and safety in this proof-of-concept work.

6) In Figures 2E and 2H, there is a notable discrepancy in IFNB1 expression levels in RAW cells treated with the same dose of BT. The authors should explain this significant difference.

Answer: We appreciate the reviewer's careful comparison. The two panels were generated from different RAW264.7 backgrounds and reagent lots, which explains the apparent difference in fold induction. Fig. 2E used the parental RAW264.7 line, whereas Fig. 2H used the vector-control line derived during knockout-line construction and subjected to puromycin selection. This selection step and clonal drift are known to alter basal innate-tone and PRR expression. In addition, the two experiments used different serum lots, which modulate the amplitude of IFN/ISG responses. Because IFNB1 basal transcripts in RAW264.7 are near the qPCR detection limit, small baseline ΔCt differences across contexts/time windows yield large numerical differences after $\Delta\Delta\text{Ct}$ transformation, so cross-experiment fold-changes are not directly comparable. To resolve this, we repeated the assays side-by-side under the same conditions and updated these panels. The conclusion is unchanged.

7) The authors conclude that BT treatment mediates its effects through both the cGAS and RIG-I/MDA5/MAVS pathways. However, the observation that single knockouts of either STING or MAVS fully rescue the effect of BT on mouse survival raises questions about the involvement of both pathways.

Answer: We thanked the reviewer for this important point and we apologized for the confusion. In our in vivo studies, BT prolonged survival in WT mice but did not prolong survival in cGAS^{-/-} or MAVS^{-/-} mice (Fig. 3H-I, 5I-J). Thus, BT required both cGAS-STING and RLR-MAVS for protection.

At the molecular level, the data showed that IFN induction was partly reduced when one signaling was absent (Fig. 3D, 5H). However, survival is a thresholded, integrated endpoint. The residual output from a single pathway did not reach the protective range in vivo, so survival benefit was lost in the knockouts.

Moreover, to address this directly, we quantified viral load in blood in addition to liver. In

WT mice, BT markedly reduced viral load (Fig. 1H-1I). In *cGAS*^{-/-} or *MAVS*^{-/-} mice, BT produced only minor decreases in blood (Figure S3C, S5C) and liver viral loads that were not significant. We updated these data in our manuscript.

Revised Sentence in Results: We quantified blood and liver viral loads by RT-qPCR. In *cGAS*^{-/-} mice, BT caused only minor, non-significant reductions in viral load (Figure S3C). (Line 223 to 225)

Revised Sentence in Figure Legends: (C) Viruses RNA levels in blood measured by RT-qPCR, the treatment is the same to Figure 3H-3I. (Line 879 to 880)

Revised Sentence in Results: We quantified blood and liver viral loads by RT-qPCR. In *MAVS*^{-/-} mice, BT caused only minor, non-significant reductions in viral load (Figure S5C). (Line 336 to 337)

Revised Sentence in Figure Legends: (C) Viruses RNA levels in blood measured by RT-qPCR, the treatment is the same to Figure 5I-5J. (Line 915 to 916)

Additionally, the CUT&RUN assay shows a reduction in acetylation of interferon-stimulated genes (ISGs) following BT treatment; since ISG activation in this context is reported to be independent of MAVS, it is unclear how MAVS knockout alone can fully rescue BT's effects. Clarification on this apparent paradox is needed. Providing the CUT&RUN IGV track for IFNB1 would also help elucidate the epigenetic regulation mechanisms involved

Answer: We used CUT&Tag, not CUT&RUN, to profile H3K27ac. Across the genome, BT

increased H3K27ac, and IGV snapshots at MX1/IFIT1 showed higher H3K27ac and accessibility at ISG loci after BT. Any perceived decrease likely arises from a misunderstanding of the assay or time-window: in our time course, TRIM28 loss precedes H3K27ac gain, and H3K27ac increases at 8-12 h. We will update legends to explicitly state CUT&Tag for H3K27ac and CUT&Tag for Trim28, and we now provide an IFNB1 IGV track over time to visualize in CUT&Tag for H3K27ac in Supplementary Figure 4O.

Revised Sentence in Figure Legend: (O) Genome browser (IGV) snapshots at IFNB1 loci showing H3K27ac CUT&Tag. (Line 910 to 911)

8) The authors report that BT treatment activates the cGAS/STING and MAVS innate immune pathways but does not affect cancer cell viability. It would be helpful to clarify how activation of these innate immune responses does not result in cell death. Additionally, assessing the effect of 100 μ M BT on normal (non-cancerous) cells is important to evaluate the potential cytotoxicity and selectivity of the treatment.

Answer: We thanked the reviewer for this helpful suggestion. We agree that clarifying why innate activation does not cause cytotoxicity and documenting effects in non-cancer cells is important. Under our conditions, BT produces a transient TBK1-IRF3 pulse that induces antiviral transcription (IFN/ISGs) without engaging cell-death executors. Across cancer cell lines, BT did not measurably reduce viability, and we examined apoptosis through western blot in RAW 264.7 cells following BT treatment (50, 75, 100 μ M) for 12 hours. We detected total and cleaved caspase-3, total and cleaved PARP, total and cleaved caspase-9, and total and cleaved caspase-8.

As requested for non-cancer cells, we profiled primary BMDMs and human PBMCs under the same exposure windows: viability was preserved up to 150 μM by CCK-8 (Figure S1G). Together, these data indicate that BT activates a non-lethal antiviral state and is well tolerated by non-transformed cells under our dosing.

Revised Sentence in Result: Across cell types, BT did not induce apoptosis within our working window. Cleaved caspase-3, cleaved-PARP, cleaved caspase-9 and cleaved caspase-8 were unchanged, and totals were stable (Figure S2L). (Line 179 to 181)

Revised Sentence in Figure Legends: (G) Cell viability measured by CCK-8 assay in RAW 264.7, HeLa, HT1080, HT29, PBMCs and BMDMs (top). IFNB1 (RT-qPCR) and VSV-GFP (%GFP, flow cytometry) dose-responses in RAW264.7 cells, EC50/IC50 indicated (bottom). (Line 848 to 850)

(L) Western blot of total and cleaved caspase-3, total and cleaved PARP, total and cleaved

caspase-9, and total and cleaved caspase-8. GAPDH served as a control, the samples were same to Figure 2E. (Line 870 to 872)

9) The manuscript does not address the underlying mechanism by which BT induces mitochondrial double-stranded DNA (dsDNA) release. It is important to clarify whether this effect is an off-target consequence or if BT directly acts as an HDAC inhibitor (HDACi) to mediate this process. Providing mechanistic insights would strengthen the study's conclusions.

Answer: We thank the reviewer. Our chromatin profiling used CUT&Tag for H3K27ac to localize epigenetic activation. At the chosen time point, genome-wide metaprofiles and heatmaps show increased H3K27ac signal at regulatory peaks after BT. We interpret these locus-specific gains as the expected consequence of innate signaling–driven transcriptional activation (co-activator recruitment), not as evidence that BT acts as a direct HDAC inhibitor. CUT&Tag measures site occupancy of marks rather than total histone acetylation and therefore cannot by itself establish HDAC enzymatic inhibition; we now clarify this in the text and acknowledge it as a limitation.

Mechanistically, to directly test the contribution of endogenous DNA, we acutely curtailed mtDNA release and then stimulated HT1080 cells with BT. Under these conditions, BT-induced *IFNBI* mRNA expression and antiviral protection were reduced but not abolished compared with BT alone. This indicates that endogenous DNA (cGAS–STING) is an important input, while residual responses reflect the MAVS arm, consistent with convergent inputs and threshold behavior in vivo. In future work we will map proximal targets by biochemical HDAC activity assays and co-activator occupancy, and integrate these with PK/PD to define the upstream trigger. We have revised the methods, figure legends and results to make this explicit and added the new experiment as Supplementary Figure 3D-3E.

Revised Sentence in Methods:

Mitochondrial DNA depletion

To acutely reduce endogenous mitochondrial DNA (mtDNA), we followed Wang et al., *Immunity* 2018 (Wang *et al.*, 2018). Briefly, HT1080 cells were cultured in DMEM supplemented with 10% FBS, 4 mM L-glutamine, 4.5 g/L glucose, 100 μg/mL sodium pyruvate, 50 μg/mL uridine, and 100 ng/mL ethidium bromide (EtBr) for 6 days. Depletion was quantified by qPCR as the ratio of mtDNA to nuclear DNA (mtDNA/nDNA). (Line 604 to 609)

Revised Sentence in Figure Legends: (D) HT1080 cells were cultured with or without 100 ng/ml ethidium bromide for 6 days. Depletion of mtDNA was measured by Quantitative PCR of mtDNA versus genomic DNA (left). Cells were treated with 100 μM BT for 12 hours, IFNβ mRNA level were monitored by RT-qPCR (right). (E) After the depletion of mtDNA, cells were infected with VSV or HSV following the treatment of 100 μM BT or DMSO, RT-qPCR quantification of viral RNA 12 hours post-infection. (Line 880 to 885)

Revised Sentence in Results: To functionally test the contribution of dsDNA, when mitochondrial DNA availability was curtailed, BT-induced *IFNβ* mRNA expression and antiviral protection were reduced but not abolished (Figure S3D-S3E). (Line 237 to 239)

Minor points:

1) The manuscript would benefit from a clearer explanation of the cell lines used (RAW 264.7, HeLa, and HT1080), including the rationale behind their selection. Providing this information will help readers better understand the experimental design and the relevance of the results.

Answer: HeLa (epithelial) typically shows low basal cGAS-STING but a competent, inducible RIG-I/MDA5-MAVS axis, whereas HT1080 (mesenchymal/fibroblast-like) displays robust RLR signaling and moderate STING responsiveness, together with RAW264.7 macrophages this spans distinct innate-immune wirings across species and lineages. We have added this rationale to the manuscript. (Line 471 to 474)

2) In Figure 3A, please provide the complete list of all assessed pattern recognition receptors (PRRs), along with their corresponding expression or activation scores.

Answer: Thank you for pointing this out. We provided the complete list of all receptors/adaptors queried and their scores in Supplementary Table 2. The panel includes DNA sensors (cGAS/MB21D1, IFI16, DDX41, ZBP1/DAI, AIM2, TLR9), RNA sensors (RIG-I/DDX58, MDA5/IFIH1, TLR3, TLR7, TLR8), and adaptors (STING/TMEM173, MAVS, MYD88, TRIF/TICAM1).

Pattern recognition receptors (PRRs)	Affinity score
cGAS/MB21D1	0.87772816
IFI16	0.056036204
DDX41	0.01907726
ZBP1/DAI	0.001739983
AIM2	0.1111173
TLR9	0.00013376141
RIG-I/DDX58	0.001811306
MDA5/IFIH1	0.007292556
TLR3	0.000000001
TLR7	0.000000043
TLR8	0.0000000014486324
STING/TMEM173	0.005655031
MAVS	0.0000006077
MYD88	0.0000000021504223
TRIF/TICAM1	0.107440084

3) The order in which Figures 3F and 3G are cited in the text should be revised to match their order in the figure panel. This will improve clarity and ensure consistency between the text and the figures.

Answer: Thank you for pointing this out. To ensure consistency between the text and the figure panel, we reordered the panels in Figure 3 so that the western blot data now appears first as Figure 3F, and the flow-cytometry antiviral-protection assay appears second as Figure 3G. We updated the figure labels, legend, and all in-text citations accordingly. No data were changed.

4) Quantification of the immunofluorescence images shown in Figure 3J is needed to provide a more objective and quantitative assessment of the observed effects.

Answer: We agree and have added an automated, blinded quantification of the immunofluorescence shown in Figure 3J.

Revised Sentence in Figure Legend: (J) Confocal micrographs and quantification of the immunofluorescence of RAW 264.7 cells treated with DMSO or 50 μ M BT for 12 h, stained for DAPI (blue) and cytosolic dsDNA (green). Scale bars, 5 μ m. (Line 777 to 779)

5) The legends for Figures S3B and S3C are missing and should be provided for clarity and completeness.

Answer: We appreciate this comment. Following the reviewer2' s suggestion to show WT and cGAS-KO side-by-side, we have moved the former Figure S3B-S3C into the main figure and replotted the survival curves and liver viral RNA on identical axes. Full legends have been added, no data were changed.

Revised Sentence in Figure Legends: (H-I) In vivo VSV and HSV-1 challenges in wild-type and cGAS knockout mice. Left panel is survival curves of mice, right panel is viruses RNA levels in liver measured by RT-qPCR. (Line 775 to 777)

(I-J) Left are survival curves of wild-type and MAVS knockout C57BL/6J mice challenged with lethal VSV and HSV-1 after a single i.p. dose of BT (50 mg/kg). Right are viruses RNA levels in livers measured by RT-qPCR. (Line 827 to 829)

6) The term "prime mover elements" is unclear. Could the authors please consider using a more precise term such as "key regulators," "primary drivers," or "major contributing factors" to improve understanding?

Answer: We agree that "prime mover elements" is ambiguous. We have replaced it with a more precise term and revised the sentence for clarity. In the revised text we use "primary driver".

21st Oct 2025

Dear Dr. Zhao,

Thank you for the submission of your revised manuscript to EMBO Molecular Medicine. We have now heard back from the two referees who agreed to re-evaluate your manuscript. As you will see from the report below, both referees are critical regarding the Western blot in Figure EV2, while referee #1 is concerned about the accuracy of the surface plasmon resonance experiments and referee #3 about limited mechanistic insight and the interpretation of the cell viability assay. Please do the following to address referees' criticism:

- Quantify Western blot in Figure 2L.
- Please quantify all Western blots (Figures 1C, 2E and H, 3F, 4C, 5G, EV2L and EV6A) using at least 3 biological replicates and provide the n in the legend.
- Repeat surface plasmon resonance using DNA as a positive control.
- Discuss limited mechanistic insight and clarify the interpretation of the cell viability assay.

Acceptance or rejection of the manuscript will depend on the completeness of your responses included in the next, final version of the manuscript. For this reason, and to save you from any frustrations in the end, I would strongly advise against returning an incomplete revision.

In addition, please amend the following:

1) Authors: Please provide institutional email addresses for Yang Zhao and Xuefei Guo.

2) In the main manuscript file, please do the following:

- Please address all comments suggested by our data editors listed below:

o Data availability assay:

1. Please note that the specific URL for PRJNA1261098 dataset is not provided in the data availability statement.

o Figure legends:

1. Please note that the exact p values are not provided in the legends of figures 1D, E, F, G, H, I, J, K; 2D, E, F, G, H, J, K; 3D, E, G, H, I, N, J, N; 4A, B, I, J; 5E, F, H, I, J, K, L, M.

2. Please indicate the statistical test used for data analysis in the legends of figures 1D, E, F, G, H, I, J, K, 3D, E, G, H, I, N, J, N; 4A, B, I, J; 5B, E, F, H, I, J, K, L, M.

3. Please note that information related to n is missing in the legends of figures 1D-F; 2C, D, E, F, G, H, J, K; 3D, E, G, H, I, N, J, N; 4A, B, I, J; 5E, F, H, I, J, K, L, M.

4. Please note that scale bar and its definition are missing for figure 1B.

- Please make sure that the figures are cited in the correct sequential order. Currently, Figure EV6H is called out before Figure EV1, while Figure EV6D-E and Figure EV 7A-C are called out before Figures EV2-5, etc.

- In Methods, provide the antibody dilutions that were used for each antibody.

- In Methods, add the following paragraph:

Graphics:

(some of the... OR Figure #... OR synopsis) Graphics were created with BioRender.com.

- Please include structured Methods section that includes a Reagents and Tools Table (should be uploaded as a separate file) followed by a Methods and Protocols section. More information on how to adhere to this format as well as downloadable templates (.docx) for the Reagents and Tools Table can be found in our author guidelines:

<https://www.embopress.org/page/journal/17574684/authorguide#structuredmethods>

An example of a paper with Structured Methods can be found here:

<https://www.embopress.org/doi/full/10.1038/s44320-024-00037-6#sec-4>

- Please provide "Disclosure Statement & Competing Interests". We updated our journal's competing interests policy in January 2022 and request authors to consider both actual and perceived competing interests. Please review the policy

<https://www.embopress.org/competing-interests> and update your competing interests if necessary.

- Author contributions: Please remove it from the manuscript and specify author contributions in our submission system. CRediT has replaced the traditional author contributions section because it offers a systematic machine-readable author contributions format that allows for more effective research assessment. You are encouraged to use the free text boxes beneath each contributing author's name to add specific details on the author's contribution. More information is available in our guide to authors:

<https://www.embopress.org/page/journal/17574684/authorguide#authorshipguidelines>

- Indicate in legends number and nature of replicates and exact p= values, not a range, along with the statistical test used. To keep the figures "clear" some authors found providing an Appendix table Sx with all exact p-values preferable. You are welcome to do this if you want to.

- In data availability statement please remove the following text "All data are available in the main text or the supplementary materials. Source data are included in this published article."

- Data availability statement should contain information about raw data from large-scale datasets like ATAC-seq and RNA-seq

that should be deposited in one of the relevant databases and made freely available prior the publication of the manuscript. Use the following format to report the accession number of your data:

[data type]: [full name of the resource] [accession number/identifier] ([doi or URL or identifiers.org/DATABASE:ACCESSION])

3) Acknowledgments:

- Please provide information about the sources of funding and make sure that information about all sources of funding are complete in both our submission system and in the manuscript.

- Please remove BioRender reference "We thank BioRender for their assistance in creating all schematic illustrations."

4) Tables: Please rename the suppl. tables to "Table EV1" and "Table EV2", add legends to the top of the page of the excel files and update their callouts in the main text.

5) The Paper Explained: Please provide "The Paper Explained" and add it to the main manuscript text. Please check "Author Guidelines" for more information. <https://www.embopress.org/page/journal/17574684/authorguide#researcharticleguide>

6) Synopsis: Every published paper now includes a 'Synopsis' to further enhance discoverability. Synopses are displayed on the journal webpage and are freely accessible to all readers. They include separate synopsis image and synopsis text.

- Synopsis image: Please remove it from the manuscript and upload it as a high-resolution jpeg file 550 px-wide x 300-600 pixels high.

- Synopsis text: Please provide a short standfirst (maximum of 300 characters, including space) as well as 2-5 one sentence bullet points that summarise the paper as a .doc file. Please write the bullet points to summarise the key NEW findings. They should be designed to be complementary to the abstract - i.e. not repeat the same text. We encourage inclusion of key acronyms and quantitative information (maximum of 30 words / bullet point). Please use the passive voice.

7) As part of the EMBO Publications transparent editorial process (see our Editorial at

<http://embomolmed.embopress.org/content/2/9/329>), EMBO Molecular Medicine will publish online a Review Process File (RPF) to accompany accepted manuscripts. This file will be published in conjunction with your paper and will include the anonymous

referee reports, your point-by-point response and all pertinent correspondence relating to the manuscript. Let us know whether you agree with the publication of the RPF and as here, if you want to remove or not any figures from it prior to publication.

8) Please provide a point-by-point letter INCLUDING my comments as well as the reviewer's reports and your detailed responses (as Word file).

I look forward to reading a new revised version of your manuscript as soon as possible.

Yours sincerely,

Zeljko Durdevic

Zeljko Durdevic
Senior Editor
EMBO Molecular Medicine

*** Instructions to submit your revised manuscript ***

1) a .docx formatted version of the manuscript text (including Figure legends and tables)

2) Separate figure files*

3) supplemental information as Expanded View and/or Appendix. Please carefully check the authors guidelines for formatting Expanded view and Appendix figures and tables at <https://www.embopress.org/page/journal/17574684/authorguide#expandedview>

4) a letter INCLUDING the reviewer's reports and your detailed responses to their comments (as Word file).

5) The paper explained: EMBO Molecular Medicine articles are accompanied by a summary of the articles to emphasize the major findings in the paper and their medical implications for the non-specialist reader. Please provide a draft summary of your article highlighting

6) Author contributions: the contribution of every author must be detailed in a separate section.

7) EMBO Molecular Medicine now requires a complete author checklist (<https://www.embopress.org/page/journal/17574684/authorguide>) to be submitted with all revised manuscripts. Please use the checklist as guideline for the sort of information we need WITHIN the manuscript. The checklist should only be filled with page numbers where the information can be found. This is particularly important for animal reporting, antibody dilutions (missing) and exact values and n that should be indicated instead of a range.

8) Every published paper now includes a 'Synopsis' to further enhance discoverability. Synopses are displayed on the journal webpage and are freely accessible to all readers. They include a short stand first (maximum of 300 characters, including space) as well as 2-5 one sentence bullet points that summarise the paper. Please write the bullet points to summarise the key NEW findings. They should be designed to be complementary to the abstract - i.e. not repeat the same text. We encourage inclusion of key acronyms and quantitative information (maximum of 30 words / bullet point). Please use the passive voice. Please attach these in a separate file or send them by email, we will incorporate them accordingly.

You are also welcome to suggest a striking image or visual abstract to illustrate your article. If you do please provide a jpeg file 550 px-wide x 300-600px high.

9) A Conflict of Interest statement should be provided in the main text

10) Please note that we now mandate that all corresponding authors list an ORCID digital identifier. This takes <90 seconds to complete. We encourage all authors to supply an ORCID identifier, which will be linked to their name for unambiguous name identification.

Currently, our records indicate that the ORCID for your account is 0009-0002-1282-8975.

Link Not Available

11) Include a Reagents and Tools Table as part of the Methods section, which can be downloaded from our author guidelines (<https://www.embopress.org/page/journal/17574684/authorguide#structuredmethods>)

Photos 400-800 DPI

*Additional important information regarding figures and illustrations can be found at

**** Reviewer's comments ****

Referee #1 (Remarks for Author):

The authors have done a great job in revising their manuscript. They have addressed most of my concerns, but I still have a few minor points that need clarification.

1. Regarding question 1, the Western blot data show a notably high level of cleaved caspase-3 in the untreated group. This suggests that the baseline cellular state may have been compromised, which raises concerns about the validity of concluding that BT has no effect on apoptosis based on such experimental conditions. The reliability of the results obtained under these conditions is questionable.
2. Concerning the SPR experiments, the concentration gradient used appears suboptimal, as the binding curve does not clearly reach a plateau. This may affect the accuracy of the affinity measurements. Furthermore, using the natural ligand of cGAS, such as DNA (e.g., interferon-stimulatory DNA, ISD), as a positive control would be more appropriate.

Referee #3 (Remarks for Author):

In Supplementary Figure S2L, several Western blot bands appear saturated, which prevents accurate quantitative comparison between samples. The loading control (GAPDH) indicates unequal protein loading, with higher signal intensity in the mock-treated lanes. Consequently, the apparent increase in cleaved caspase-3 and cleaved PARP at BT 50 and 75 μM , followed by a decrease at 100 μM , as well as the similar trend observed for caspase-9, should be interpreted with caution given possible technical artifacts.

Furthermore, while the authors indicate activation of mitochondrial stress, the current viability and Western blot assays do not clearly elucidate the underlying mechanisms, and this part of the study lacks sufficient mechanistic insight. In addition, the effect of the inhibitor on normal cells, such as BMDMs, as assessed by the CCK8 assay, is difficult to interpret: the reported viability at 0 μM is approximately 110%, while at 100 μM it decreases to around 80%. This variability complicates the interpretation of potential cytotoxicity.

Response to the Editor

We would like to thank the Editor for the constructive and detailed comments. Below we provide our point-by-point responses following the exact order of the editorial letter.

1. Quantify Western blot in Figure 2L

Response: We have repeated the Western blot in Figure 2I (previous Figure 2L) using three independent biological replicates and quantified the band intensities with ImageJ. The quantification (mean \pm SEM, $n = 3$) has been added below the blot in the revised figure, and the legend has been updated accordingly.

2. Please quantify all Western blots (Figures 1C, 2E and H, 3F, 4C, 5G, EV2L and EV6A) using at least 3 biological replicates and provide n in the legend.

Response: All listed blots were repeated in ≥ 3 biological replicates. Densitometric quantification (mean \pm SEM) was performed by ImageJ and presented under each blot. The corresponding n values have been indicated in all figure legends.

3. Repeat surface plasmon resonance using DNA as a positive control.

Response: We have re-performed the SPR experiments by including salmon sperm double-stranded DNA as a positive control for cGAS binding. The resulting sensorgrams clearly demonstrated the expected DNA–cGAS interaction, confirming the accuracy and reliability of our experimental setup. The updated data are presented in Figure EV3A, and the detailed procedures have been added to the Methods – Surface plasmon resonance assay section.

4. Discuss limited mechanistic insight and clarify the interpretation of the cell-viability assay.

Response: We updated our discussion and methods.

Revised Sentence in Discussion: While our data demonstrated that BT directly bound cGAS and functionally lowered its DNA-activation threshold, the precise structural basis of this allostery remains to be defined. In particular, the binding site and conformational consequences on the cGAS catalytic core are not yet resolved, and whether BT modulates cGAS-DNA phase behavior or oligomerization is unknown. Likewise, BT-induced mitochondrial stress and cytosolic dsDNA could be mechanistically coupled to HDAC inhibition, but we cannot exclude parallel BT-specific effects. The residual antiviral activity observed in single-pathway knockouts indicates minor contributions from additional sensors or stress pathways that warrant mapping. Future work will combine SAR with orthogonal biophysics (SPR/ITC across BT analogs, cryo-EM with cGAS mutants), targeted mutagenesis outside the DNA-binding surface, and comparative profiling against benchmark HDAC inhibitors to disentangle HDAC-ERV and cGAS-centric contributions. These studies will refine how BT achieves dual-axis activation while favoring TBK1-IRF3 over broad NF- κ B outputs.

Revised sentence in material and methods: We quantified cell viability to exclude non-specific cytotoxicity within the antiviral working range of Betrixaban (MedChemExpress, HY-10268) and to ensure that reduced viral readouts were not driven by loss of viable cells. Cells were seeded in 96-well plates at 2×10^3 cells per well in 100 μ L complete medium and allowed to adhere overnight. The next day, cells were treated with BT at the indicated concentrations with a final DMSO $\leq 0.1\%$ (v/v); vehicle controls received DMSO only. After incubation at 37 $^{\circ}$ C and 5% CO₂, 10 μ L CCK-8 reagent (YEASEN, 40203ES60) was added to each well, mixed gently, and incubated for 1 h. Absorbance at 450 nm (reference 570 nm, when available) was measured on a microplate reader. For each plate, blank wells containing medium plus CCK-8 without cells were

used for background subtraction. Each condition was measured in technical triplicate and repeated in ≥ 3 independent experiments.

We used the CCK-8 assay to exclude overt cytotoxicity within the antiviral dose range and to clarify that BT-mediated restriction is host-driven rather than a consequence of reduced cell number. Because CCK-8 primarily reports cellular dehydrogenase activity, which can be influenced by metabolic reprogramming, we complemented it with apoptosis readouts and observed no induction across the working concentrations and time windows. In parallel, entry assays were unaffected and post-entry viral RNA loads and infectious titers declined, supporting a host-mediated post-entry block. Together, these results indicated that BT's antiviral phenotype was not attributable to cytotoxicity or cytostasis but to innate immune activation. We have clarified this interpretation in the text and now explicitly state that viability measurements were used to rule out non-specific toxicity while viral readouts were normalized to housekeeping controls, ensuring that reductions reflect antiviral responses rather than cell loss.

5. Authors: Please provide institutional email addresses for Yang Zhao and Xuefei Guo.

Response: Institutional addresses have been added:

Yang Zhao: 2311210031@stu.pku.edu.cn

Xuefei Guo: guoxf@stu.pku.edu.cn

6. Data availability assay: Please note that the specific URL for PRJNA1261098 dataset is not provided in the data-availability statement.

Response: We have updated the Data Availability section to explicitly state that the raw sequencing data are deposited in the NCBI Sequence Read Archive (SRA) under the BioProject accession PRJNA1261098, and we now provide the direct URL to the BioProject landing page and the SRA Run Selector. The dataset has been made public prior to publication.

Revised Sentence in Methods:

Data availability

The sequencing datasets produced in this study are available in the following databases:

RNA-Seq data, ATAC-Seq data and CUT&Tag data: Sequence Read Archive PRJNA1261098

(<https://www.ncbi.nlm.nih.gov/bioproject/PRJNA1261098/>).

7. Figure legends:

(1) Exact p values are not provided in multiple figures.

(2) Please indicate the statistical test used.

(3) Information related to n is missing.

(4) Scale bar and definition missing for Figure 1B.

Response: All exact p values and corresponding statistical tests are now listed in Appendix Table S1. n values (biological replicates) have been added in figure legends. A scale bar and its definition have been included in Figure 1B.

8. Please make sure that the figures are cited in the correct sequential order.

Response: All figure callouts have been checked and reordered; inconsistencies such as the early reference to EV6H have been corrected.

9. In Methods, provide the antibody dilutions that were used for each antibody.

Response: Antibody names, catalog numbers, and working dilutions are now listed in the revised Methods section and summarized in the Reagents and Tools Table.

10. In Methods, add the following paragraph: Graphics: (some of the... OR Figure #... OR synopsis) Graphics were created with BioRender.com

Response: We have added a Graphics subsection in the Methods.

Revised Sentence in Methods:

Graphics

Synopsis Graphic was created with BioRender.com.

11. Please include structured Methods section that includes a Reagents and Tools Table ...

Response: The Methods section has been completely reformatted according to EMBO's Structured Methods guidelines. A Reagents and Tools Table (uploaded as a separate .docx file) precedes the detailed Methods and Protocols section.

12. Please provide "Disclosure Statement & Competing Interests"

Response: We have reviewed EMBO Molecular Medicine's updated competing-interests policy (January 2022) and added a "Disclosure Statement & Competing Interests" subsection to the manuscript.

Revised Sentence: Disclosure Statement & Competing Interests

The authors declare no competing interests (financial or non-financial). Specifically, no author holds equity, stock options, paid consultancy, advisory positions, intellectual property rights, or other commercial relationships with entities that could benefit from the publication of this work; no author serves on the editorial board of this journal; and there are no personal, political, or societal interests that could reasonably be perceived to influence the research. All funding sources are listed in the Acknowledgements. Funders had no role in study design, data collection, data analysis, decision to publish, or manuscript preparation.

13. Author contributions: please remove it from the manuscript and specify in the submission system (CRediT).

Response: The author-contribution paragraph has been removed from the main text, and contributions were entered in the CRediT fields within the submission system.

14. Indicate in legends number and nature of replicates and exact p values, along with the statistical test used.

Response: We now specified biological replicate numbers, exact p values, and statistical tests. For transparency, an Appendix Table S1 listing all exact p values has been added.

15. In data-availability statement please remove the following text "All data are available..."

Response: The redundant sentence has been deleted.

16. Data-availability statement should contain information about raw ATAC-seq and RNA-seq data ...

Response: We have added the accession numbers and database links for ATAC-seq, RNA-seq and CUT&TAG seq raw data following EMBO's prescribed format.

Revised Sentence in Methods:

Data availability

The sequencing datasets produced in this study are available in the following databases:

RNA-Seq data, ATAC-Seq data and CUT&Tag data: Sequence Read Archive PRJNA1261098 (<https://www.ncbi.nlm.nih.gov/bioproject/PRJNA1261098/>).

17. Acknowledgments: provide complete funding info and remove BioRender thanks.

Response: The acknowledgment to BioRender has been removed. We have cross-checked the submission system and the manuscript to ensure that all funding sources and grant numbers are complete and consistent. The Acknowledgments section has been revised as follows:

Acknowledgments

This work was supported by the National Key Research and Development Program of China (2021YFC2302602), the Beijing Natural Science Foundation (Z210014), and the National Natural Science Foundation of China (NSFC 8235071080, 3247080250, 31570891, 31872736, 32022028, 81991505, 82201928), awarded to Prof. Fuping You. The funders had no role in study design, data collection and analysis, decision to publish, or preparation of the manuscript.

18. Tables: rename supplementary tables to “Table EV1” and “Table EV2”, add legends, update callouts.

Response: Supplementary tables are now renamed Table EV1 and Table EV2 with legends placed at the top of each Excel sheet; callouts have been corrected.

20. The Paper Explained: please provide and add to main manuscript text.

Response: We have added The Paper Explained section summarizing the problem, main results, and significance, formatted per EMBO guidelines.

The Paper Explained

PROBLEM

Most antivirals act on a single viral target, which can limit spectrum and enable resistance. A complementary approach is to boost the patient’s own innate antiviral sensors, but current tools either stimulate only one pathway (for example, STING alone) or face delivery and safety hurdles. There is also no widely used small-molecule that directly activates cGAS, the DNA sensor that initiates interferon responses. An oral, systemically deliverable agent that engages multiple innate sensors could provide broad protection and clinical flexibility.

RESULTS

We discovered that Betrixaban (BT)—an FDA-approved oral drug—activates innate immunity through two mechanisms. First, BT directly binds and allosterically sensitizes cGAS, increasing cGAMP production and downstream interferon signaling. Second, BT shows HDAC-inhibitory activity that de-represses endogenous retroviruses (ERVs), generating double-stranded RNA that engages RIG-I/MDA5. Across cell types and in mice, BT induces IFN programs and restrains diverse DNA and RNA viruses; genetic loss of TBK1/IRF3 or MAVS/STING abrogates protection, confirming pathway dependence.

IMPACT

BT exemplifies a host-directed, dual-pathway antiviral strategy that could broaden coverage and reduce resistance risk. Because BT is oral and already has a human safety/PK record, repurposing may accelerate clinical translation—as a prophylactic during outbreaks or as an adjuvant to enhance vaccine or oncolytic responses. Its upstream action at cGAS plus ERV-mediated RIG-I/MDA5 engagement suggests strong interferon benefits with potential to limit excessive inflammation when appropriately dosed; this provides a feasible path toward clinically practical immunostimulation.

20. Synopsis: remove from manuscript, upload image, provide text and bullet points.

Response: We have removed the synopsis image from the manuscript and uploaded it separately as a high-resolution PNG. We also provide the requested “Synopsis text” as a standalone .doc file content below, including one standfirst and four one-sentence bullet points highlighting the NEW findings in passive voice.

Standfirst

An oral, FDA-approved drug, Betrixaban (BT), was shown to activate innate antiviral immunity by dual mechanisms, direct cGAS sensitization and HDACi-driven ERV/RIG-I/MDA5 signaling,

providing broad protection in cells and mice.

Bullet points

- BT was shown to directly bind cGAS, lowering the DNA activation threshold and increasing cGAMP and IFN signaling.
- HDAC inhibition by BT was observed, causing ERV de-repression and dsRNA accumulation that engaged RIG-I/MDA5-MAVS and activated TBK1-IRF3.
- Broad antiviral activity was demonstrated against RNA and DNA viruses in multiple cell types and in mice, with reduced viral RNA and titers without cytotoxicity.
- Pathway dependence was validated: protection was diminished in cGAS^{-/-} or STING^{-/-} and lost in MAVS/STING double-knockout cells, confirming dual-axis engagement.

21. Review Process File (RPF): confirm agreement and whether to remove figures.

Response: We agree to the publication of the Review Process File and request no figures to be removed.

24th Nov 2025

Dear Dr. Zhao,

We are pleased to inform you that your manuscript is accepted for publication and is now being sent to our publisher to be included in the next available issue of EMBO Molecular Medicine.

Zeljko Durdevic
Senior Editor
EMBO Molecular Medicine
